# TUT1-catalyzed U6 snRNA 3′-end maturation is essential for RNA splicing and stem cell survival

Yin Fang [1,3], Tong Qiu [1,3], Hong Luo[1,3], Yan Wang [1], Chao Yang[1], Min Wang [1], Qian Dai[1], Wenyue Zheng[2], Rutie Yin[1], Xue Xiao [1] & Qintong Li [1✉]

## Abstract

Post-transcriptional maturation of the U6 snRNA 3′-end, important for spliceosome assembly, is catalyzed by sequential actions of TUT1 and USB1. It is believed that the TUT1-catalyzed oligo(U) tail at the U6 snRNA 3′-end serves merely as a substrate for USB1 to generate a final 2′,3′-cyclic phosphate group to mature the U6 snRNA. However, biallelic inactivation of *TUT1* or *USB1* is linked to distinct human developmental disorders, suggesting that they have different physiological functions. Here, using genetically engineered mouse models, we show that *Tut1* is required to maintain stem cell pools during embryogenesis, whereas unexpectedly *Usb1* is dispensable for this. Loss of *Tut1* weakens the interaction of the U6 snRNA with the Lsm2-8 protein complex, causes defective RNA splicing, and triggers massive DNA damage and subsequent cell death. Splicing defects and cell death can be mitigated by recombinant U6 snRNA containing an oligo(U) tail. We propose that the TUT1-catalyzed oligo(U) tail is essential for splicing and cell proliferation. Further modification of this oligo(U) tail by USB1 is ubiquitous but only functionally required in specific cell types.

**Subject Categories** Development; RNA Biology; Stem Cells & Regenerative Medicine

## Introduction

RNA splicing, the removal of introns from mRNA precursors, is essential for eukaryotic gene expression and cellular function (Rogalska et al, 2023). Its regulation by alternative splicing is well recognized as a major contributor to the high transcriptomic and proteomic complexities of multicellular organisms (Marasco and Kornblihtt, 2023). Dysregulated RNA splicing has been implicated in the pathogenesis of human diseases such as cancer (Bradley and Anczukow, 2023), cardiomyopathy (Gotthardt et al, 2023), and neurodegenerative diseases (Nikom and Zheng, 2023). RNA cleavage and ligation reactions necessary for intron removal are catalyzed by the spliceosome, a large ribonucleoprotein machinery composed of U1, U2, U4, U5 and U6 small nuclear RNA (snRNA) as well as dozens of proteins (Wilkinson et al, 2020). It is well recognized that U6 snRNA forms the catalytic core of the spliceosome to regulate RNA splicing (Wan et al, 2020).

It has been known for more than three decades that the U6 snRNA 3′-end is subject to post-transcriptional modifications, although in vivo functions of these modifications remain poorly understood. After being transcribed by RNA polymerase III (Kunkel et al, 1986; Reddy et al, 1987), nascent U6 snRNA contains 4 uridines encoded by the genome at its 3′-end. Subsequently, this oligo(U) tail can be extended to various lengths up to 20 uridines via post-transcriptional modification (Rinke and Steitz, 1985). Nevertheless, approximately 90% of cellular U6 snRNA contains an oligo(U) tail of 5 uridines, with the 3′-terminal uridine containing a 2′,3′-cyclic phosphate group (Lund and Dahlberg, 1992). Recent biochemical and structural studies have established that terminal uridylyl transferase 1 (TUT1) is responsible for adding the oligo(U) tail at the 3′-end of U6 snRNA (Trippe et al, 2006; Trippe et al, 2003; Yamashita et al, 2017; Yamashita and Tomita, 2023). At this stage, the 3′-end consists of terminal 2′ and 3′ hydroxyl groups (a *cis*-diol). Subsequently, this 3′-end oligo(U) tail is shortened by U6 snRNA biogenesis phosphodiesterase 1 (USB1), resulting in a switch from the *cis*-diol to a 2′,3′-cyclic phosphate group on the terminal uridine of U6 snRNA (Didychuk et al, 2017; Hilcenko et al, 2013; Mroczek et al, 2012; Shchepachev et al, 2012). Importantly, in HeLa cells, knockdown experiments demonstrated that TUT1 and USB1 are the only enzymes for extending and shortening, respectively, the oligo(U) tail at the 3′-end of U6 snRNA (Mroczek et al, 2012). Thus, the formation of the 2′,3′-cyclic phosphate group at the U6 snRNA 3′-end requires consecutive enzymatic activities of TUT1 and USB1.

The understanding of the formation and function of the U6 snRNA 3′-end is largely based on studies in vitro. In the current model, the TUT1-catalyzed oligo(U) tail is viewed as an intermediate step to generate the final 2′,3′-cyclic phosphate

[1]Departments of Obstetrics & Gynecology and Pediatrics, Cancer Center, West China Second University Hospital, Key Laboratory of Birth Defects and Related Diseases of Women and Children, Ministry of Education, Development and Related Diseases of Women and Children Key Laboratory of Sichuan Province, Sichuan University, Chengdu 610041 Sichuan, China. [2]Department of Pediatric Dentistry, West China Second University Hospital, Sichuan University, Chengdu 610041 Sichuan, China. [3]These authors contributed equally: Yin Fang, Tong Qiu, Hong Luo.✉E-mail: liqintong@scu.edu.cn

modification (Didychuk et al, 2018). It is thought that TUT1-catalyzed oligo(U) tail allows binding of SSB/La protein to U6 snRNA to protect newly synthesized U6 snRNA from exonuclease digestion (Rinke and Steitz, 1985; Terns et al, 1992). Subsequently, USB1-catalyzed trimming of this oligo(U) tail results in a 2′,3′-cyclic phosphate group on the terminal uridine of U6 snRNA. U6 snRNA can bind to the Lsm2-8 protein complex without a 2′,3′-cyclic phosphate group in vitro. However, this modification greatly enhances the affinity of Lsm2-8 to U6 snRNA, enabling further assembly of a functional spliceosome for RNA splicing (Achsel et al, 1999; Licht et al, 2008; Montemayor et al, 2018; Montemayor et al, 2020). Thus, 2′,3′-cyclic phosphate modification at the U6 snRNA 3′-end, catalyzed sequentially by TUT1 and then USB1, is thought to be essential for proper RNA splicing and cellular function.

The exact physiological functions of TUT1, USB1 and their associated post-transcriptional modifications of U6 snRNA are unknown. Human genetics studies hinted that TUT1 and USB1 may have distinct functions in vivo. In humans, biallelic *TUT1* inactivation was linked to cortical atrophy, microcephaly, and cerebellar atrophy (Karaca et al, 2015), whereas biallelic *USB1* inactivation was associated with short stature, defective skin and immune system (Mroczek and Dziembowski, 2013; Parajuli et al, 2024). Nevertheless, the causality between TUT1 or USB1 deficiency and these abnormal conditions have not been investigated in animal models.

In the present study, we generated engineered mouse models to investigate the physiological functions of *Tut1* and *Usb1* during mammalian development. We find that *Tut1* knockout causes defective splicing in many genes, leading to massive DNA damage and cell death in stem cells during embryogenesis. Concomitant deletion of *Trp53* (encoding p53 protein) or the restoration of the oligo(U) tail at the U6 snRNA 3′-end can prevent *Tut1* loss-induced cell death. In contrast, *Usb1* is dispensable for the proliferation and differentiation of most embryonic cell lineages. These results demonstrate that, contrary to what was previously assumed, USB1 and thus 2′,3′-cyclic phosphate modification of the U6 snRNA 3′-end, is functionally dispensable in most embryonic cell types until mid-embryogenesis. Instead, the 3′-end oligo(U) tail of the U6 snRNA, catalyzed by TUT1, is essential for RNA splicing and cell proliferation. We discuss the implications of our findings for development and disease.

# Results

## *Tut1* is required for epiblast and embryonic stem cell maintenance

Based on biochemical, structural and cell-based studies, the current model depicts that sequential actions of TUT1 and USB1 catalyze the post-transcriptional maturation of the U6 snRNA 3′-end (Fig. 1A). This model would predict that knockout of *Tut1* or *Usb1* should yield similar phenotypes. To investigate the physiological function of *Tut1*, we attempted to generate conventional *Tut1* knockout mice using CRISPR/Cas9 genome editing technology (Fig. 1B). No viable homozygous *Tut1* knockout offspring (*Tut1*$^{-/-}$) could be obtained (n = 177, P7). Wild-type and heterozygous pups (*Tut1*$^{+/-}$) were born at the expected ratio (Fig. 1C). To establish the

time of lethality, embryos were genotyped (n = 57). Homozygous *Tut1*-null blastocysts were produced at the expected ratio at embryonic day 3.5 (E3.5), suggesting that *Tut1* is not required for the specification of inner cell mass and trophectoderm. However, no homozygous *Tut1*-null conceptuses could be obtained beyond E5.5 (Fig. 1C). Between E3.5 and E5.5, pluripotent epiblast (descendant of the inner cell mass) proliferates intensively, and begins to differentiate into three embryonic germ layers (Tam and Loebel, 2007). These findings demonstrate that *Tut1* is essential for mammalian early embryogenesis and suggest that loss of *Tut1* may compromise epiblast proliferation and/or differentiation.

Embryonic stem cells are fast proliferating, pluripotent cells, and they are generally perceived as the in vitro counterpart of epiblast in vivo (Nichols and Smith, 2009). Despite several attempts to screen for hundreds of clones, we could not obtain *Tut1*-null embryonic stem cells using CRISPR/Cas9 genome editing technology, suggesting that *Tut1* is required to maintain embryonic stem cells. Therefore, we used short-hairpin RNA (shRNA) to knock down *Tut1* expression. Two independent shRNA constructs, targeting different regions of *Tut1*, markedly reduced the number of embryonic stem cells (Fig. 1D). Nevertheless, the remaining cells still formed the typical dome-shaped colonies, and were stained positively for alkaline phosphatase, a marker for embryonic stem cells (Fig. 1E). In addition, these cells expressed core pluripotency transcription factors Pou5f1 (commonly known as Oct4), Sox2 and Nanog at comparable levels as cells treated with scrambled shRNA (Fig. 1F). Taken together, these results indicate that *Tut1* is required for epiblast proliferation and survival during early embryogenesis.

## Conditional knockout of *Tut1* causes profound defects in the embryonic brain

To circumvent early embryonic lethality, we generated *Tut1*-floxed mice (*Tut1*$^{f/f}$) (Fig. 2A). Human genomics studies have linked biallelic inactivating mutations of *TUT1* to cortical atrophy, microcephaly, and cerebellar atrophy (Karaca et al, 2015). Thus, we investigated the role of *Tut1* in mammalian neurodevelopment. *Tut1*-floxed mice were crossed with nestin-Cre mice (Tronche et al, 1999) to conditionally delete *Tut1* in the developing central nervous system (*Tut1*$^{f/f}$;nestin-Cre) (Fig. 2B). Heterozygous pups (*Tut1*$^{f/+}$;nestin-Cre) were born at expected ratios (Fig. 2C). During a 12-month follow-up period, heterozygous mice were healthy without overt defects in growth and reproduction, compared to wild-type mice (*Tut1*$^{f/+}$ and *Tut1*$^{f/f}$ mice were used as controls hereafter) (Fig. EV1A). Histological examinations showed no overt abnormalities in the brain of heterozygous mice (Fig. EV1B). To detect potential defects in the brain of heterozygotes, we carried out several behavioral tests, including the open field task, three-chamber sociability and social novelty test, the elevated plus maze, and the Morris water maze test (Fig. EV1C–F). In all cases, heterozygotes behaved in a similar manner to the wild-type mice. Taken together, these results demonstrate the haplosufficiency of *Tut1* in the brain and overall development.

In contrast, no homozygous offspring (*Tut1*$^{f/f}$;nestin-Cre) could be born (n = 114, P7) (Fig. 2C), suggesting embryonic lethality. At E12.5, the overall morphology of developing brains was indistinguishable between knockout and wild-type embryos (Fig. 2D). However, between E14.5-E18.5, enlarged brain ventricles

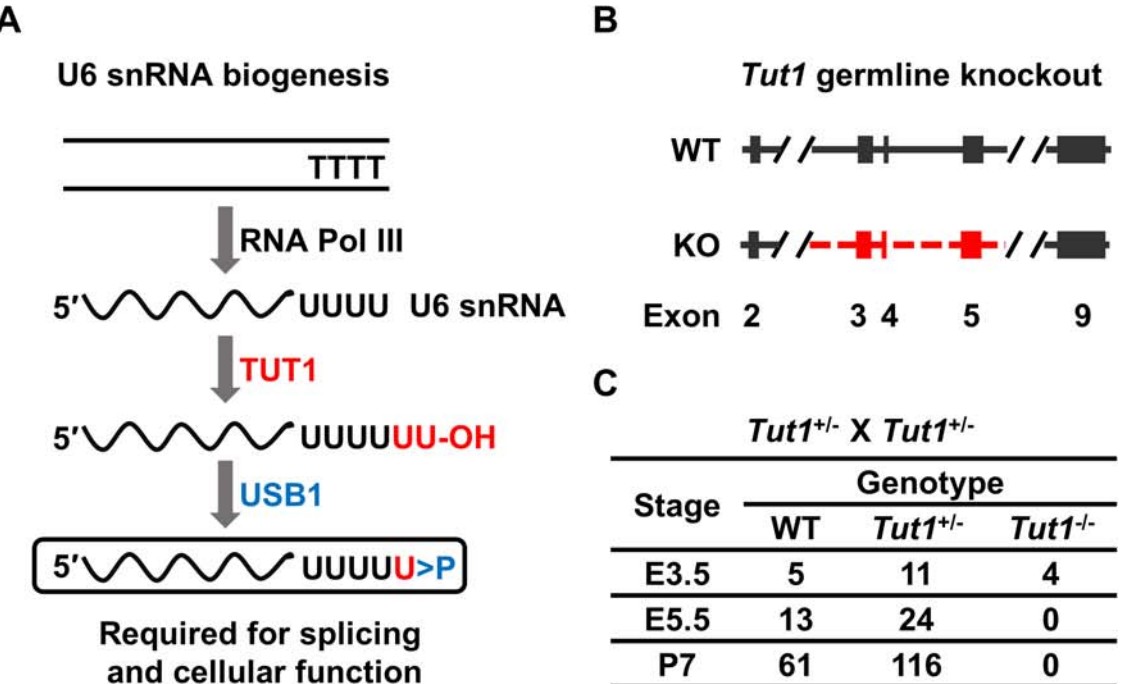

Figure 1. *Tut1* is required for epiblast and embryonic stem cell maintenance.

(A) Previous model of U6 snRNA 3′-end maturation. After being transcribed by RNA polymerase III (RNA Pol III), nascent U6 snRNA contains four Us at its 3′-end (depicted by four Us in black). Then, TUT1 catalyzes the post-transcriptional addition of an oligo(U) tail (depicted by two Us in red), followed by USB1-catalyzed truncation of this tail and the addition of the terminal 2′,3′-cyclic phosphate group (depicted in blue). The mature form of U6 snRNA contains four Us derived from the genome, followed by one post-transcriptionally added U with a 2′,3′-cyclic phosphate group at its 3′-end. This form of U6 snRNA is thought to be essential for proper splicing and cellular function. (B) The strategy to generate a *Tut1* germline knockout (conventional knockout) by CRISPR/Cas9 technology. Exons 3–5 would be deleted in knockout lines. WT, wild-type. KO, knockout. (C) The number of embryos and pups recovered from *Tut1*-heterozygous (*Tut1*$^{+/-}$) intercrosses at the indicated embryonic (E) and postnatal (P) ages. (D) The effect of *Tut1* knockdown on the proliferative rate of murine embryonic cells (mESCs). Two independent short-hairpin RNA (shRNA) constructs were used (T1 and T2). Scrambled shRNA (scr) was used as a control. The data represent mean ± SEM from three independent experiments ($n = 3$). Statistical significance was assessed using the two-sided unpaired Student's *t*-test. (E) Alkaline phosphatase (AP) staining of mESCs with or without *Tut1* knockdown. Positive AP staining is a marker of self-renewing mESCs. Scale bar: 100 μm. (F) Protein blot analysis of indicated proteins after *Tut1* knockdown for 3 and 6 days. Oct4, Sox2 and Nanog are well-established core transcription factors required for mESC self-renewal and pluripotency. Source data are available online for this figure.

accompanied by intracerebral hemorrhage could be reproducibly detected only in knockout embryos ($n = 19$) (Fig. 2D). The appearance of enlarged brain ventricles was caused by two structural abnormalities. One was the significantly thinner dorsal as well as medial palliums (DP and MP), and the other was the disappearance of the medial and lateral ganglionic eminence regions (MGE and LGE) (Fig. 2E). These anatomic regions are where nestin-Cre is preferentially expressed at E14.5 (Liang et al, 2012), suggesting that *Tut1* deletion-induced defects in the developing brain may be caused by cell-intrinsic mechanisms. Marker analyses revealed comparable numbers of Sox2$^+$ neural stem and Tbr2$^+$ intermediate progenitor cells between wild-type and *Tut1*$^{f/f}$;nestin-Cre brains at E12.5. At E14.5, profound loss of Sox2$^+$ neural stem and Tbr2$^+$ intermediate progenitor cells occurred in *Tut1*$^{f/f}$;nestin-Cre telencephalon ($n = 15$) (Fig. 2F). Consistently, the number of Dcx$^+$ differentiating neuronal cells was also markedly reduced at E14.5 (Fig. 2G). These results establish that *Tut1* is required for the development of embryonic cerebral cortex, and support biallelic *TUT1* inactivation as the cause of cortical hypoplasia observed in humans (Karaca et al, 2015).

## *Tut1* is essential to maintain the neural stem cell pool during embryogenesis

Notably, loss of *Tut1* markedly reduced, but did not eliminate Sox2$^+$ neural stem and Tbr2$^+$ intermediate progenitor cells at E14.5 (Fig. 2F). We considered two possibilities to account for this observation. First, this may suggest that a subset of neural stem and progenitor cells do not require *Tut1* for survival. Secondly, this may suggest that *Tut1* is not sufficiently deleted by nestin-Cre in these surviving neural stem and progenitor cells at E14.5. The latter explanation is plausible because the recombination activities of nestin-Cre are modest in neural stem and progenitor cells at E12.5 (Liang et al, 2012).

To differentiate these two possibilities, *Tut1*-floxed mice were crossed with Emx1-Cre mice. Like nestin-Cre, Emx1-Cre also functions in the dorsal and medial pallium of the developing neocortex (Gorski et al, 2002) (Fig. 2H). However, unlike nestin-Cre, Emx1-Cre fully functions in neural stem cells before E10.5 (Gorski et al, 2002). Thus, if a subset of neural stem and progenitor cells do not require *Tut1* for survival, one would expect that *Tut1* deletion using Emx1-Cre or nestin-Cre should result in similar defects in the dorsal and medial pallium of the developing neocortex. However, if remaining stem and progenitor cells are a resultant from insufficient activities of nestin-Cre, one would expect that *Tut1* deletion using Emx1-Cre may eliminate all stem and progenitor cells by E14.5.

Conditional ablation of *Tut1* using Emx1-Cre (*Tut1*$^{f/f}$;Emx1-Cre) resulted in high penetrance of embryonic lethality between E16.5 and E18.5 (Fig. 2I). Occasionally, *Tut1*$^{f/f}$;Emx1-Cre pups could be born, but inevitably died within 3 weeks (Fig. 2I). The growth of *Tut1*$^{f/f}$;Emx1-Cre mice were severely stunted at postnatal day (P) 14 (Fig. 2J). These mice also exhibited an unsteady gait (Fig. 2K; Movie EV1). Histological assessment showed that the dorsal and medial regions of neocortex were completely missing in these mice at P14 (Fig. 2L), mirroring the expression pattern of Emx1-Cre (Fig. 2H). Because Emx1-Cre functions at E10.5 and high penetrance of embryonic lethality occurred around E16.5, we analyzed the developing neocortex at E11.5, E12.5 and E14.5. At E11.5, the dorsal and medial palliums were morphologically indistinguishable between wild-type and *Tut1*$^{f/f}$;Emx1-Cre embryos, and the number of Pax6$^+$ stem and progenitor cells were comparable (upper panel, Fig. 2M). At E12.5, the number of stem and progenitor cells were dramatically reduced in *Tut1*$^{f/f}$;Emx1-Cre brains (middle panel, Fig. 2M). At E14.5, the dorsal and medial palliums of the neocortex were essentially eliminated in *Tut1*$^{f/f}$;Emx1-Cre embryos (lower panel, Fig. 2M). The disappearance of Tbr2$^+$ intermediate progenitor cells (Fig. 2N) and Dcx$^+$ differentiating neuronal cells (Fig. 2O) mirrored that of Pax6$^+$ stem and progenitor cells in *Tut1*$^{f/f}$;Emx1-Cre brains. Thus, *Tut1* is required to maintain all neural stem cells in the developing neocortex.

To assess whether *Tut1* is required for the maintenance of mature neurons after birth, *Tut1*-floxed mice were crossed with Camk2a-Cre (also known as T29-1) mice. The recombination activities of Camk2a-Cre peak during the third postnatal week in the CA1 pyramidal cell layer of the hippocampus (Fig. 2P) (Tsien et al, 1996). In contrast to the lethality in *Tut1*$^{f/f}$;nestin-Cre and *Tut1*$^{f/f}$;Emx1-Cre embryos, *Tut1*$^{f/f}$;Camk2a-Cre homozygous offspring could be obtained at the expected ratio (Fig. 2Q). At 10 months of age, *Tut1*$^{f/f}$;Camk2a-Cre mice were healthy (Fig. 2R), and the histological features of their hippocampi were indistinguishable from those of wild-type mice (Fig. 2S). The efficiency of *Tut1* deletion was around 90% as determined by qPCR (Fig. EV1G). Taken together, we conclude that *Tut1* is essential for the maintenance of neural stem cells during embryogenesis.

## Loss of *Tut1* causes massive DNA damage and triggers *Trp53*-dependent cell death in neural stem cells

A complete loss of Emx1-expressing cells in *Tut1*$^{f/f}$;Emx1-Cre brains suggests that *Tut1* deletion may cause cell death and/or proliferation arrest. In *Tut1*$^{f/f}$;Emx1-Cre embryos, the dorsal and

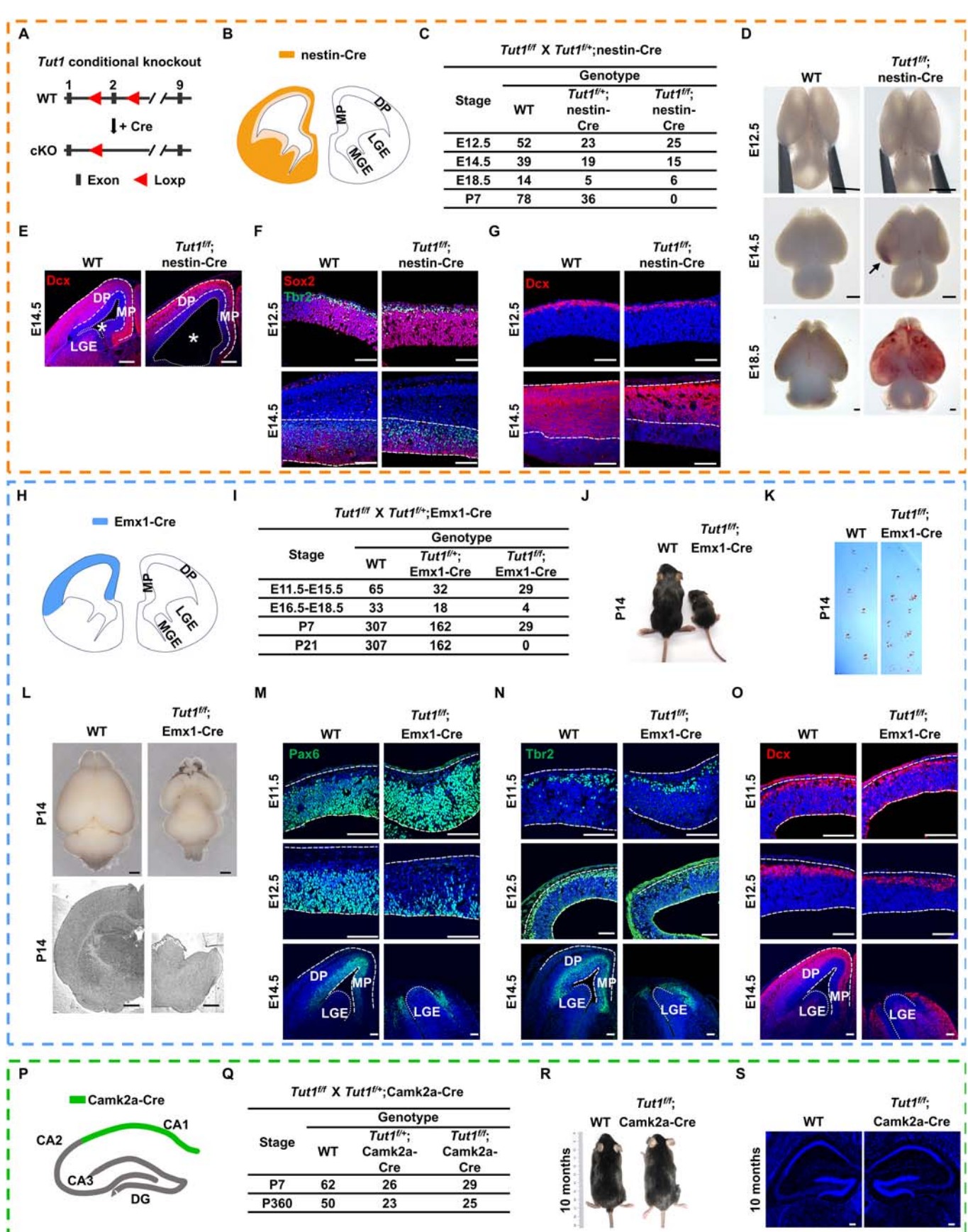

**Figure 2. Conditional knockout of *Tut1* causes profound defects in the embryonic brain.**

(A) The strategy to generate *Tut1*-floxed mice using CRISPR/Cas9 technology in order to produce conditional knockout mouse models. cKO, conditional knockout. (B–G) Phenotypic analyses of *Tut1*$^{f/f}$;nestin-Cre embryonic brains. (B) Schematic presentation of the expression pattern of nestin-Cre in murine embryonic neocortex. MP medial pallium, DP dorsal pallium, LGE lateral ganglionic eminence, MGE medial ganglionic eminence. Of note, the recombination activity of nestin-Cre at E12.5 is minimal in the zone of neural stem and progenitor cells (denoted by light orange), but is significant in the zone of differentiating neuronal cells (denoted by dark orange). (C) Genotype analyses of embryos and pups generated from crossing of *Tut1*$^{f/+}$;nestin-Cre and *Tut1*$^{f/f}$ mice. Wild-type (WT) denotes both *Tut1*$^{f/+}$ and *Tut1*$^{f/f}$ genotypes. (D) Dorsal views of wild-type (WT) and *Tut1* knockout (*Tut1*$^{f/f}$;nestin-Cre) brains at E12.5 ($n = 20$), E14.5 ($n = 15$), and E18.5 ($n = 4$). Intracerebral hemorrhage could be reproducibly detected around E14.5 (denoted by the arrow), and became progressively widespread at E18.5. Scale bar: 500 μm. (E) Coronal sections of E14.5 wild-type (WT) and *Tut1* knockout (*Tut1*$^{f/f}$;nestin-Cre) brains ($n = 4$). Immunofluorescence of doublecortin (Dcx, red) denotes neuroblasts and immature neurons. The Dcx-negative but 4′,6′-diamidino-2-phenylindole (DAPI)-positive zone (blue) contains neural stem and progenitor cells. Asterisk (*) denotes brain ventricles. Note that the brain ventricle is enlarged in the *Tut1* knockout brain. MP, medial pallium. DP dorsal pallium, LGE lateral ganglionic eminence. Scale bar: 200 μm. (F) Comparison of neural stem and progenitor cells in neocortices of wild-type (WT) and *Tut1* knockout (*Tut1*$^{f/f}$;nestin-Cre) brains. A representative section of the dorsal pallium was shown. Immunofluorescence of Sox2 protein denotes neural stem and progenitor cells (red), and that of Tbr2 protein denotes neuronal intermediate progenitor cells (green). Note that the number of Sox2- and Tbr2-positive cells were comparable at E12.5 between WT and *Tut1*$^{f/f}$;nestin-Cre brains ($n = 3$), whereas their numbers were much reduced at E14.5 in *Tut1*$^{f/f}$;nestin-Cre brains (demarcated by dashed white lines) ($n = 4$). Scale bar: 100 μm. (G) Comparison of differentiating neuronal cells in neocortices of wild-type (WT) and *Tut1* knockout (*Tut1*$^{f/f}$;nestin-Cre) brains. Note that the number of Dcx-positive cells were comparable at E12.5 between WT and *Tut1*$^{f/f}$;nestin-Cre brains ($n = 3$), whereas their numbers were much reduced at E14.5 in *Tut1*$^{f/f}$;nestin-Cre brains (demarcated by dashed white lines) ($n = 4$). Scale bar: 100 μm. (H–O) Phenotypic analyses of *Tut1*$^{f/f}$;Emx1-Cre embryos and pups. (H) Schematic presentation of the expression pattern of Emx1-Cre in murine embryonic neocortex. Of note, Emx1-Cre functions in neural stem and progenitor cells at E10.5, whereas little nestin-Cre activity can be detected in neural and stem cells at E12.5. MP medial pallium, DP dorsal pallium, LGE lateral ganglionic eminence, MGE medial ganglionic eminence. (I) Genotype analyses of embryos and pups generated from crossing of *Tut1*$^{f/+}$;Emx1-Cre and *Tut1*$^{f/f}$ mice. Wild-type (WT) denotes both *Tut1*$^{f/+}$ and *Tut1*$^{f/f}$ genotypes. Of note, all *Tut1*$^{f/f}$;Emx1-Cre mice died by postnatal day (P) 21. (J) Littermates of wild-type (WT) and *Tut1* knockout (*Tut1*$^{f/f}$;Emx1-Cre) mice at P14 ($n = 29$). (K) Gait differences between wild-type (WT) and *Tut1*$^{f/f}$;Emx1-Cre mice. Representative walking footprint patterns of P14 mice were shown ($n = 3$). (L) Comparison of P14 wild-type (WT) and *Tut1*$^{f/f}$;Emx1-Cre brains. Both dorsal views (upper panel) and coronal sections (lower panel) were shown. Note that the medial and dorsal palliums were completely missing in *Tut1*$^{f/f}$;Emx1-Cre brains ($n = 3$). Scale bar: 500 μm. (M–O) Comparison of neural stem, progenitor and differentiating cells in neocortices of wild-type (WT) and *Tut1* knockout (*Tut1*$^{f/f}$;Emx1-Cre) brains at E11.5 ($n = 3$), E12.5 ($n = 3$), and E14.5 ($n = 3$). A representative section of the dorsal pallium was shown for each timepoint. Immunofluorescence of Pax6 protein denotes neural stem and progenitor cells (green) (M), that of Tbr2 protein denotes progenitor cells (N), and that of Dcx protein denotes differentiating cells (O). Note that the number of Pax6-, Tbr2- and Dcx-positive cells were comparable at E11.5 between WT and *Tut1*$^{f/f}$;Emx1-Cre brains. Their numbers were much reduced at E12.5 and eliminated in the medial and dorsal palliums of *Tut1*$^{f/f}$;Emx1-Cre brains at E14.5. White dashed and dotted lines indicate the cortical and LGE areas, respectively. Scale bar: 100 μm. (P–S) Phenotypic analyses of *Tut1*$^{f/f}$;Camk2a-Cre mice. (P) Schematic presentation of the expression pattern of Camk2a-Cre in the murine hippocampus. Of note, the recombination activity of Camk2a-Cre starts at P19 and peaks at P29 in the CA1 subregion of the hippocampus (denoted by green). (Q) Genotype analyses of pups generated from crossing of *Tut1*$^{f/+}$;Camk2a-Cre and *Tut1*$^{f/f}$ mice. P postnatal day. Wild-type (WT) denotes both *Tut1*$^{f/+}$ and *Tut1*$^{f/f}$ genotypes. (R) Littermates of wild-type (WT) and *Tut1* knockout (*Tut1*$^{f/f}$;Camk2a-Cre) mice at 10 months ($n = 18$). (S) Comparison of coronal sections of 10 months WT and *Tut1*$^{f/f}$;Camk2a-Cre brains ($n = 3$). Scale bar: 100 μm. Source data are available online for this figure.

medial palliums of the developing neocortex were morphologically normal at E11.5, but became much thinner at E12.5 (Fig. 2M–O). At E11.5, bromodeoxyuridine (BrdU) labeling and Ki67 immunochemistry demonstrated that cell proliferation status was comparable between wild-type and *Tut1*$^{f/f}$;Emx1-Cre dorsal and medial palliums ($n = 3$) (Fig. 3A). At E11.5, the level of cleaved caspase 3 (c-Casp3), a marker labeling dying cells, was minimal in wild-type, but started to increase in *Tut1*$^{f/f}$;Emx1-Cre dorsal and medial palliums. Interestingly, increased c-Casp3 signals were scattered predominantly in the proliferative zone containing Sox2$^+$ stem and progenitor cells (Fig. 3B), but not in Dcx$^+$ differentiating cells (Fig. 3C). By E12.5, the level of c-Casp3 was dramatically increased in dorsal and medial palliums of *Tut1*$^{f/f}$;Emx1-Cre neocortices. Notably, the signal of c-Casp3 was overlapped with the expression pattern of Emx1-Cre (Fig. 3D). These results demonstrate that cell death is the primary cause of disappearing dorsal and medial palliums in *Tut1*$^{f/f}$;Emx1-Cre neocortices. The inhibition of RNA splicing machinery can trigger DNA damage. Indeed, the intensities of γH2AX and TUNEL, two markers of genomic damage, were markedly increased (Fig. 3E,F).

Cell death was initiated in neural stem cells expressing Emx1-Cre (Fig. 3B–D), indicating that *Tut1* loss triggers cell-autonomous death. To corroborate this notion, we carried out the neurosphere assay. Wild-type neural stem cells derived from E11.5 dorsal pallium continued to proliferate as neurospheres in vitro. In contrast, neural stem cells derived from E11.5 *Tut1*$^{f/f}$;Emx1-Cre dorsal pallium were rapidly lost within 4 days in the neurosphere

assay (Fig. 3G). Thus, *Tut1* loss causes neural stem cell death by cell-intrinsic mechanisms.

To further strengthen the causal relationship between cell death and the missing neocortex, we investigated the underlying molecular mechanism. E12.5 wild-type and *Tut1*$^{f/f}$;Emx1-Cre dorsal and medial palliums were dissected, respectively, and subjected to transcriptomic profiling by RNA-seq. Gene set enrichment analysis (GSEA) indicated the activation of the p53 signaling pathway in *Tut1*$^{f/f}$;Emx1-Cre palliums (Fig. 3H). To assess whether *Tut1* deletion-induced cell death is dependent on *Trp53* (encoding p53 protein), *Tut1*-floxed mice were crossed with *Trp53*-floxed mice (Fig. 3I) to generate double-floxed mice (*Tut1*$^{f/f}$;*Trp53*$^{f/f}$). These mice were then crossed with Emx1-Cre to simultaneously delete *Tut1* and *Trp53* in the dorsal and medial palliums of the developing neocortex (*Tut1*$^{f/f}$;*Trp53*$^{f/f}$;Emx1-Cre). At E14.5, *Tut1* deletion resulted in the elimination of the dorsal and medial pallium (Fig. 2M–O). In sharp contrast, the missing pallium structure in E14.5 *Tut1*$^{f/f}$;Emx1-Cre brain was fully restored in *Tut1*$^{f/f}$;*Trp53*$^{f/f}$;Emx1-Cre brains (Fig. 3J). Consistently, neural stem cells derived from *Tut1*$^{f/f}$;*Trp53*$^{f/f}$;Emx1-Cre dorsal and medial palliums could be expanded in vitro ($n = 4$) (Fig. 3K), unlike those derived from *Tut1*$^{f/f}$;Emx1-Cre neocortex (Fig. 3G). These *Tut1*$^{f/f}$;*Trp53*$^{f/f}$;Emx1-Cre neural stem cells exhibited a comparable proliferative rate as those derived from wild-type neocortex in Incucyte live-cell imaging assay ($n = 3$) (Fig. 3L). One trivial explanation for this genetic rescue was that somehow *Tut1* and *Trp53* were not fully deleted. To exclude this possibility, we genotyped cells cultured at the end of

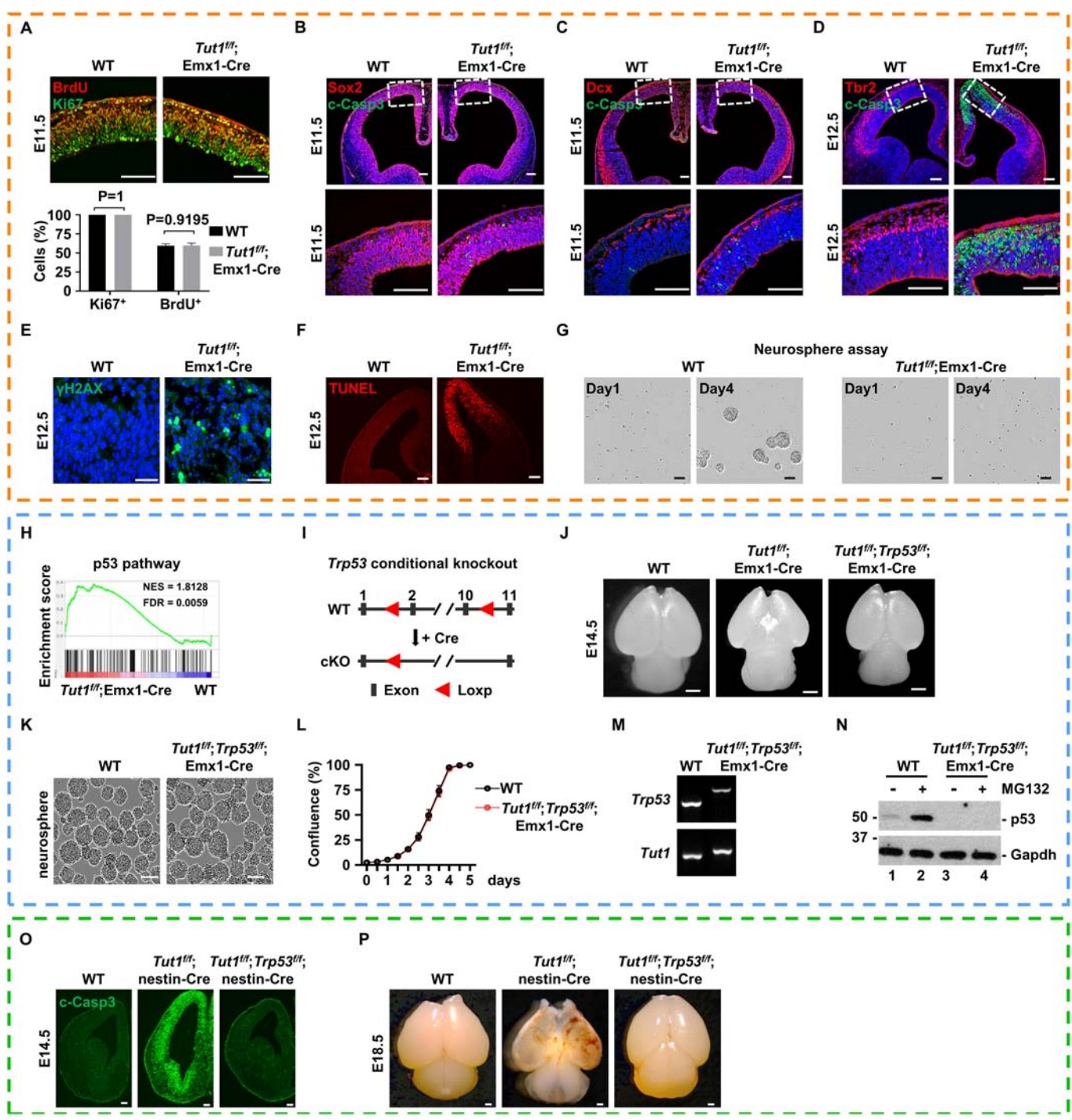

the neurosphere assay. Both *Tut1* and *Trp53* remained deleted in *Tut1^{f/f}*;*Trp53^{f/f}*;Emx1-Cre neural stem cells (Fig. 3M). In addition, we treated these cells with MG132, a potent proteasome inhibitor, to stabilize p53 protein. p53 protein was rapidly accumulated by MG132 treatment in wild-type cells, but remained undetectable in *Tut1^{f/f}*;*Trp53^{f/f}*;Emx1-Cre cells (Fig. 3N). These results support the notion that the missing pallium in *Tut1^{f/f}*;Emx1-Cre is indeed caused by the death of neural stem cells.

Furthermore, we examined whether simultaneous *Trp53* deletion could prevent the defects in *Tut1^{f/f}*;nestin-Cre embryonic

brains. Like in *Tut1^{f/f}*;Emx1-Cre brains (Fig. 2M–O), the dorsal and medial palliums of *Tut1^{f/f}*;nestin-Cre brains is largely missing due to substantial death of neural stem cells (Fig. 2D–G). Notably, beyond this similarity, a prominent cerebral hemorrhage was only present in *Tut1^{f/f}*;nestin-Cre in brains (Fig. 2D). This phenotype is likely caused by the fact that the recombination activity of nestin-Cre is more widespread in additional cell types and anatomic regions than Emx1-Cre. Indeed, the medial and lateral ganglionic eminence regions (MGE and LGE) were also missing in *Tut1^{f/f}*;nestin-Cre embryonic brains (Fig. 2E). Remarkably, simultaneous *Trp53*

**Figure 3. Loss of *Tut1* causes massive DNA damage and triggers *Trp53*-dependent cell death in neural stem cells.**

(A–G) Phenotypic analyses of cell proliferation and cell death in *Tut1^f/f*;Emx1-Cre brains. (A) Analysis of cellular proliferation in E11.5 wild-type (WT) and *Tut1* knockout (*Tut1^f/f*;Emx1-Cre) brains. A representative section of the dorsal pallium was shown (upper panel) (*n* = 3). Note that at this developmental stage, most cells are neural stem and progenitor cells. The number of Ki67-positive and BrdU-positive were quantified (lower panel). The data represent mean ± SEM. Statistical significance was assessed using the two-sided unpaired Student's *t*-test. Scale bar: 100 μm. (B, C) Analysis of cell death in E11.5 wild-type (WT) and *Tut1* knockout (*Tut1^f/f*;Emx1-Cre) brains (*n* = 3). Immunofluorescence of cleaved caspase 3 protein (c-Casp3) indicates cell death. Note that c-Casp3 signals were initially detected in Sox2-positive (B) but not in Dcx-positive cells (C) at E11.5. Coronal sections were shown in the upper panel, and areas demarcated by white dashed lines were amplified in the lower panel. Scale bar: 100 μm. (D–F) Analyses of cell death in E12.5 wild-type (WT) and *Tut1* knockout (*Tut1^f/f*;Emx1-Cre) brains (*n* = 3). Note that at E12.5, the staining signals of c-Casp3 (D) (scale bar: 100 μm), γH2AX (E) (scale bar: 25 μm) and TUNEL (F) (scale bar: 100 μm) were widespread, and mirrored the expression pattern of Emx1-Cre (Fig. 2H). (G) Analysis of the proliferation of neural stem cells derived from E11.5 wild-type (WT) and *Tut1* knockout (*Tut1^f/f*;Emx1-Cre) brains by neurosphere assay (*n* = 4). Scale bar: 100 μm. (H–N) Simultaneous *Trp53* deletion fully restores neural stem cells and neocortex development in *Tut1^f/f*;Emx1-Cre brains. (H) Analysis of dysregulated pathways in E12.5 *Tut1* knockout (*Tut1^f/f*;Emx1-Cre) neocortex using the hallmark gene set collection in the Molecular Signatures Database (MSigDB). NES normalized enrichment score, FDR false discovery rate. (I) The strategy to generate *Trp53*-floxed mice using CRISPR/Cas9 technology. cKO conditional knockout. (J) Dorsal views of E14.5 wild-type (WT), *Tut1* knockout (*Tut1^f/f*;Emx1-Cre), *Tut1* and *Trp53* double knockout (*Tut1^f/f*;*Trp53^f/f*;Emx1-Cre) brains (*n* = 4). Scale bar: 500 μm. (K) Phase contrast images of cultured neural stem cells derived from E12.5 wild-type and *Tut1^f/f*;*Trp53^f/f*;Emx1-Cre neocortices (*n* = 4), respectively. Scale bars: 100 μm. (L) Incucyte live-cell imaging assay to assess the proliferation rate of neural stem cells derived, respectively, from wild-type (WT) and *Tut1^f/f*;*Trp53^f/f*;Emx1-Cre neocortices (*n* = 3). (M) Genotype analyses of *Trp53* and *Tut1* in wild-type (WT) and *Tut1^f/f*;*Trp53^f/f*;Emx1-Cre neural stem cells. (N) Analysis of p53 protein expression in neural stem cells derived, respectively, from wild-type (WT) and *Tut1^f/f*;*Trp53^f/f*;Emx1-Cre neocortices (*n* = 3). MG132 is a proteasomal inhibitor used to stabilize p53 protein. (O, P) Simultaneous *Trp53* deletion fully restores neocortex development in *Tut1^f/f*;nestin-Cre brains. (O) Analysis of cell death in E14.5 wild-type (WT), *Tut1* knockout (*Tut1^f/f*;nestin-Cre), *Tut1* and *Trp53* double knockout (*Tut1^f/f*;*Trp53^f/f*;nestin-Cre) brains via immunofluorescence of cleaved caspase 3 (c-Casp3) protein (*n* = 3). Scale bar: 100 μm. (P) Dorsal views of E18.5 wild-type (WT), *Tut1* knockout (*Tut1^f/f*;nestin-Cre), *Tut1* and *Trp53* double knockout (*Tut1^f/f*;*Trp53^f/f*;nestin-Cre) brains (*n* = 5). Scale bar: 500 μm. Source data are available online for this figure.

deletion (*Tut1^f/f*;*Trp53^f/f*;nestin-Cre) prevented cell death (Fig. 3O), eliminated cerebral hemorrhage, and fully restored missing structures in E18.5 *Tut1^f/f*;nestin-Cre embryonic brains (Fig. 3P). Taken together, we conclude that loss of *Tut1* causes massive DNA damage, and triggers *Trp53*-dependent cell death in neural stem cells.

## The *Tut1*-catalyzed oligo(U) tail at the 3′-end of U6 snRNA is required for RNA splicing and the survival of neural stem cells

We performed a neurosphere assay to evaluate whether *Tut1*-catalyzed oligo(U) tail at the 3′-end of U6 snRNA is required for the survival of neural stem cells. Neural stem cells cannot be directly isolated and maintained as neurospheres from *Tut1*-null brains (Fig. 3G). To circumvent this technical challenge, we used neurosphere culture derived from E12.5 *Tut1*-floxed (*Tut1^f/f*) neocortices and achieved *Tut1* genetic ablation by using a lipid nanoparticle (LNP) system to deliver in vitro-transcribed mRNA encoding Cre recombinase (Cre-mRNA) (Fig. 4A). High-quality Cre-mRNA (Fig. 4B), containing 5′-Cap and N1-methylpseudouridine modifications to ensure high translational efficiency and low immunogenicity, was generated using our recently established pipeline (Zhang et al, 2024). To ensure high transfection efficiency, we first tested 63 LNP formulations containing various lipid components and ratios to deliver GFP-mRNA into neural stem cells. We found that a previously published, SM-102-based formulation (Escalona-Rayo et al, 2023), was the most efficient to transfect neural stem cells with nearly 100% efficiency. Importantly, the expression of GFP was more uniform than traditional lentiviral transduction (Fig. EV2A,B). Then we examined the efficacy of Cre-mRNA using neurospheres derived from Ai14 reporter mice. These cells will show robust RFP expression after Cre-mediated recombination (Madisen et al, 2010). We found that virtually all cells exhibited red fluorescence after LNP/Cre-mRNA transfection (Fig. EV2C).

We used this optimized LNP/RNA delivery system to transfect *Tut1^f/f* neurospheres with Cre-mRNA. Two controls were performed. The first control was neural stem cells derived from Ai14 reporter mice (Ai14), transfected with LNP/Cre-mRNA. The second control was *Tut1^f/f* neural stem cells transfected with LNP containing no Cre-mRNA. The efficiency of *Tut1* deletion was around 95% as determined by qPCR or Nanopore long-read sequencing analyses (Fig. 4C). Ai14 neural stem cells treated with LNP/Cre-mRNA and *Tut1^f/f* neural stem cells treated with LNP continued to proliferate as neurospheres at a comparable rate. In contrast, most *Tut1^f/f* cells treated with LNP/Cre-mRNA died within 5 days (Fig. 4D), similar to neural stem cells derived from E11.5 *Tut1^f/f*;Emx1-Cre neocortices (Fig. 3G). TUT1 protein can add an oligo(U) tail containing 2–4 uridines (Us) to the 3′-end of U6 snRNA, with 2 Us being the most catalytically efficient form (Mroczek et al, 2012; Yamashita and Tomita, 2023). Therefore, we generated in vitro-transcribed recombinant U6 snRNA, containing 0, 2, 4, or 6 Us at its 3′-end, respectively (Fig. 4E). Recombinant U6 snRNA with six Us mimics endogenous mature U6 snRNA containing four Us from the U6 snRNA gene and two Us post-transcriptionally added by TUT1. Neural stem cells were co-transfected with Cre-mRNA and in vitro-transcribed recombinant U6 snRNA, and cell proliferation was monitored by Incucyte live-cell imaging assay. The efficiency of *Tut1* deletion was around 95% as determined by qPCR (Fig. 4F). Recombinant U6 snRNA with 0 or 2 Us had little effect, whereas U6 snRNA with 4Us partially prevented *Tut1* deletion-induced cell death. Importantly, recombinant U6 snRNA with six Us largely prevented cell death (Fig. 4F). Taken together, we conclude that *Tut1*-catalyzed oligo(U) tail at the 3′-end of U6 snRNA is required for the survival of neural stem cells.

Because the 3′-end of U6 snRNA interacts with several essential splicing factors, such as the Lsm2-8 protein complex, we examined whether *Tut1* ablation would compromise the interaction between U6 snRNA and the Lsm2-8 protein complex. Endogenous Lsm4 and Lsm8 proteins were immunoprecipitated, respectively, and associated RNA were extracted to quantitate Lsm4- and

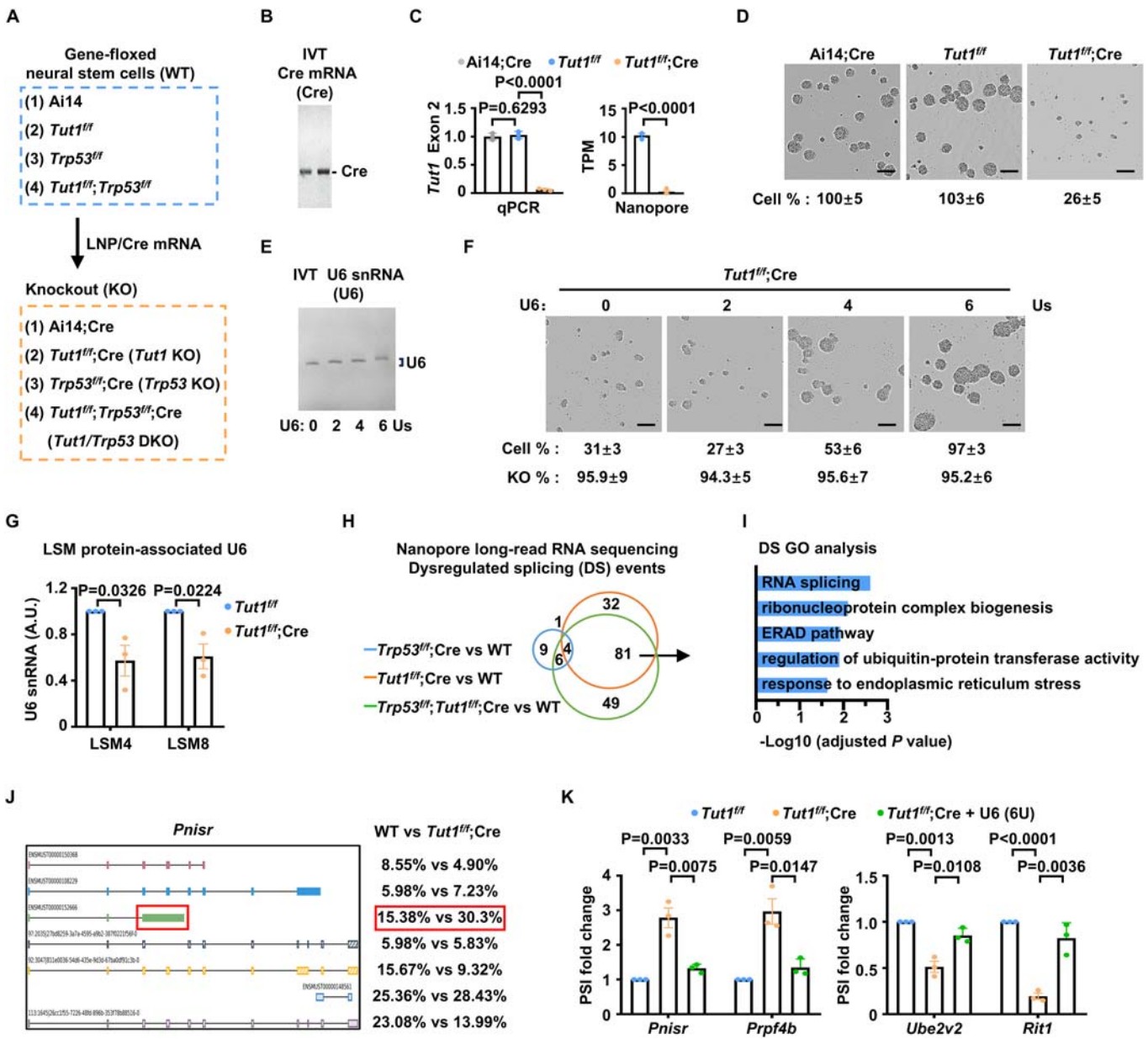

Lsm8-interacting U6 snRNA by qPCR. When *Tut1* was ablated in neural stem cells (2 days after LNP/Cre-mRNA transfection), the amount of U6 snRNA associated with Lsm4 or Lsm8 protein was decreased (Fig. 4G). Interestingly, the total U6 snRNA level was also decreased at this timepoint (Fig. EV2D), indicating that the 3′-end oligo(U) tail promotes the interaction between Lsm2-8 and U6 snRNA. Because the interaction between Lsm2-8 and U6 snRNA is essential for splicing, we performed Nanopore long-read RNA sequencing to investigate *Tut1*-dependent mRNA splicing events in neural stem cells (Fig. 4H and Fig. EV2E). Neural stem cells were derived from *Tut1*-floxed mice (*Tut1^f/f^*), *Trp53*-floxed mice (*Trp53^f/f^*), and double-floxed mice (*Tut1^f/f^;Trp53^f/f^*), respectively. *Tut1^f/f^* neural stem cells treated with LNP alone was used as a control. *Tut1^f/f^*, *Trp53^f/f^*, and *Tut1^f/f^;Trp53^f/f^* neural stem cells were treated, respectively, with LNP/Cre-mRNA for 3 days.

Treatment for 3 days was sufficient to achieve efficient deletion of *Tut1*, and no overt cell death was observed as yet in *Tut1^f/f^* neural stem cells. Because *Trp53* deletion prevents cell death induced by *Tut1* loss (Fig. 3N), simultaneous knockout of *Tut1* with *Trp53* (*Tut1^f/f^;Trp53^f/f^*) provides an additional control to minimize any complicating effects of cell death on mRNA splicing. More than 13,000 genes were detected for each group (Fig. EV2E). Compared to LNP-treated *Tut1^f/f^* cells, 119 dysregulated splicing events in 107 genes were detected in LNP/Cre-mRNA treated *Tut1^f/f^* cells. Among them, 81 dysregulated splicing events in 66 genes were shared by LNP/Cre-mRNA treated *Tut1^f/f^;Trp53^f/f^* cells, but were absent in LNP-treated *Tut1^f/f^* or LNP/Cre-mRNA treated *Trp53^f/f^* cells (Fig. 4H). Thus, these defective splicing events are caused by *Tut1* loss (Fig. 4H). The gene ontology (GO) analysis of dysregulated genes indicated defects in splicing ("RNA splicing" and

**Figure 4.  The *Tut1*-catalyzed oligo(U) tail at the 3′-end of U6 snRNA is required for RNA splicing and the survival of neural stem cells.**

(A) The strategy to generate knockout neural stem cells of the indicated genotypes with LNP/Cre-mRNA transfection. (B) The purity of in vitro-transcribed (IVT) Cre recombinase mRNA (100 ng). (C) The efficiency of *Tut1* genetic ablation in *Tut1^{f/f}* neural stem cells with or without LNP/Cre-mRNA transfection. In knockout cells, the exon 2 is deleted from *Tut1* mRNA. Thus, exon 2 expression can be used to quantify *Tut1* deletion by qPCR or Nanopore long-read RNA sequencing analysis. As a control, neural stem cells derived from Ai14 mice were transfected with LNP/Cre-mRNA. The data represent mean ± SEM from three independent experiments ($n = 3$). Statistical significance was assessed using the two-sided unpaired Student's *t*-test. (D) Incucyte live-cell imaging analysis of the proliferation of neural stem cells with or without LNP/Cre-mRNA treated with LNP/Cre-mRNA (Ai14;Cre) and *Tut1^{f/f}* neural stem cells treated with LNP (*Tut1^{f/f}*) were used as controls ($n = 3$). The cell number of *Tut1^{f/f}* was set to 100%, and data presented as means ± SEM ($n = 3$). Scale bar: 100 µm. (E) The purity of in vitro-transcribed (IVT) U6 snRNA (100 ng), containing 0, 2, 4, or 6 uridines (Us) at its 3′-end, respectively. (F) U6 snRNA containing six uridines (Us) prevents *Tut1* loss-induced neural stem cell death. The relative cell number and *Tut1* knockout efficiency were shown. The cell number of *Tut1^{f/f}* was set to 100%, and data presented as means ± SEM ($n = 3$). The knockout efficiency was quantified by qPCR analysis of the exon 2 of *Tut1*. Scale bar: 100 µm. (G) The interaction between U6 snRNA and Lsm proteins in *Tut1^{f/f}* neural stem cells treated with or without LNP/Cre-mRNA. Lsm4- or Lsm8-associated U6 snRNA was quantified by qPCR. The data represent mean ± SEM from three independent experiments ($n = 3$). AU arbitrary unit. Statistical significance was assessed using the two-sided unpaired Student's *t*-test. (H) Venn diagram presentation of the number of alternatively spliced (AS) genes in *Tut1^{f/f}*;Cre, *Trp53^{f/f}*;Cre, or *Tut1^{f/f}*;*Trp53^{f/f}*;Cre versus wild-type neural stem cells, determined by Nanopore long-read sequencing. (I) GO enrichment analysis of 81 alternatively spliced (AS) genes caused by *Tut1* ablation. Top five enriched biological pathways were shown. Statistical significance was assessed using the two-sided Wilcoxon rank-sum test. (J) An example of a dysregulated intron retention event (red box) in *Pnisr*, determined by Nanopore long-reads sequencing. The percentage on the right denotes the frequency of intron retention in the wild-type (WT) and *Tut1^{f/f}*;Cre neural stem cells, respectively. (K) U6 snRNA containing six uridines (6U) mitigates defective RNA splicing in neural stem cells. Intron retention events in *Pnisr* and *Prpf4b*, and alternative exon usage events in *Ube2v2* and *Rit1* were determined by qPCR analysis. The data represent mean ± SEM from three independent experiments ($n = 3$). Statistical significance was assessed using the two-sided unpaired Student's *t*-test. Source data are available online for this figure.

"ribonucleoprotein complex biogenesis") and the activation of the unfolded protein stress pathway ("endoplasmic reticulum associated protein degradation (ERAD) pathway", "regulation of ubiquitin-protein transferase activity", and "response to endoplasmic reticulum stress") (Fig. 4I). The activation of the unfolded protein stress pathway is likely caused by the translation of incorrectly spliced mRNA. An example of unproductive splicing was shown for *Pnisr* (Fig. 4J). Furthermore, co-transfection of recombinant U6 snRNA with Cre-mRNA mitigated the splicing defects in *Tut1* knockout neural stem cells (Fig. 4K). Taken together, we conclude that *Tut1*-catalyzed oligo(U) tail at the 3′-end of U6 snRNA is important for U6 snRNA stability. Loss of *Tut1* lowers U6 snRNA level, leading to defects in splicing and subsequent neural stem cell death during embryogenesis.

### *Tut1* ablation causes defective RNA splicing in vivo

To further strengthen the role of *Tut1* in splicing in vivo, we carried out Nanopore long-read RNA sequencing, using micro-dissected dorsal and medial palliums from E11.5 wild-type and *Tut1^{f/f}*;Emx1-Cre neocortices. More than 12,000 genes were detected for each group (Fig. EV3). E11.5 timepoint was chosen because the dorsal and medial palliums were still morphologically normal with only minimal activation of DNA damage and downstream death effectors such as c-Casp3 (Fig. 3A–D). In addition, most cells at this stage were neural stem and progenitor cells (Fig. 3B). Thus, most RNA splicing defects detected at the E11.5 timepoint are expected to be the trigger, rather than the outcome, of DNA damage response and cell death programs.

In total, 533 dysregulated splicing events were detected in 376 genes in the micro-dissected dorsal and medial palliums of E11.5 *Tut1^{f/f}*;Emx1-Cre neocortex (FDR <0.05 and |delta PSI|≥10, FLAIR) (Fig. 5A). In addition, we identified 16 differentially expressed genes (Fold change >2, FDR <5%, FLAIR). Nevertheless, only two genes were both differentially expressed and spliced (Fig. 5B). These observations demonstrate that the major defect of *Tut1* loss on gene expression is dysregulated splicing. Out of 533

dysregulated splicing events induced by *Tut1* loss, 35.27% was increased intron retention (IR) events, consistent with a weakened U6 snRNA and spliceosomal function (Fig. 5C). Interestingly, other types of alternative splicing were also dysregulated, including 47.84% alternative exon usage (AE), 10.5% alternative 5′ splice site (A5SS), and 7.88% alternative 3′ splice site events (A3SS) (Fig. 5C), suggesting *Tut1* deletion might induce defects in other regulators of alternative splicing. Consistently, the gene ontology (GO) analysis of dysregulated genes identified "RNA splicing" as top enriched biological processes (Fig. 5D). Intron retention likely introduces premature termination codons. We estimated unproductive isoforms, defined as those with a premature termination codon 55 nucleotides or more upstream of the 3′ splice junction (Tang et al, 2020). Using this criterion, 174 out of 180 intron retention events were estimated to generate unproductive transcripts (Fig. 5E). This observation suggests an underestimation of intron retention events, because unproductive splicing is expected to trigger rapid degradation of transcripts via quality control mechanisms such as nonsense-mediated mRNA decay (Lykke-Andersen and Jensen, 2015). An example of unproductive splicing was shown for *C1qbp* (Fig. 5F), a gene required for efficient DNA repair (Bai et al, 2019). Dysregulated intron retention events in several genes were further validated by qPCR analysis (Fig. 5G). These results demonstrate that *Tut1* is required for mRNA splicing during embryogenesis.

### *Usb1* is dispensable for self-renewal and pluripotency of embryonic stem cells in vitro

In the current model of U6 snRNA 3′-end maturation, TUT1-catalyzed oligo(U) tail serves merely as an intermediate substrate for USB1 (Fig. 1A). This model predicts that knockout of *Usb1* would phenocopy defects caused by *Tut1* deletion. To validate this prediction, we examined the function of *Usb1* in embryonic stem cells. We have shown that *Tut1* is essential for embryonic stem cell maintenance, and *Tut1*-null cells could not be obtained (Fig. 1). Unexpectedly, we found that *Usb1*-null embryonic stem cells could be easily generated using CRISPR/Cas9 genome editing technology

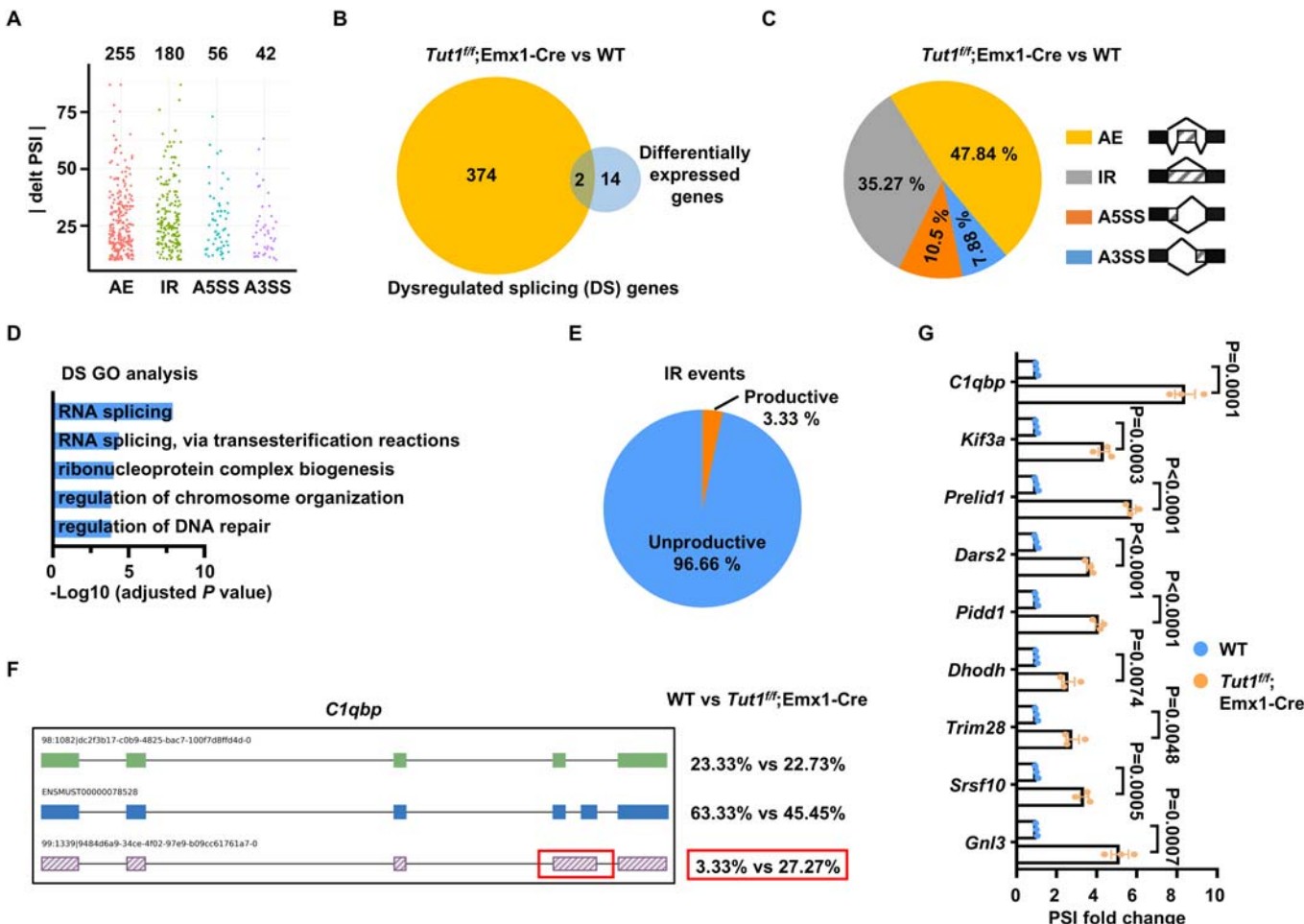

**Figure 5. *Tut1* ablation causes defective RNA splicing in vivo.**

(A) Dot plot showing the distribution of percent spliced in (PSI) values for dysregulated splicing events in E11.5 *Tut1^{f/f}*;Emx1-Cre versus wild-type dorsal and medial palliums of neocortices. Of note, most cells are neural stem and progenitor cells in these subregions, and there is minimal cell death in *Tut1^{f/f}*;Emx1-Cre brains at E11.5. Numbers on the top denote the number of events detected. AE alternative exon usage, IR intron retention, A5SS alternative 5′ splice site, A3SS alternative 3′ splice site. (B) Venn diagram depicting the overlap between differentially expressed genes (*n* = 16) and alternatively spliced genes (*n* = 376) in the micro-dissected dorsal and medial palliums of E11.5 *Tut1^{f/f}*;Emx1-Cre neocortices. (C) Pie chart depicting the distribution of four alternative splicing types dysregulated in the micro-dissected dorsal and medial palliums of E11.5 *Tut1^{f/f}*;Emx1-Cre neocortices. (D) GO enrichment analysis of 376 alternatively spliced (AS) genes. (E) Estimation of unproductive splicing events caused by intron retention. (F) An example of a dysregulated intron retention event (red box) in *C1qbp*. The percentage on the right denotes the frequency of intron retention in the micro-dissected dorsal and medial palliums of E11.5 wild-type (WT) and *Tut1^{f/f}*;Emx1-Cre neocortices, respectively. (G) qPCR analysis to validate increased intron retention events. Results are shown as the fold change of percent spliced in (PSI). The data represent means ± SEM from three independent experiments (*n* = 3). Statistical significance was assessed using the two-sided unpaired Student's *t*-test. Source data are available online for this figure.

(Fig. 6A). RNA blot analysis revealed that U6 snRNA was extended by two Us in *Usb1*-null embryonic stem cells (Fig. 6B), demonstrating that there is no other enzyme compensating for *Usb1* loss in this cell type. Loss of *Usb1* did not alter the colony morphology, the expression levels of pluripotency transcription factors (Oct4, Nanog, and Esrrb) (Fig. 6C,D), or the proliferative rate of embryonic stem cells (Fig. 6E). Furthermore, the transcriptomic profiles were highly similar between wild-type and *Usb1* knockout cells under self-renewal culture condition (Fig. 6F). These results demonstrated that, in contrast to the essentiality of *Tut1*, *Usb1* is not needed for embryonic stem cell self-renewal. To examine whether *Usb1* is required for pluripotency, we treated wild-type and *Usb1* knockout cells, respectively, with retinoic acid

(RA) for 3 days, a commonly used methodology to study embryonic stem cell differentiation. Transcriptome profiling by RNA-seq was carried out for cells without RA treatment (- RA), treated with RA for 1 day (day 1), and for 3 days (day 3) (Fig. 6G). As expected, RA treatment induced gradual changes in the expression of hundreds of genes in wild-type cells at day 1 and day 3 (Fig. 6G). Overall, the trend and the amplitude of these changes were highly similar in *Usb1*-null cells (Fig. 6G). To deduce differentially regulated splicing events in wild-type and *Usb1* knockout cells, we used rMATS (Wang et al, 2024) to analyze RNA-seq datasets. In self-renewing embryonic stem cells (Fig. 6F), this analysis revealed that 1058 splicing events in 949 genes were potentially altered in *Usb1*-null cells (Fig. 6H). Interestingly, only 5

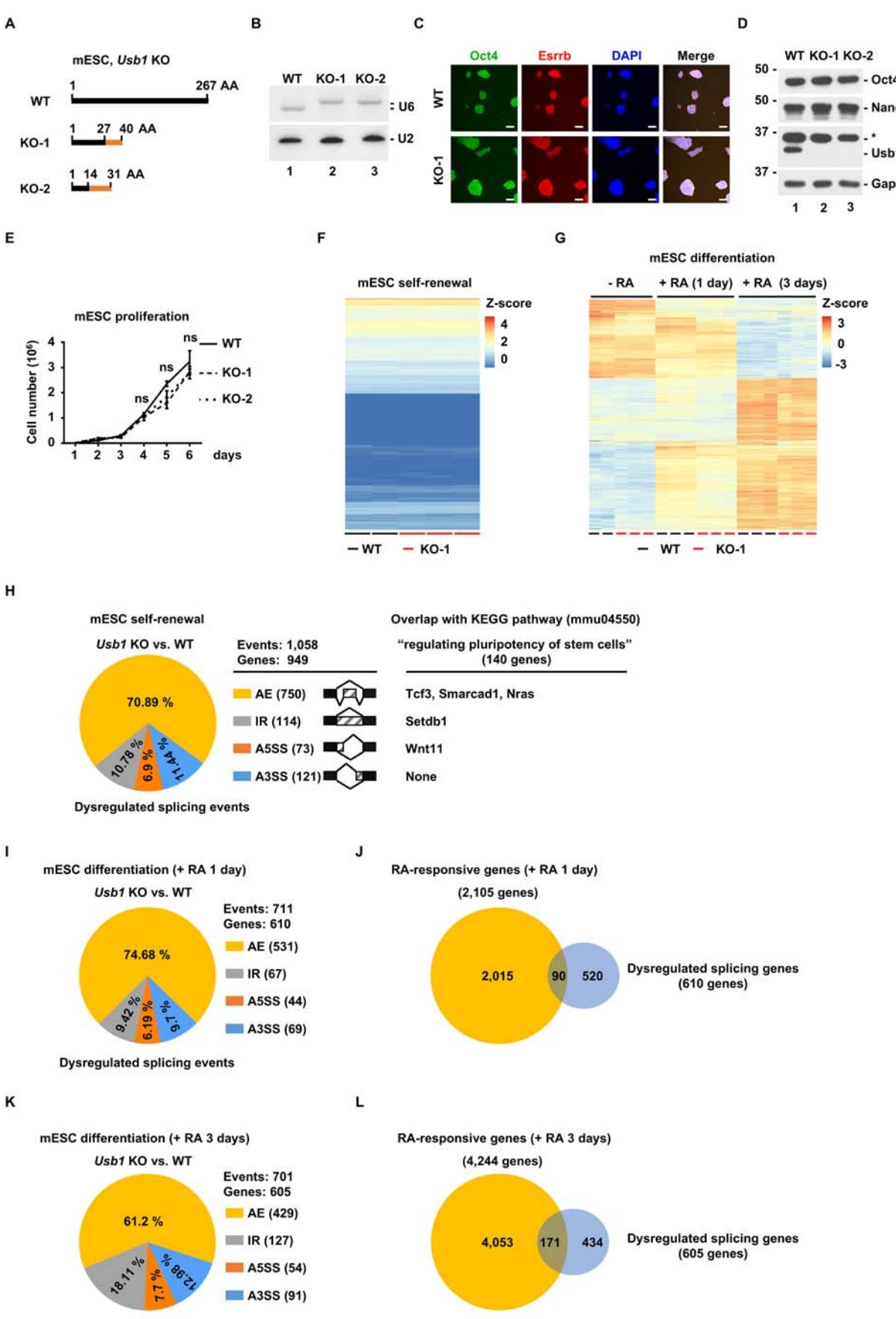

**Figure 6.  *Usb1* is dispensable for the self-renewal and pluripotency of embryonic stem cells in vitro.**

(A) Diagram of two independent *Usb1* knockout mouse embryonic stem cell lines (mESC) generated by CRISPR/Cas9 technology (KO-1 and KO-2). In KO-1, mutant Usb1 mRNA would encode a protein containing N-terminal 27 amino acid residues from the wild-type (WT), followed by 13 non-Usb1 amino acid residues. In KO-2, mutant Usb1 mRNA would encode a protein containing N-terminal 14 amino acid residues from WT, followed by 17 non-Usb1 amino acid residues. (B) RNA blot analysis of U6 snRNA in WT, KO-1, and KO-2 mESC lines ($n = 5$). U2 snRNA was used as a loading control. Of note, the length of U6 snRNA was extended in KO-1 and KO-2 mESC lines. (C) Immunofluorescence analysis of indicated proteins in WT and KO-1 mESC lines ($n = 3$). Oct4 and Esrrb are pluripotency transcription factors for mESC maintenance. Scale bar: 100 μm. (D) Protein blot analysis of indicated proteins in WT, KO-1 and KO-2 mESC lines ($n = 5$). Oct4 and Nanog are pluripotency transcription factors for mESC maintenance. *, non-specific signal detected by anti-Usb1 antibody. (E) Incucyte live-cell imaging assay to measure the proliferative rate of WT, KO-1, and KO-2 mESC lines ($n = 3$). The data represent mean ± SEM. Statistical significance was assessed using the two-sided unpaired Student's *t*-test. (F) Heatmap comparison of the transcriptomic profiles of WT and KO-1 mESC lines under self-renewal culturing conditions by RNA-seq. Two and three biological replicates were performed for WT and KO-1, respectively. (G) Heatmap comparison of the transcriptomic profiles of WT and KO-1 mESC lines upon retinoic acid (RA)-induced differentiation by RNA-seq. Two and three biological replicates were performed for WT and KO-1, respectively, under self-renewal conditions (- RA). Three biological replicates were performed for WT and KO-1, respectively, under RA-induced differentiation (+RA) for 1 day and 3 days. (H) Pie chart depicting the distribution of four alternative splicing types dysregulated in *Usb1*-null mESCs, compared to WT mESCs. In total, 1058 events in 949 genes were detected by rMATS. Numbers in parentheses denote the dysregulated splicing events in *Usb1*-null mESCs. KEGG pathway "regulating pluripotency of stem cells" (mmu04550) consists of 140 genes. AE, alternative exon usage. IR intron retention, A5SS alternative 5' splice site. A3SS, alternative 3' splice site. (I) Pie chart depicting the distribution of four alternative splicing types dysregulated in *Usb1*-null mESCs vs. WT mESCs upon retinoic acid (RA)-induced differentiation for 1 day. In total, 711 events in 610 genes were detected by rMATS. Numbers in parentheses denote the dysregulated splicing events in *Usb1*-null cells. (J) Venn diagram depicting the overlap between differentially expressed genes ($n = 2105$) and alternatively spliced genes ($n = 610$) in *Usb1*-null cells upon retinoic acid (RA)-induced differentiation for 1 day. (K) Pie chart depicting the distribution of four alternative splicing types dysregulated in *Usb1*-null mESCs vs. WT mESCs upon retinoic acid (RA)-induced differentiation for 3 days. In total, 701 events in 605 genes were detected by rMATS. Numbers in parentheses denote the dysregulated splicing events in *Usb1*-null cells. (L) Venn diagram depicting the overlap between differentially expressed genes ($n = 4224$) and alternatively spliced genes ($n = 605$) in *Usb1*-null cells upon retinoic acid (RA)-induced differentiation for 3 days. Source data are available online for this figure.

out of these 949 genes are cataloged under "regulating pluripotency of stem cells" in the KEGG pathway (mmu04550). This observation could explain why there is little functional impact on self-renewal in the absence of *Usb1*. In differentiating embryonic stem cells, 711 splicing events in 610 genes were potentially altered in *Usb1*-null cells at day 1 (Fig. 6I). Out of these 610 genes, 90 were overlapped with RA-responsive genes (RA treatment for 1 day, 2105 genes) (Fig. 6J). At day 3 of RA treatment, 701 splicing events in 605 genes were potentially altered in *Usb1*-null cells (Fig. 6K). Out of these 605 genes, 171 were overlapped with RA-responsive genes (RA treatment for 3 days, 4224 genes) (Fig. 6L). Given that RA-induced differentiation comprises multiple cell types, these observations indicate that these dysregulated splicing events may have some functional outcomes on specific cell types. These results demonstrate that *Usb1* is dispensable for embryonic stem cell self-renewal and pluripotency, that is, the capability of embryonic stem cells to differentiate into any cell types of the three germ layers.

## *Usb1* is dispensable for the development of most cell lineages during embryogenesis

We have demonstrated that *Tut1* is required for neural stem cell survival during embryonic brain development (Figs. 2–4). To investigate the role of *Usb1*, we generated *Usb1*-floxed mice (Fig. 7A), and crossed them with Emx1-Cre mice. In sharp contrast to *Tut1*^f/f;Emx1-Cre embryos (Figs. 2M–O and 3J), we found no discernable defects in developing neocortices of *Usb1*^f/f;Emx1-Cre embryos ($n = 3$) (Fig. 7B,C). In addition, *Usb1*^f/f;Emx1-Cre mice were born at the expected ratio, and had a normal life span (Fig. EV4A). Importantly, RNA blot analysis revealed that U6 snRNA was extended by two Us in *Usb1*^f/f;Emx1-Cre dorsal pallium (Fig. 7D), similar to *Usb1*-null embryonic stem cells (Fig. 6B). We derived *Usb1*-null neural stem cells from *Usb1*^f/f;Emx1-Cre dorsal pallium (Fig. EV4B). These cells proliferated normally as their wild-type counterparts. The expression level of U6 snRNA was comparable in wild-type and mutant cells, but the length of

U6 snRNA was extended as expected in *Usb1*-null cells (Fig. EV4C). Nevertheless, the amount of U6 snRNA associated with Lsm4 or Lsm8 protein was unaltered in *Usb1*-null cells (Fig. EV4D). Furthermore, we carried out 3' RACE experiment to analyze the composition of the U6 snRNA 3'-end in *Usb1*-null cells. Consistent with RNA blot analysis (Figs. 6B, 7D,J and EV4C), we found that most U6 snRNA has 6 Us in *Usb1*-null cells (Fig. EV4E). Thus, in contrast to the essentiality of *Tut1*, *Usb1* is not required for the self-renewal or differentiation of Emx1⁺ neural stem cells in vivo, even though it is fully functional in this cell type.

To probe whether *Usb1* is required in a broader range of cell types, we attempted to generate germline *Usb1* knockout mice using CRISPR/Cas9 genome editing technology (Fig. 7E). We have demonstrated that *Tut1* loss caused early embryonic lethality around E3.5-E5.5, that is, the developmental stage before the emergence of differentiated organs (Fig. 1). In sharp contrast, *Usb1*-null embryos could be obtained at the expected ratio (Fig. 7F), and were morphologically similar with wild-type embryos by E8.5 (Fig. 7G). Nevertheless, E9.5 *Usb1*-null embryos were markedly smaller than wild-type counterparts. This observation indicates that the lethality may be caused by defects in extra-embryonic tissues, and warrants further investigation. To probe potential defects in *Usb1*-null embryos at a higher resolution, we carried out single-cell sequencing analysis of E9.5 wild-type and *Usb1*-null embryos. In total, 19 cell clusters were identified in both wild-type and *Usb1*-null embryos (Figs. 7H and EV4F). On UMAP visualization, all cell clusters identified in *Usb1*-null embryos could be superimposed with their wild-type counterparts, indicating that the overall cellular states of wild-type and mutant cells are highly similar (Fig. 7H). Furthermore, the percentage of each cell type was highly similar between wild-type and *Usb1*-null embryos (Fig. 7I). These results demonstrate that the composition and the functional state of most cell lineages are highly similar in wild-type and *Usb1*-null embryos, and further support the notion that stunted development beyond E8.5 may be caused by defects in extra-embryonic tissues. These results are also consistent with the fact that *Usb1*-null

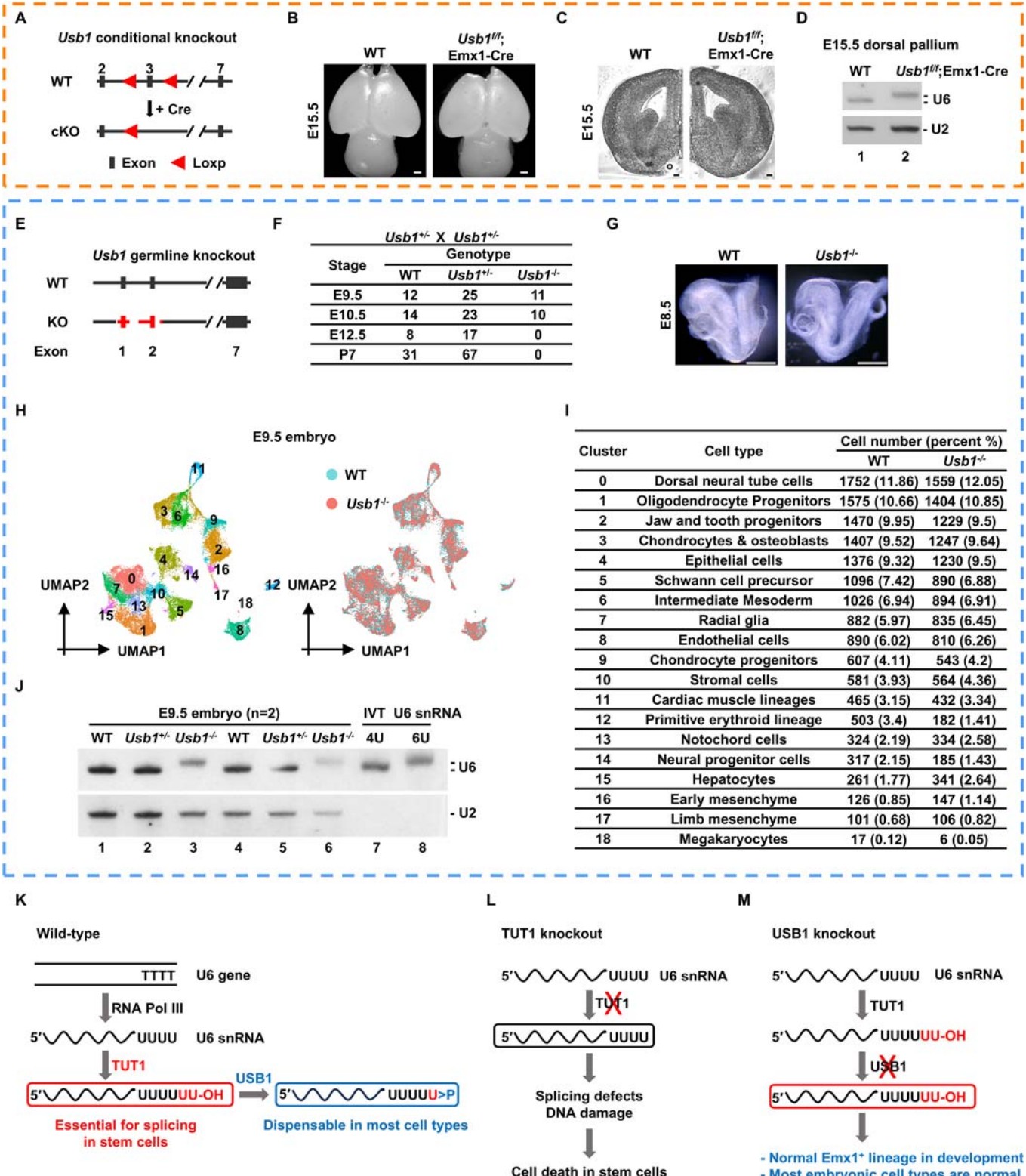

**Figure 7.** *Usb1* is dispensable for the development of most cell lineages during embryogenesis.

(A–D) Analyses of *Usb1*^f/f;Emx1-Cre brains indicate *Usb1* is dispensable for neocortex development. (A) The strategy to generate *Usb1*-floxed mice using CRISPR/Cas9 technology. cKO conditional knockout. (B) Dorsal views of E15.5 wild-type (WT) and *Usb1* knockout (*Usb1*^f/f;Emx1-Cre) brains (n = 3). Scale bar: 500 µm. (C) Coronal sections of E15.5 wild-type (WT) and *Usb1* knockout (*Usb1*^f/f;Emx1-Cre) brains (n = 3). Scale bar: 500 µm. (D) RNA blot analysis of U6 snRNA. Total RNA was extracted from E15.5 dorsal palliums of wild-type (WT) and *Usb1* knockout (*Usb1*^f/f;Emx1-Cre) brains (n = 3). U2 snRNA was used as a loading control. Of note, the length of U6 snRNA was extended in *Usb1*^f/f;Emx1-Cre brains. (E–J) Analyses of *Usb1* germline knockout mouse model (*Usb1*^−/−). (E) The strategy to generate *Usb1* germline knockout mice by CRISPR/Cas9 technology (*Usb1*^−/−). Exons 1 and 2 would be deleted in *Usb1*^−/− mice. WT wild-type. (F) The number of embryos and pups recovered from *Usb1*-heterozygous (*Usb1*^+/−) intercrosses at the indicated embryonic (E) and postnatal (P) ages. Of note, the embryonic lethality of *Usb1*^−/− embryos occurs between E10.5-E12.5, whereas *Tut1*^−/− embryos die around E3.5. (G) Bright-field images of E8.5 wild-type (WT) and *Usb1*^−/− embryos (n = 5). Scale bar: 500 µm. (H, I) Single-cell RNA sequencing analysis of E9.5 wild-type (WT) and *Usb1*^−/− embryos. A total of 19 cell clusters were identified in both WT and *Usb1*^−/− embryos (left panel) (H). Harmony integration demonstrated that all 19 clusters from *Usb1*^−/− embryos were well mixed and grouped with those from WT embryos (right panel) (H). The percentage of corresponding clusters was highly similar between WT and *Usb1*^−/− embryos (I). (J) RNA blot analysis of U6 snRNA. Total RNA was extracted from E9.5 wild-type (WT), *Usb1*^+/− or *Usb1*^−/− embryos (n = 2). U2 snRNA was used as a loading control. In vitro-transcribed (IVT) U6 snRNA containing four or six uridines (Us) were used as a size marker. Of note, the length of U6 snRNA was extended in *Usb1*^−/− embryos. (K–M) The working model based on data from the present study (see Discussion). Source data are available online for this figure.

embryonic stem cells exhibit normal self-renewal and differentiation capacities (Fig. 6). Thus, unlike the essentiality of *Tut1*, *Usb1* is dispensable for epiblast maintenance and differentiation to generate diverse cell lineages during embryogenesis.

Previous studies have shown that USB1 is the only enzyme for shortening the oligo(U) tail at the 3′-end of U6 snRNA in selected human cell lines (Mroczek et al, 2012). To account for the lack of phenotypes, we considered the scenario that in mice, there might be an unidentified protein capable of compensating for *Usb1* loss. To exclude this possibility, we examined the length of U6 snRNA. Total RNA was extracted from the dorsal pallium of *Usb1*^f/f;Emx1-Cre neocortex and *Usb1*-null whole embryos, respectively, for RNA blot analysis. In both cases, the length of U6 snRNA was extended by 2 Us in *Usb1*-null tissues and embryos (Fig. 7D,J). These observations have two implications. First, they demonstrate that *Usb1* is the only gene in the mouse genome, capable of shortening the oligo(U) tail of U6 snRNA. Secondly, they imply that Usb1 is fully active in most embryonic cell types, because all U6 snRNAs are extended by two Us in *Usb1*-null embryos.

We conclude that the function of *Usb1*, and thus 2′,3′-cyclic phosphate modification at the U6 snRNA 3′-end, is functionally dispensable for the development of most intra-embryonic cell lineages.

## Discussion

To the best of our knowledge, the present study is the first to use mouse models to investigate the physiological relevance of *TUT1* and *USB1* and provides several insights into the functions of previously underappreciated post-transcriptional modifications at the U6 snRNA 3′-end. Formulated on results from biochemical and cell-based studies in vitro, the current view of U6 snRNA 3′-end maturation depicts TUT1-catalyzed oligo(U) tail as the intermediate substrate for USB1 to generate 2′,3′-cyclic phosphate group, with the latter modification being essential for RNA splicing and cellular functions. However, our results, based on eight genetically engineered mouse models, demonstrate that TUT1-catalyzed oligo(U) tail is essential for RNA splicing to maintain embryonic and neural stem cells, whereas USB1 is dispensable in most embryonic cell types during embryogenesis (Fig. 7K). Without the

TUT1-catalyzed oligo(U) tail, splicing defects lead to massive DNA damage and cell death in stem cells (Fig. 7L). Importantly, we have shown that both cell death and splicing defects can be mitigated in *Tut1*-null neural stem cells by recombinant U6 snRNA with an oligo(U) tail containing six uridines. The essentiality of TUT1 may be explained by the fact that the LSM2-8 protein complex, the ubiquitous essential splicing factor, binds the oligo(U) tail at the 3′-end of U6 snRNA. Indeed, we find that the interaction between U6 snRNA and Lsm proteins is weakened in *Tut1*-null cells. This observation suggests that TUT1, but not USB1, is a ubiquitous, essential U6 snRNA 3′-end maturation or repair factor required for cell proliferation. This notion is supported by the large-scale studies mapping genetic vulnerabilities in human cancer cell lines (https://depmap.org/portal) (Arafeh et al, 2025). All 1183 human cancer cell lines require *TUT1* for survival, whereas the viability of only 101 cell lines partially depends on *USB1* (DepMap Public 25Q2).

In sharp contrast, we find that *Usb1* is not required in several biological contexts, in which *Tut1* is essential. Remarkably, the 3′-end of all U6 snRNAs in the *Usb1*-null embryos is extended by 2 uridines, demonstrating that there is no other enzyme encoded by the mouse genome capable of compensating for *Usb1* loss. This observation also demonstrates that *Usb1* is active in most cell types, even though it is not functionally required. This notion is further supported by the fact that mice lacking *Usb1* in Emx1^+ cell lineages are healthy with a normal life span, and by our single-cell sequencing analyses showing that the cellular compositions and transcriptional states are indistinguishable between wild-type and *Usb1*-null embryos. Of note, *Usb1*-null embryos can develop till mid-embryogenesis, but then degenerate. Our preliminary analyses indicate that extra-embryonic defects may underlie this lethality. Nevertheless, we cannot rule out the possibility that *Usb1* may be required in certain intra-embryonic lineages. Further studies will be needed to address these issues.

Our results provide experimental support for previous human genetics studies linking variants in *TUT1* and *USB1* to distinct human disorders. In humans, biallelic variants in *TUT1* are linked to cortical atrophy, microcephaly, and cerebellar atrophy. We show that conditional knockout of *Tut1* in the central nervous system results in severe cerebral atrophy, broadly resembling features in human patients. Cerebral atrophy can be explained by the fact that *Tut1* loss causes global splicing defects, and subsequently DNA

damage and cell death in neural stem and progenitor cells. In addition, we find that *Tut1* is required to maintain epiblast and embryonic stem cells. Collectively, these results suggest the hypomorphic nature of human *TUT1* variants in patients and may explain why biallelic damaging variants in *TUT1* are extremely rare in humans. On the other hand, individuals bearing biallelic inactivating variants in *USB1* are viable, albeit with defects in the immune and skeletal systems. Notably, no obvious defects are present in the brain or other organs in these individuals. These observations are consistent with our findings that *Usb1* is not functionally required in most cell types. Further studies are needed to determine the effect of *Usb1* in the immune and skeletal systems in engineered mouse models.

There are some debates on the biochemical activities of TUT1 and USB1. TUT1 was originally identified as an oligo(U) polymerase for U6 snRNA (hence its official full name: Terminal Uridylyl Transferase 1, U6 snRNA-specific) (Trippe et al, 2006; Trippe et al, 2003). Subsequent studies argued that TUT1 functions as a poly(A) polymerase (Mellman et al, 2008). However, recent biochemical and structural studies confirmed TUT1 as an oligo(U) polymerase for U6 snRNA, and found no evidence for poly(A) polymerase activity (Yamashita et al, 2017; Yamashita and Tomita, 2023; Yamashita and Tomita, 2025). Our results support that TUT1 indeed functions as an oligo(U) polymerase for U6 snRNA. We find global splicing defects in *Tut1*-null cells, consistent with a weakened U6 snRNA function. Importantly, we show that *Tut1* knockout-induced cell death and splicing defects can be mitigated by U6 snRNA containing an oligo(U) tail. On the other hand, biochemical, structural, and cell-based studies have established that USB1 is an exonuclease to trim the oligo(U) tail at the U6 snRNA 3′-end (Didychuk et al, 2017; Hilcenko et al, 2013; Mroczek et al, 2012; Shchepachev et al, 2012; Yamashita et al, 2017). Our results provide in vivo evidence to support this activity of USB1. We show that in a variety of *Usb1*-null cell types, the 3′-end of U6 snRNA is extended by 2 extra uridines. Of note, one recent study, using in vitro cultured human iPSCs, suggested that USB1 can remove the oligo(A) tail from mature microRNAs (Jeong et al, 2023). However, it is unclear how USB1, a nuclear protein (Mroczek et al, 2012), could act upon mature miRNAs in the cytoplasm. In addition, it is well established that PARN is a nuclease removing oligo(A) tails from many microRNAs (Shukla et al, 2019). It is unclear why USB1 ablation affects the same miRNAs that are also targets of PARN. Regardless of whether USB1 can trim oligo(A) tail of miRNAs or not, we show that knockout of *Usb1* does not affect embryonic stem cell self-renewal or differentiation in vitro and does not affect epiblast proliferation and its differentiation into most intra-embryonic cell types in vivo. These results indicate that USB1-catalyzed 2′,3′-cyclic phosphate modification at the U6 snRNA 3′-end is not functionally required in most cell types, at least under homeostatic conditions. However, it is tempting to speculate that USB1 may function in certain stress response pathways, considering that it is an evolutionarily conserved enzyme and its ubiquitous presence in most cell types. The eight mouse models and two stem cell models generated in this study provide a starting point for future efforts to resolve these debates, and to investigate the temporal and spatial regulation and functions of different U6 snRNA 3′-end post-transcriptional modifications in development and disease.

# Methods

## Reagents and tools table

| Reagent/resource | Reference or source | Identifier or Catalog Number |
|---|---|---|
| **Experimental models** | | |
| *Tut1*^fl/fl | This paper | |
| *Usb1*^fl/fl | This paper | |
| *Tut1* conventional knockout mice | This paper | |
| *Usb1* conventional knockout mice | This paper | |
| nestin-Cre (B6.Cg-Tg (Nes-cre)1Kln/J) | Jackson Lab, #003771 | |
| Emx1-Cre (B6.129S2-Emx1^tm1(cre)Krj/J) | Jackson Lab, #005628 | |
| Camk2a-Cre (B6.Cg-Tg(Camk2a-cre) T29-1Stl/J) | Jackson Lab, #005359 | |
| **Antibodies** | | |
| anti-Sox2 | R&D, AF2018 | |
| anti-Gapdh | KangChen, KC-5G5 | |
| anti-Dcx | Millipore, SAB2501666 | |
| anti-Tbr2 | Abcam, ab23345 | |
| anti-Oct4 | Santa Cruz, sc-9081 | |
| anti-Pax6 | Proteintech, 12323-1-AP | |
| anti-Ki67 | Invitrogen, MA5-14520 | |
| anti-Nanog | Bethyl, A300-397A | |
| anti-Cleaved-caspase 3 | CST, 9664 s | |
| anti-LSM4 | Zenbio, 671053 | |
| anti-LSM8 | Santa Cruz, sc-390542 | |
| anti-Esrrb | R&D, PP-H6705-00 | |
| anti-γH2AX | Abcam, ab2893 | |
| anti-BrdU | Invitrogen, B35128 | |
| anti-mouse IgG-HRP | CST, 7076S | |
| anti-rabbit IgG-HRP | CST, 7074S | |
| Alexa Fluor 488 AffiniPure Donkey Anti-Rabbit IgG | Jackson Immunoresearch, 711-545-152 | |
| CyTM3 AfriniPure Donkey Anti-Mouse IgG | Jackson Immunoresearch, 715-165-150 | |
| Alexa Fluor 488 AffiniPure Donkey Anti-Mouse IgG | Jackson Immunoresearch, 715-545-150 | |
| CyTM3 AffiniPure Donkey Anti-Goat IgG | Jackson Immunoresearch, 705-165-147 | |
| **Oligonucleotides and other sequence-based reagents** | 5′-3′ | |
| U6 probe | GGAACGCTTCACGAATTTGCGTGT CATCCTTGCGCAGGGGCCA/iBiodT/gc/ iBiodT/A | |
| Sn7sk probe | CGGGGAAGGTCGTCCTC/ iBiodT / iBiodT/ C | |
| U2 probe | GAGCAAGCTCCTATTCCAACTCCT ACTTCCAAAAA iBiodT / iBiodT/ | |

| Reagent/resource | Reference or source | Identifier or Catalog Number |
|---|---|---|
| **Genotyping PCR primer sequences** | | |
| *Tut1*-germline KO- F | CCAGCCATCTTTCTAACTCCTTT | |
| *Tut1*-germline KO- R1 | CTTTCAAAGGACCTAGGTTTGGT | |
| *Tut1*-germline KO- R2 | ATGCGGACACAAATATAAGAAGC | |
| *Usb1*-germline KO- F1 | CCTGACAGTGTGCTAAGTATGTT | |
| *Usb1*-germline KO- F2 | GAGCCTGTGGGTACGGA | |
| *Usb1*-germline KO- R | GCAGTTAGGCAGTCCATTGT | |
| *Tut1*-floxed- F2 | ACCACTGCCTGGCTATGGAATACC | |
| *Tut1*-floxed- R2 | CCCTTTTCTACACAGTGCATTCTC | |
| *Tut1*-cKO- F1 | ACCACTGCCTGGCTATGGAATACC | |
| *Tut1*-cKO- R2 | GCCTTATAGATATACTGGGTCACCTG | |
| *Trp53*-floxed- F2 | AAGGGGTATGAGGGACAAGG | |
| *Trp53*-floxed- R2 | GAAGACAGAAAAGGGGAGGG | |
| *Trp53*-cKO- F1 | CACAAAAACAGGTTAAACCCA | |
| *Trp53*-cKO- R2 | GAAGACAGAAAAGGGGAGGG | |
| *Usb1*-floxed- F | CTTCTCCAGTTGCTCTCAGGATTC | |
| *Usb1*-floxed- R | CCTATTGATGGAGTGAAGAACCAC | |
| Cre- F | ACCCTGTTACGTATAGCCGA | |
| Cre- R | CTCCGGTATTGAAACTCCAG | |
| **Guide RNAs oligonucleotide sequences** | | |
| gRNA-1 for germline knockout *Tut1* mouse | TTGTGGGTACCCGGTGCCTT | |
| gRNA-2 for germline knockout *Tut1* mouse | TCCTACGCTTTTAACCCCAA | |
| gRNA-1 for germline knockout *Usb1* mouse | GGGCGGAGCAAACCGGGTAC | |
| gRNA-2 for germline knockout *Usb1* mouse | GGCCCAGTTGCCCCGCTCAT | |
| gRNA-1 for *Tut1* knockout mESCs | GGTGCTGGGCTCCAATCTTG | |
| gRNA-2 for *Tut1* knockout mESCs | TCCAATCTTGAGGCATCCTT | |
| gRNA-3 for *Tut1* knockout mESCs | GCAACTCCGAGCTACCAGGA | |
| gRNA-4 for *Tut1* knockout mESCs | GAGATGTATCCCTCAGTAAC | |
| gRNA-1 for *Usb1* knockout mESCs | GAAGCGGAGGCTGTCGCCGC | |
| gRNA-2 for *Usb1* knockout mESCs | GGGCGGAGCAAACCGGGTAC | |
| **Target shRNA sequences** | | |
| shTut-1 | GTGTGTTTGTCAGTGGCTTCC | |
| shTut1-2 | TTAGAGCTGGTGGGATCTATT | |
| **Primers for qPCR analysis** | | |
| Abcd4-intron-F | CCCCAGGGGAGCCTTCTAGTT | |
| Abcd4-intron-R | AGTGCCACAGACGTAGCTTT | |
| Abcd4-F | AGGAAGCTGGGAGCTGACTA | |
| Abcd4-R | ATCGGGCTTCCTTGTCTGTG | |
| Dhodh-intron-F | GATATGTGTGGAGCCCCGGT | |
| Dhodh-intron-R | GTAATGACCCAGGTCTCCGT | |
| Dhodh-F | ACACAGTGACAGACGCCATT | |
| Dhodh-R | CAGCTCTCCCTCTCAGGACT | |
| Pidd1-intron-F | TGTCAGCTTTGTTTCACCAACC | |
| Pidd1-intron-R | AGGGACATCTGGAAGCAGTGA | |
| Pidd1-F | GCGTTGCTGCTTTCTCACAA | |

| Reagent/resource | Reference or source | Identifier or Catalog Number |
|---|---|---|
| Pidd1-R | GAGATCGAGACGCTGAAGGG | |
| Prelid1-intron-F | GGTGGAGGAACGATGTGTTT | |
| Prelid1-intron-R | GAGGGAGAAGGATGGGAGAGTT | |
| Prelid1-F | GCAGCTACAGAGAAGGCCAA | |
| Prelid1-R | AGAGGGAGAGGAAGCGAGTT | |
| Gnl3-intron-F | CTATTGCCATCCCCCTGCAT | |
| Gnl3-intron-R | CTAGCATTCTGTCGTCATTCACT | |
| Gnl3-F | TCAGATGTGGCCCCTGTAGA | |
| Gnl3-R | ACGTTAGTGTGTGGCGGTTA | |
| Dars2-intron-F | AGTCAGCCCATCACCCATTT | |
| Dars2-intron-R | CTCCTGAGGGAATCCAGAAGTT | |
| Dars2-F | CAATGCTCCAGACTCCGTGT | |
| Dars2-R | CCTCTTCTGAGTCTGCTGGC | |
| Srsf10-intron-F | TGTGCATCTTGGATGCTTCGT | |
| Srsf10-intron-R | AGCATACCAGAAAAAGCAAACAGT | |
| Srsf10-F | GTAATCGGCAGCTCTTGGGT | |
| Srsf10-R | CACACAACTTGGCACAGCAA | |
| C1qbp-intron-F | ACAACAGCATCCCTCCAACAT | |
| C1qbp-intron-R | GAAGGTCAGAAGACTCCACCC | |
| C1qbp-F | TTTGCGGATGAGTTGGTGGA | |
| C1qbp-R | CTGGCCAAAGCTTGCCATTT | |
| Kif3a-intron-F | GGCTGCAGACAGCACATAGA | |
| Kif3a-intron-R | GGTGCTTTTCCCCTGAAGGT | |
| Kif3a-F | GGTGGGGAGGTCTGTTGAAG | |
| Kif3a-R | AGTCCATGAGAAGCAGCCAC | |
| Smndc1-intron-F | GCTAATTCTTGGGTGGTTGCC | |
| Smndc1-intron-R | ACCACTCTTCTCCCCAAACT | |
| Smndc1-F | CCACAGCTTCCTCAGTGTGT | |
| Smndc1-R | ATGGGCCTGACGTTCAGAAG | |
| Trim28-intron-F | CCTGGTACGAACTCCACAGG | |
| Trim28-intron-R | GCCCAATTTCCAAGGCACAA | |
| Trim28-F | CTCTTGTGGTCAGCCCAGTC | |
| Trim28-R | AGCAAGAACAGGAGTCAGGC | |
| Pnisr-intron-F | ACAGTCTGCTAGTGTTGTGGC | |
| Pnisr-intron-R | GACGCCAAGTGAGGTTTCCA | |
| Pnisr-F | CCAGCAAGATCCAAGCCAGA | |
| Pnisr-R | ACGATGCTTTGCTGTCCTGA | |
| Prpf4b-intron-F | TGGCAGAATTCCCTGCATGT | |
| Prpf4b-intron-R | ACAGTACACTGAAGGCGCTC | |
| Prpf4b-F | AAAGATGCCAGCCCCATCAA | |
| Prpf4b-R | TCCTATGACCACCCCGTGAT | |
| Ube2v2-AE-F | ACCCAGAAGCTCCTCCATCA | |
| Ube2v2-AE-R | CCACTGGAATTATTGATCCCATTCA | |
| Ube2v2-F | GATGCACGGAGCATACCAGT | |
| Ube2v2-R | CTGTCCTTCTGGAGGCTGTG | |
| Rit1-AE-F | TGTCCTGAGTTGCATCTCTT | |
| Rit1-AE-F | CGGTGGCTGATGAACTGCAT | |
| Rit1-F | CAGCCATGCGGGATCAGTAT | |
| Rit1-R | CTGCCTCAGCTGCTTTAGGT | |
| **Primers for U6 cDNA cloning** | | |
| U6-F | TAATACGACTCACTATAGTGC TCGCTTCGGCAG | |
| U6-0U-R | CGTCTCCTATGGAACGCTTCACG | |

| Reagent/resource | Reference or source | Identifier or Catalog Number |
|---|---|---|
| U6-2U-R | CGTCTCCAATATGGAACGCTTCACG | |
| U6-4U-R | CGTCTCCAAAATATGGAACGCTTCACG | |
| U6-6U-R | CGTCTCCAAAAAATATGGAACGCTTCACG | |
| Sequences of peptides for the generation of the TUT1 antibody | AGEGEQVEVDGWDCSFP | |
| Sequences of peptides for the generation of USB1 antibody | CDSTKHGGRVRTFPHERGN | |
| **Chemicals, enzymes, and other reagents** | | |
| DMEM/F12 | | Gibico, 11320-033 |
| N2 | | 17502048 |
| B27 | | 17504044 |
| EGF | | Sino Biological, 10605-HNAE |
| bFGF | | Sino Biological, 10014-HNAE |
| 5-BrdU | | Sigma, B5002 |
| Accutase | | Gibico, A1110501 |
| qPCRmix (ssofast evagreen@ supermix) | | Bio-Rad, 1725201 |
| Fluoromount-G® | | SouthernBiotech, 0100-01 |
| Trizol | | Invitrogen, 15596018 |
| Matrigel | | Corning, 356237 |
| **Software** | | |
| CellRanger (v6.1.2) | PMID: 28091601 | |
| Seurat (v4.3) | PMID: 25867923 | |
| Harmony (v0.1.0) | PMID: 31740819 | |
| clusterProfiler (v4.10) | PMID: 22455463 | |
| Minimap2 (v2.24) | PMID: 29750242 | |
| FastQC (v0.11.9) | Andrews S (2010) FastQC: a quality control tool for high-throughput sequence data | |
| Trimmomatic (v0.39) | PMID: 24695404 | |
| Bowtie2 (v2.4.5) | PMID: 22388286 | |
| SAMtools (v1.15) | PMID: 19505943 | |
| BEDTools (v2.30.0) | PMID: 20110278 | |
| deepTools (v3.5.1) | PMID: 24799436 | |
| R (v4.2.2) | Ihaka R, Gentleman R (1996) R: a language for data analysis and graphics. J Comput Graph Stat 5:299–314 | |
| Bioconductor (v3.16) | PMID: 15461798 | |
| rMATS v4.3.0 | PMID: 38396040 | |
| FLAIR v2.0.0 | PMID: 32188845 | |
| **Other** | | |
| TUNL | | YEASON, China |

## Generation of germline and conditional knockout mouse models

Germline deletion of *Tut1* (ENSMUSG00000071645) or *Usb1* (ENSMUSG00000031792), respectively, was carried out using CRISPR/Cas9 genomic editing technology to generate conventional knockout mice models. For *Tut1*, two guide RNAs parsing the region of exon 3 to exon 5 of *Tut1*, and for *Usb1*, two guide RNAs parsing the region of exon 2 of *Usb1* were used. To generate germline knockout mouse embryos, heterozygous mice were intercrossed. To generate *Tut1*- and *Usb1*- floxed mice, two loxp sequences were inserted into the introns flanking exon 2 of *Tut1* (*Tut1^(f/f)*), or exon 3 of *Usb1* (*Usb1^(f/f)*), respectively. To obtain conditional knockout mice, floxed mice were crossed with nestin-Cre (B6. Cg-Tg (nes-cre) 1Kln/J; Jackson Lab, #003771), Emx1-Cre (B6.129S2-Emx1^(tm1(cre)Krj)/J; Jackson Lab, #005628), or Camk2a-Cre (B6.Cg-Tg(Camk2a-cre)T29-1Stl/J, Jackson Lab, #005359) mice individually. To generate double knockout mice, *Tut1^(f/f)* mice were crossed with *Trp53^(f/f)* (B6.129P2-*Trp53*^(tm1Brn)/J, Jackson Lab, #008462) mice to obtain *Tut1^(f/f)*;*Trp53^(f/f)* mice. *Tut1^(f/f)*;*Trp53^(f/f)* mice were then crossed with nestin-Cre or Emx1-Cre mice, respectively. Noon on the day of the vaginal plug was defined as E0.5. Embryos and offsprings were genotyped by PCR of genomic DNA. Mice were housed under a 12-h light–dark cycle and had ad libitum access to food and water in a controlled animal facility. Animal studies were approved by the Institutional Animal Care and Use Committee at West China Second University of Sichuan University (2022-054).

## Stem cell models and LNP transfection

Primary neural stem cells were derived from E12.5 neocortices of gene-floxed embryos (Ai14, *Tut1^(f/f)*, *Trp53^(f/f)*, and *Tut1^(f/f)*;*Trp53^(f/f)*), and of *Tut1^(f/f)*;*Trp53^(f/f)*;Emx1-Cre and *Usb1^(f/f)*;Emx1-Cre embryos, using a previously described protocol (Ahlenius and Kokaia, 2010). For *Tut1^(f/f)*;Emx1-Cre, E11.5 neocortices were used. Briefly, the dorsal pallium was dissociated by accutase into single cell, and cultured in N2B27 medium containing bFGF and EGF. Cell proliferation was monitored by the Incucyte S3 live-cell analysis system (Sartorius). In vitro-transcribed Cre-mRNA was generated as previously described (Zhang et al, 2024). U6 snRNA cDNA was cloned using total RNA derived from mouse primary neural stem cells, and its sequence is identical to NR_003027.2 (without the terminal 5 Ts). To generate recombinant U6 snRNA with 0, 2, 4, or 6 Us at the 3′-end, U6 snRNA cDNA with corresponding tails was sub-cloned into the pUC57 vector using primers listed in Reagents and Tools Table (forward primer: U6-F; reverse primers: U6-0U-R, U6-2U-R, U6-4U-R, and U6-6U-R), linearized by Esp3I (Thermo Fisher), and then used as the template for in vitro transcription by T7 polymerase (Novoprotein, China) following the manufacturer's instruction. Of note, Esp3I is a Type IIS enzyme generating a precise 3′-end without any vector-derived nucleotides. The recombinant U6 snRNA was purified by the RNeasy Mini kit (QIAGEN). 100 ng recombinant RNA was resolved by 8% denaturing gel, and stained with ethidium bromide to inspect the purity. The length and integrity of U6 snRNA and Cre-mRNA was routinely monitored by Agilent 2100 using RNA 6000 nano kit. Cre-mRNA and U6 snRNA were individually packed with LNP. To transfect neural stem cells, LNP/RNA complexes containing 1 μg of Cre-mRNA and/or 3 μg of U6 snRNA were incubated with $10^5$ cells. To probe the interaction between U6 snRNA and Lsm proteins, endogenous Lsm4 and Lsm8 proteins were immunoprecipitated from cultured neural stem cells, respectively, followed by TRIzol extraction of associated RNA, and the detection of U6 snRNA by qPCR. The amount of immunoprecipitated Lsm proteins from *Tut1*

knockout cells were normalized to those from wild-type cells. The amount of immunoprecipitated U6 snRNA/Lsm protein from wild-type cells were arbitrarily set to value 1.

R1/E mouse embryonic stem cells (mESCs) (ATCC, SCRC-1036) were maintained as previously described (Fang et al, 2022). To knock down *Tut1* expression in mESCs, two short-hairpin RNA (shRNA) constructs were used. To knockout *Tut1* in mESCs, four independent guide RNA sequences were used. However, no *Tut1* knockout mESC clones could be obtained (*n* = 157). To generate *Usb1* knockout mESCs, two guide RNA sequences were used. Knockout cell lines were identified by protein blotting and DNA sequencing. Alkaline phosphatase staining was performed following the manufacturer's instructions (Sigma). To initiate mESC differentiation, retinoic acids were added and cells were collected at the indicated time points.

## Northern blot, 3′ RACE analysis and RNA immunoprecipitation (RIP)

Northern blot analysis was carried out using Chemiluminescent Nucleic Acid Detection Module (Thermo Fisher), following the manufacturer's instructions. Briefly, 250 ng of total RNA were separated on 8% denaturing gel, and then transferred to a Nytran N$^+$ membrane (GE). The membrane was crosslinked by UV, blocked with NorthernMax™ (Invitrogen™), and then incubated with biotin-labeled probes. 3′ RACE experiments were performed as previously described (Shchepachev et al, 2015), using total RNA derived from wild-type and *Usb1* knockout neural stem cells, respectively (*n* = 3). Sixteen clones were sequenced for each genotype (96 clones in total). RIP procedure was carried out as previously described (Gagliardi and Matarazzo, 2016). Briefly, cell lysates were incubated with protein A/G beads and anti-Lsm4 or anti-Lsm8 antibody for 4 h at 4 °C, and the beads were washed four times. Associated RNA was extracted by TRIzol, reverse transcribed by M-MLV reverse transcriptase, and then amplified using U6 snRNA-specific primers.

## Tissue analyses

For embryonic brain tissues, samples were fixed with 4% paraformaldehyde overnight at 4 °C, and cryopreserved in 25% sucrose for 1 day before embedding in Tissue-Tek OCT Compound (Sakura Finetek, USA). Samples were sectioned on a cryostat at 8–10 μm, and then stained with the indicated antibodies. Terminal deoxynucleotidyl transferase-mediated deoxyuridine triphosphate (dUTP) nick end labeling (TUNEL) assay was performed following the manufacturer's instructions (YEASON, China). Images were obtained using a fluorescent confocal microscope (FV3000; Olympus, Japan).

## Behavioral testing

Behavioral analyses were carried out as previously described (Rein et al, 2020; Vorhees and Williams, 2006; Walf and Frye, 2007). Briefly, 8- to 12-week-old wild-type and *Tut1^{f/+}*;nestin-Cre mice with an equal sex ratio were placed in the test room one hour prior to each test. Wild-type and *Tut1^{f/+}*;nestin-Cre mice were evaluated on the same day, and equipment was cleaned with 70% ethanol after each animal. Tests were performed starting with the least aversive to the most aversive task, scored automatically and analyzed by SMART 3.0 (Panlab). For the open field test, the time spent in three areas was determined, defined as the "Periphery (5 cm from the walls)", "center (a 14 cm × 14 cm square in the center)" and "non-periphery" (areas excluding periphery and center). For the three-chamber sociability and social novelty test, active interactions between the mice were scored with 10 min of total test time. For the elevated plus maze test, mice were placed in the center of a black elevated plus maze (each arm is 30 cm long and 6 cm wide with two arms enclosed by 15 cm-high walls), and allowed to explore for 5 min in a dimly lit room. Then the time spans spent in the open, closed arms and in the middle were determined. For the Morris water maze, mice were trained for 5 days to locate the quadrant containing an escape platform. On day 7, the escape platform was removed, and each mouse was allowed to swim for 1 min to determine whether the animal remembered the location of the platform. The time each animal spends in and the number of times the animal crosses the quadrant previously containing an escape platform during the 5-day training period were documented. These two parameters were regarded as indicators of how well the animal has learned the spatial location of the platform.

## Single-cell RNA-seq, Nanopore long-read RNA-seq, bulk RNA-seq, and bioinformatic analyses

Micro-dissected dorsal and medial palliums from wild-type and *Tut1^{f/f}*;Emx1-Cre E11.5 embryos were used for long-read RNA sequencing to analyze alternative splicing (AS) events in poly(A)-enriched RNA. Briefly, minimap2 (v2.24) (Li, 2018) was used to align reads with the ENSEMBL (v102) mouse reference transcriptome. Alternative acceptor, alternative donor, alternative exon usage, and intron retention events were identified by FLAIR (v2.0.0) diffSplice and genes with multiple isoforms identified were considered alternatively spliced (Tang et al, 2020). Differentially spliced genes were further subjected to gene ontology (GO) enrichment analysis using clusterProfiler (v4.10) in the R package. Enriched GO terms were further grouped based on similarity using Simplify in clusterProfiler (v4.10) (Wu et al, 2021). For gene set enrichment analysis, GSEA (v4.1.0) and Hallmark gene set from Molecular Signatures Database were used (Liberzon et al, 2015).

Single-cell RNA-seq was used to compare cell types derived from E9.5 wild-type (*Usb1^{+/+}*) and *Usb1^{−/−}* embryos. Embryos were dissociated into single-cell suspension, loaded into Chromium microfluidic chips with 3′ reagent kit (v3.1 chemistry), and barcoded with a 10X Chromium Controller (10X Genomics). Sequencing was performed using the Illumina NovaSeq platform following the manufacturer's instructions (Annoroad Gene Technology (Beijing) Co., Ltd.). The bioinformatics pipeline was previously described (Li et al, 2023). After quality control, 14,776 and 13,034 cells from wild-type and *Usb1*-null embryos, respectively, were used for bioinformatic analyses. Briefly, the sequencing data were aligned to the mouse transcriptome to generate the expression matrix using the CellRanger (v6.1.2). Cells with detected gene numbers between 200 and 6000, and a percentage of mitochondrial genes below 30%, are used for unbiased cell clustering by Seurat (v4.3). Batch effects were corrected using the Harmony package v0.1.0 in UMAP visualization. The putative identities of each cluster were assigned using pre-defined markers (Cao et al, 2019) (Fig. EV4F).

To compare wild-type and *Usb1* knockout murine embryonic stem cells under self-renewal and differentiation conditions, two or three independent biological replicas were subjected to

transcriptomic profiling by bulk RNA-seq. The sequencing and bioinformatics pipeline were previously described (Fang et al, 2022). Briefly, sequencing libraries were generated using NEBNext® UltraTM RNA Library Prep Kit for Illumina (NEB, USA), and sequenced on an Illumina NovaSeq platform. Transcripts per kilobase of exon model per million mapped reads (TPM) was calculated based on the length of the gene and reads count mapped to this gene. Differential gene expression analysis was performed using the DESeq2 R package (1.16.1).

## Data availability

The sequencing data produced in this study have been deposited in the Gene Expression Omnibus (GEO) repository under the accession numbers GSE305663 (https://www.ncbi.nlm.nih.gov/geo/query/acc.cgi?acc=GSE305663), GSE305664 (https://www.ncbi.nlm.nih.gov/geo/query/acc.cgi?acc=GSE305664), GSE277835 (https://www.ncbi.nlm.nih.gov/geo/query/acc.cgi?acc=GSE277835), GSE277838 (https://www.ncbi.nlm.nih.gov/geo/query/acc.cgi?acc=GSE277838) and GSE277839 (https://www.ncbi.nlm.nih.gov/geo/query/acc.cgi?acc=GSE277839).

The source data of this paper are collected in the following database record: biostudies:S-SCDT-10_1038-S44319-026-00759-8.

## Peer review information

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

## Acknowledgements

We thank Dr. Tingjun Lei for the in vitro-transcribed RNA and LNP preparations. We thank Mr. Shucheng Zhao for his dedication to this research project. This work was supported by the National Natural Science Foundation of China (32271348 and 82473224), the Science and Technology Department of Sichuan Province (2024NSFSC0731), and the West Women and Children Medical Research Development Center (XBFY-YJXM2025004).

## Author contributions

**Yin Fang**: Formal analysis; Validation; Investigation; Methodology; Writing—original draft; Writing—review and editing. **Tong Qiu**: Data curation; Investigation. **Hong Luo**: Investigation. **Yan Wang**: Investigation. **Chao Yang**: Data curation. **Min Wang**: Investigation. **Qian Dai**: Investigation. **Wenyue Zheng**: Resources. **Rutie Yin**: Resources. **Xue Xiao**: Resources. **Qintong Li**: Conceptualization; Resources; Data curation; Formal analysis; Supervision; Funding acquisition; Writing—original draft; Writing—review and editing.

Source data underlying figure panels in this paper may have individual authorship assigned. Where available, figure panel/source data authorship is listed in the following database record: biostudies:S-SCDT-10_1038-S44319-026-00759-8.

## Disclosure and competing interests statement

The authors declare no competing interests.

# Expanded View Figures

**Figure EV1.  *Tut1* heterozygous mice generated by nestin-Cre (*Tut1^{f/+}*;nestin-Cre) are normal without behavioral abnormalities.** ▶

(A) The number of pups recovered from *Tut1^{f/+}*;nestin-Cre intercrosses. (B) Sagittal sections and H&E staining of P70 wild-type (WT) and *Tut1^{f/+}*;nestin-Cre brains ($n = 3$). Scale bar: 50 μm. (C) Open field test. Representative traces of WT ($n = 17$) and *Tut1^{f/+}*;nestin-Cre ($n = 16$) mice in the open field (left panel), and time spent in the indicated areas (right panel). The data represents mean ± SEM. *P* values, unpaired Student's *t*-test. (D) Tree-chamber sociability and social novelty test. Representative traces of WT ($n = 16$) and *Tut1^{f/+}*;nestin-Cre (Het) ($n = 15$) mice in three-chamber (left panel), and total sniffing time spent toward stranger 1 or stranger 2 (right panel). The data represents mean ± SEM. *P* values, unpaired Student's *t*-test. (E) Elevated plus maze test. Representative traces of WT ($n = 17$) and *Tut1^{f/+}*;nestin-Cre (Het) ($n = 16$) mice in the elevated plus maze (left panel), and time spent in the indicated areas (right panel). The data represent mean ± SEM. *P* values, unpaired Student's *t*-test. (F) Morris water maze test. Representative traces of WT ($n = 19$) and *Tut1^{f/+}*;nestin-Cre (Het) ($n = 20$) mice in Morris water maze (left panel), and the number of target quadrant crossings as well as time spent in the indicated areas (right panel). The data represent mean ± SEM. *P* values, unpaired Student's *t*-test. (G) The knockout efficiency of *Tut1^{f/f}*;Camk2a-Cre, determined by qPCR analysis of *Tut1* exon 2 expression as in Fig. 4C. Source data are available online for this figure.

**A**

Tut1^{f/+};nestin-Cre X Tut1^{f/+};nestin-Cre

| Stage | Genotype | | |
|---|---|---|---|
| | WT | Tut1^{f/+};nestin-Cre | Tut1^{f/f};nestin-Cre |
| P7 | 29 | 24 | 0 |
| 10 months | 29 | 24 | 0 |

**B**

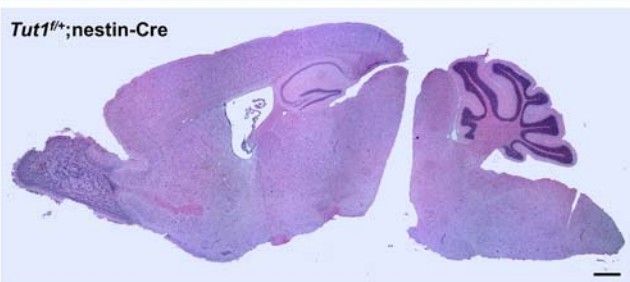

P70

**C**

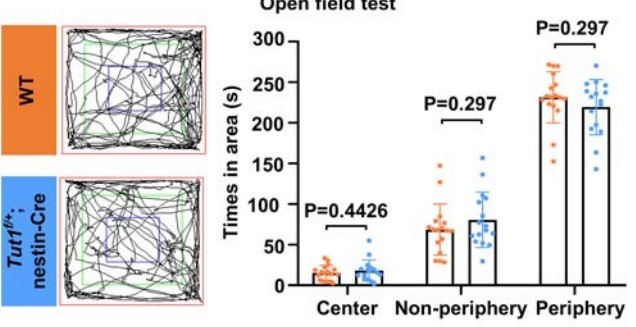

Open field test

**D**

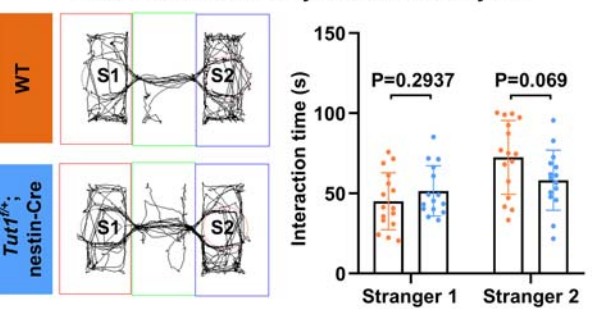

Three-chamber sociability and social novelty test

**E**

Elevated plus maze

**F**

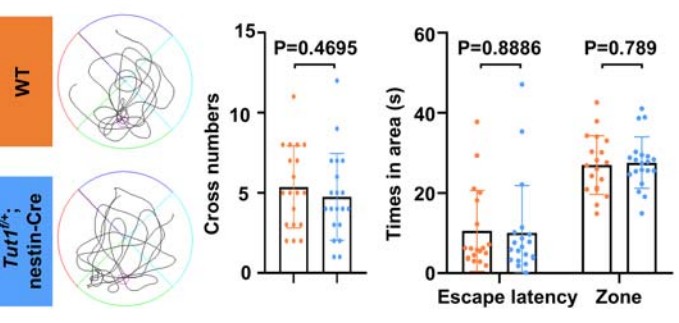

Morris water maze

**G**

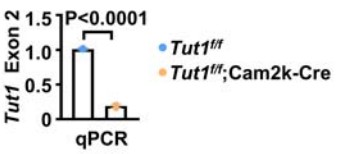

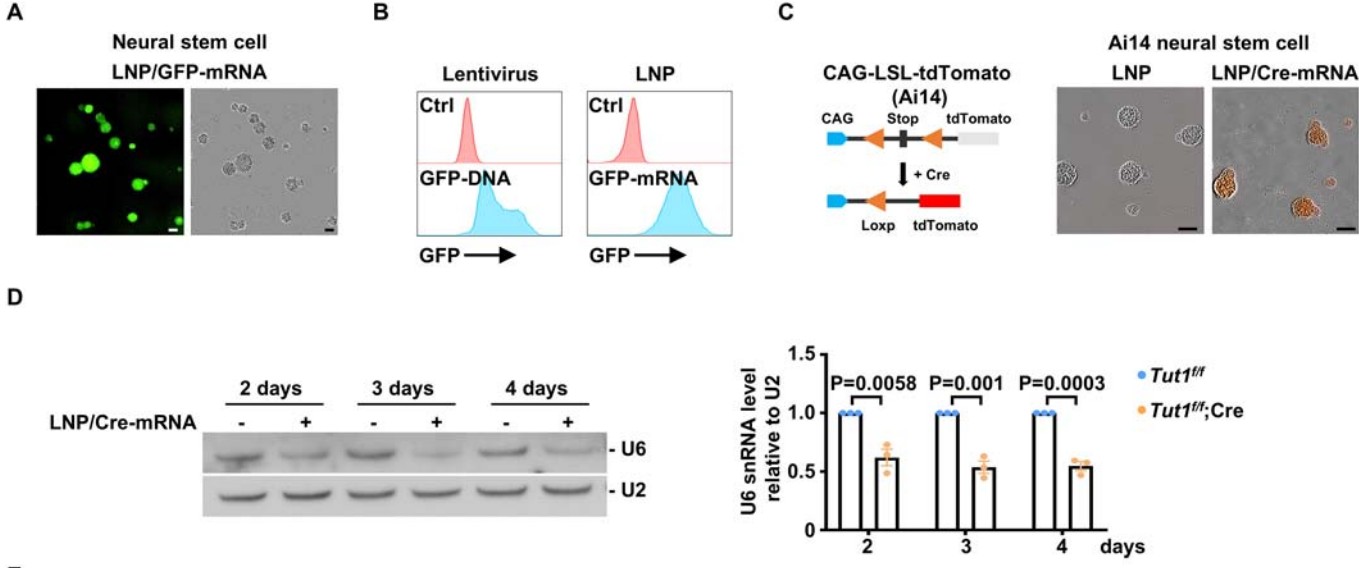

**A** Neural stem cell LNP/GFP-mRNA

**B** Lentivirus / LNP — GFP flow cytometry (Ctrl, GFP-DNA, GFP-mRNA)

**C** CAG-LSL-tdTomato (Ai14); Ai14 neural stem cell LNP / LNP/Cre-mRNA

**D**

LNP/Cre-mRNA: 2 days (−/+), 3 days (−/+), 4 days (−/+) — U6, U2 bands.

Bar graph: U6 snRNA level relative to U2. P=0.0058, P=0.001, P=0.0003 at 2, 3, 4 days. *Tut1^{f/f}* vs *Tut1^{f/f}*;Cre.

**E**

| Sample | Data (Gb) | Total reads | % PASS | % full length | Median pass read length | Mean pass read length | No. of genes detected | % genes with AS |
|---|---|---|---|---|---|---|---|---|
| WT-1 | 12.022 | 26144584 | 33.24 | 88.11 | 623 | 900 | 13263 | 58.50% |
| WT-2 | 7.022 | 9851047 | 77.97 | 80.24 | 684 | 842 | 13549 | 56.40% |
| WT-3 | 7.343 | 11946731 | 74.21 | 82.99 | 621 | 737 | 13637 | 55.10% |
| *Tut1^{f/f}*;Cre-1 | 7.365 | 10899850 | 76.67 | 81.26 | 661 | 803 | 13689 | 55.80% |
| *Tut1^{f/f}*;Cre-2 | 10.536 | 16970857 | 72.30 | 77.43 | 632 | 762 | 13894 | 56.20% |
| *Tut1^{f/f}*;Cre-3 | 6.736 | 8966639 | 80.15 | 79.49 | 700 | 871 | 13555 | 55.80% |
| *p53^{f/f}*;Cre-1 | 8.24 | 12658680 | 76.61 | 81.63 | 651 | 772 | 13698 | 55.80% |
| *p53^{f/f}*;Cre-1 | 8.401 | 11383350 | 79.96 | 82.17 | 691 | 855 | 13742 | 56.90% |
| *p53^{f/f}*;Cre-1 | 7.597 | 10987475 | 79.62 | 84.05 | 666 | 801 | 13612 | 56.30% |
| *p53^{f/f}*;*Tut1^{f/f}*;Cre-1 | 7.393 | 9704333 | 79.20 | 79.15 | 707 | 893 | 13620 | 56.30% |
| *p53^{f/f}*;*Tut1^{f/f}*;Cre-1 | 6.934 | 9500305 | 80.38 | 80.95 | 686 | 841 | 13603 | 55.60% |
| *p53^{f/f}*;*Tut1^{f/f}*;Cre-1 | 6.311 | 9576746 | 76.89 | 82.41 | 648 | 778 | 13435 | 54.80% |

Figure EV2. **Analysis of *Tut1* function in neural stem cells in vitro.**

(A) Incucyte live-cell imaging analysis of neural stem cells for visual inspection of overall cell morphology (n = 3), with or without transfection by LNP containing GFP-mRNA (LNP/GFP-mRNA),. Scale bar: 100 μm. (B) Lentiviral transduction of neural stem cells with GFP cDNA (left, n = 3) or LNP transfection with GFP-mRNA (LNP/GFP-mRNA) (right, n = 3). Flow cytometry was used to analyze GFP-positive cells 2 days after transduction or transfection. Notably, compared to lentiviral transduction, LNP transfection showed a higher percentage of GFP-positive cells and more uniform GFP expression. (C) Ai14 neural stem cells transfected with LNP/Cre-mRNA. (D) Northern blot analysis of U6 snRNA expression in neural stem cells after LNP/Cre-mRNA transfection. Time course analysis of U6 snRNA expression. Total RNA was derived from *Tut1^{f/f}* neural stem cells with or without LNP/Cre-mRNA transfection for 2, 3, or 4 days (n = 3). U2 snRNA was used as a loading control. (E) Nanopore-sequencing statistics for Fig. 4H. The percentage of full-length (% full length) reads was calculated as the number of reads covering 80% of nucleotides for the transcript. The number of genes (No. of genes detected) was computed by counting the genes, including all the isoforms. Genes with multiple isoforms identified were considered alternatively spliced (% genes with AS). Source data are available online for this figure.

| Sample | Data (Gb) | Total reads | % PASS | % full length | Median pass read length | Mean pass read length | No. of genes detected | % genes with AS |
|---|---|---|---|---|---|---|---|---|
| *Tut1^{f/f}*-1 | 1.606 | 7436242 | 96.72 | 54.83 | 728 | 1068 | 12442 | 61.50% |
| *Tut1^{f/f}*-2 | 1.69 | 7886748 | 97.19 | 56.50 | 687 | 1011 | 12788 | 61.30% |
| *Tut1^{f/f}*;Emx1-Cre-1 | 1.712 | 7903509 | 96.91 | 56.28 | 696 | 1039 | 13158 | 61.50% |
| *Tut1^{f/f}*;Emx1-Cre-2 | 1.247 | 5850069 | 97.24 | 61.85 | 662 | 975 | 12522 | 60.30% |

**Figure EV3.  Nanopore-sequencing statistics for Fig. 5.**

The percentage of full-length (% full length) reads was calculated as the number of reads covering 80% of nucleotides for the transcript. The number of genes (No. of genes detected) was computed by counting the genes, including all the isoforms. Genes with multiple isoforms identified were considered alternatively spliced (% genes with AS).

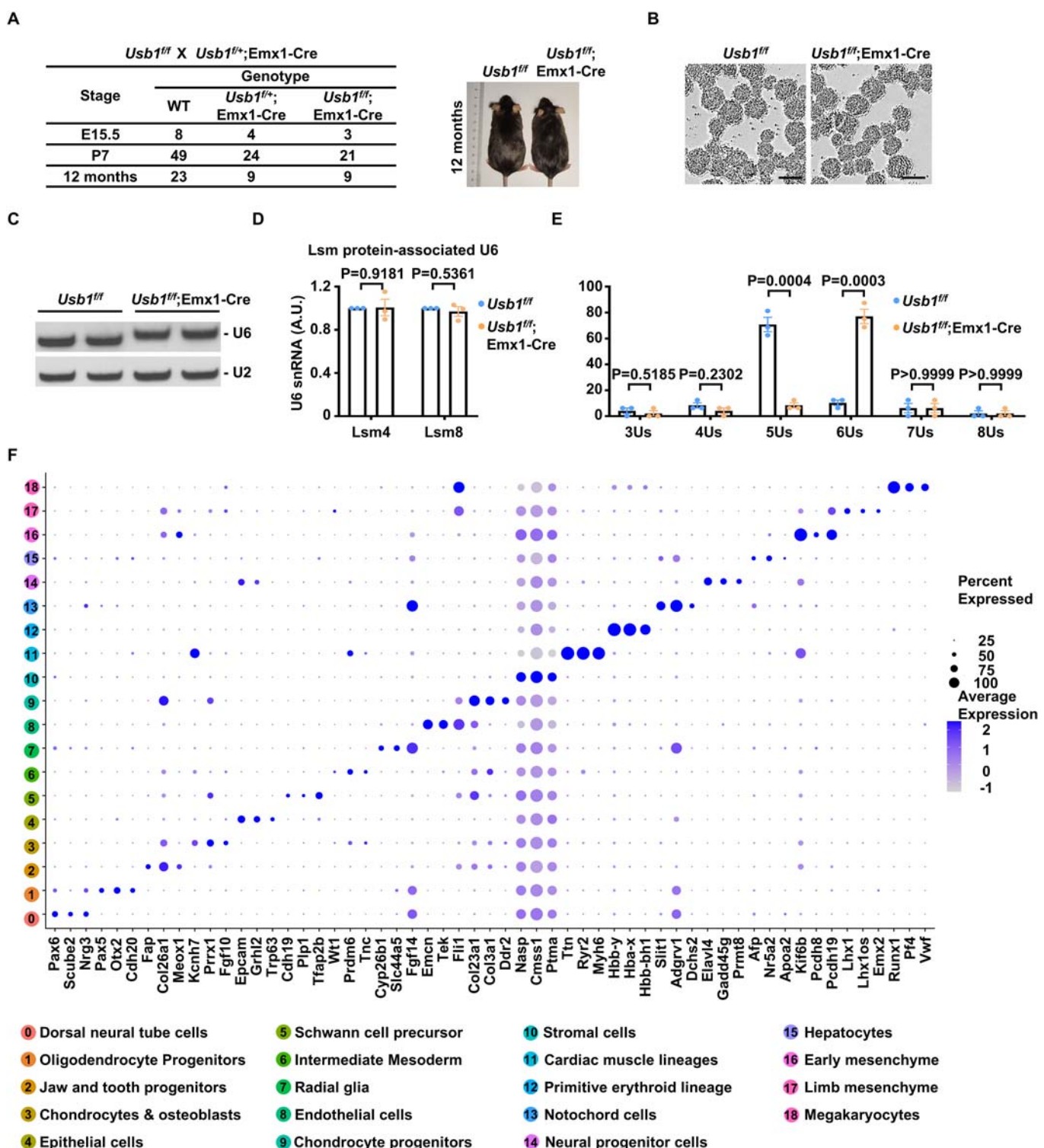

**A**

| Stage | WT | *Usb1^{f/+}*;<br>Emx1-Cre | *Usb1^{f/f}*;<br>Emx1-Cre |
|-------|-----|---------------------------|---------------------------|
| E15.5 | 8 | 4 | 3 |
| P7 | 49 | 24 | 21 |
| 12 months | 23 | 9 | 9 |

*Usb1^{f/f}* X *Usb1^{f/+}*;Emx1-Cre

**F**

0 Dorsal neural tube cells
1 Oligodendrocyte Progenitors
2 Jaw and tooth progenitors
3 Chondrocytes & osteoblasts
4 Epithelial cells
5 Schwann cell precursor
6 Intermediate Mesoderm
7 Radial glia
8 Endothelial cells
9 Chondrocyte progenitors
10 Stromal cells
11 Cardiac muscle lineages
12 Primitive erythroid lineage
13 Notochord cells
14 Neural progenitor cells
15 Hepatocytes
16 Early mesenchyme
17 Limb mesenchyme
18 Megakaryocytes

**Figure EV4. Analysis of *Usb1* function by germline and conditional knockout mouse models.**

(A) Genotype analyses of pups generated from crossing of $Usb1^{f/+}$;Emx1-Cre and $Usb1^{f/f}$ mice. Wild-type (WT) denotes both $Usb1^{f/+}$ and $Usb1^{f/f}$ genotypes. Littermates of $Usb1^{f/f}$ and $Usb1^{f/+}$;Emx1-Cre mice at the age of 12 months were shown ($n = 21$). (B) Phase contrast images of cultured neural stem cells derived from E12.5 wild-type and $Usb1^{f/f}$;Emx1-Cre neocortices ($n = 3$), respectively. Scale bars: 100 μm. (C) RNA blot analysis of U6 snRNA. Total RNA was extracted from wild-type (WT) and *Usb1* knockout ($Usb1^{f/f}$;Emx1-Cre) neural stem cells ($n = 3$). U2 snRNA was used as a loading control. Of note, the length of U6 snRNA was extended in $Usb1^{f/f}$;Emx1-Cre cells. (D) The interaction between U6 snRNA and Lsm proteins in $Usb1^{f/f}$;Emx1-Cre neural stem cells. Lsm4- or Lsm8-associated U6 snRNA was quantified by qPCR. The data represent mean ± SEM. AU arbitrary unit. (E) 3′ RACE analysis of U6 oligo(U) tails in wild-type ($Usb1^{f/f}$) and *Usb1* knockout ($Usb1^{f/f}$;Emx1-Cre) neural stem cells ($n = 3$). The data represent mean ± SEM. P values, unpaired Student's t-test. (F) Dot plot showing the expression of three marker genes for each cell type. The size of the dot denotes the percentage of cells expressing the indicated genes, and the color of the dot denotes the average expression level. Source data are available online for this figure.

