## [Peer Review File · EMBO Reports]

TUT1-catalyzed U6 snRNA 3'-end maturation is essential for RNA splicing and stem cell survival

Yin Fang, Tong Qiu, Hong Luo, Yan Wang, Chao Yang, Min Wang, Qian Dai, Wenyue Zhen, Rutie Yin, Xue Xiao, and Qintong Li

Corresponding author(s): Qintong Li (liqintong@scu.edu.cn)

Review Timeline:

Submission Date:	22nd Aug 25
Editorial Decision:	30th Sep 25
Revision Received:	12th Jan 26
Editorial Decision:	12th Feb 26
Revision Received:	18th Feb 26
Accepted:	20th Mar 26

Editor: Achim Breiling

Transaction Report:

Dear Prof. Li,

Thank you for the transfer of your manuscript to EMBO reports. I have now received reports from the three referees that were asked to evaluate your study, which can be found at the end of this email. As you will see, the referees think that these findings are of interest. However, they have several comments, concerns, and suggestions, indicating that a major revision of the manuscript is necessary to allow publication of the study in EMBO reports. As the reports are below, and all the referee concerns need to be addressed, I will not detail them here.

Given the constructive referee comments, I would thus like to invite you to revise your manuscript with the understanding that the concerns of the referees must be addressed in the revised manuscript and/or in a detailed point-by-point response. Acceptance of your manuscript will depend on a positive outcome of a second round of review. It is EMBO reports policy to allow a single round of revision only and acceptance of the manuscript will therefore depend on the completeness of your responses included in the next, final version of the manuscript.

- 1) a .docx formatted version of the final manuscript text (including legends for main figures, EV figures and tables), but without the figures included. Figure legends should be compiled at the end of the manuscript text.
- 2) individual production quality figure files as .eps, .tif, .jpg (one file per figure), of main figures and EV figures. Please upload these as separate, individual files upon re-submission.

- 4) a complete author checklist, which you can download from our author guidelines

(<https://www.embopress.org/page/journal/14693178/authorguide>). Please insert page numbers in the checklist to indicate where the requested information can be found in the manuscript. The completed author checklist will also be part of the RPF.

- 5) that primary datasets produced in this study (e.g. RNA-seq, ChIP-seq, structural and array data) are deposited in an appropriate public database. If no primary datasets have been deposited, please also state this in a dedicated section (e.g. 'No

primary datasets have been generated and deposited'), see below.

The accession numbers and database should be listed in a formal "Data Availability" section that follows the model below. This is now mandatory (like the COI statement). Please note that the Data Availability Section is restricted to new primary data that are part of this study. This section is mandatory. As indicated above, if no primary datasets have been deposited, please state this in this section

Data availability

6) We now request the publication of original source data with the aim of making primary data more accessible and transparent to the reader. You will receive a separate email with instructions for providing source data with your revised manuscript, including information how to upload and organize the files.

8) Regarding data quantification and statistics, please make sure that the number "n" for how many independent experiments were performed, their nature (biological versus technical replicates), the bars and error bars (e.g. SEM, SD) and the test used to calculate p-values is indicated in the respective figure legends (also for EV and Appendix figures). Please also check that all the p-values are explained in the legend, and that these fit to those shown in the figure. Please provide statistical testing where applicable. Please avoid the phrase 'independent experiment' but clearly state if these were biological or technical replicates. Please also indicate (e.g. with n.s.) if testing was performed, but the differences are not significant. In case n=2, please show the data as separate datapoints without error bars and statistics. See also:
<http://www.embopress.org/page/journal/14693178/authorguide#statisticalanalysis>

9) Please add scale bars of similar style and thickness to all microscopic images, using clearly visible black or white bars (depending on the background). Please place these in the lower right corner of the images themselves. Please do not write on or near the bars in the image but define the size in the respective figure legend.

10) Please also note our reference format:

12) We now use CRedit to specify the contributions of each author in the journal submission system. CRedit replaces the author contribution section. Please use the free text box to provide more detailed descriptions and do NOT provide your final manuscript text file with an author contributions section. See also our guide to authors:
<https://www.embopress.org/page/journal/14693178/authorguide#authorshipguidelines>

13) All Materials and Methods need to be described in the main text using our 'Structured Methods' format, which is required for all research articles. According to this format, the Methods section should include a Reagents and Tools Table (listing key

reagents, experimental models, software, and relevant equipment and including their sources and relevant identifiers), uploaded as separate file, and a Methods section in which we encourage the authors to describe their methods using a step-by-step protocol format with bullet points, to facilitate the adoption of the methodologies across labs. More information on how to adhere to this format as well as downloadable templates (.doc) for the Reagents and Tools Table can be found in our author guidelines (section 'Structured Methods'):

14) Please add up to five keywords to the manuscript and order the manuscript sections like this, using only these names: Title page - Abstract - Keywords - Introduction - Results - Discussion - Methods - Data availability section - Acknowledgements (please put here all the funding information) - Disclosure and Competing Interests Statement - References - Figure legends - Expanded View Figure legends

15) Please make sure that all the funding information is also entered into the online submission system and that it is complete and similar to the one in the acknowledgement section of the manuscript text file.

I look forward to seeing a revised form of your manuscript when it is ready.

Yours sincerely,

Referee #1:

This study deals with the physiological roles of two enzymes, TUT1 and USB1, in the 3' end maturation of U6 snRNA using a mouse model. While conventional models suggested that both enzymes act sequentially in U6 snRNA maturation, this paper shows that TUT1 is essential for stem cell survival and RNA splicing, whereas USB1 is functionally dispensable in many embryonic cell types. TUT1 deficiency induces DNA damage and cell death, revealing its critical role in maintaining neural stem cells. These findings advance our understanding that the TUT1-catalyzed oligo(U) tail at the 3' end of U6 snRNA plays a key role in regulating RNA splicing, rather than USB1, which trims the 3'-oligo(U) for U6 snRNA stabilization. These findings suggest that TUT1 and USB1 may have distinct functions in the pathogenesis of human diseases associated with these enzymes.

Overall, the manuscript is well organized and clearly communicated. However, the conclusion that "We propose that TUT1, but not USB1, is a ubiquitous U6 snRNA 3'-end maturation or repair factor required for RNA splicing" is not sufficiently supported by the experimental results presented here. While this paper demonstrates that the phenotype in mouse development is weaker in USB1 KO than in TUT1 KO, the impact of USB1 on splicing has not been explicitly investigated.

Comments:

The oligonucleotide (U) tail catalyzed by TUT1 has been shown to promote the interaction between the essential splicing factor Lsm2-8 protein complex and U6 snRNA. In TUT1-deficient neural stem cells, the amount of U6 snRNA binding to Lsm4 or Lsm8 decreased (Figure 4G); however, it is necessary to confirm whether this is also the case in cells lacking USB1.

Page 4, line 62 should cite the following paper:

Montemayor, E.J., Didychuk, A.L., Yake, A.D. et al. Architecture of the U6 snRNP reveals specific recognition of 3'-end processed U6 snRNA. *Nat Commun* 9, 1749 (2018). <https://doi.org/10.1038/s41467-018-04145-4>

Related to comment 1), the total U6 snRNA (input) should be shown in Figure 4G. It might be possible that the expression level

of U6 snRNA is changed by TUT1 KO.

While it has been clearly demonstrated that TUT1 deficiency causes widespread RNA splicing defects (such as increased intron retention) affecting numerous genes, this manuscript does not show any direct data indicating that USB1 deficiency does not cause similar splicing defects.

Page 20, lines 420-429: The authors should confirm the length and composition of the 3'-tail of U6 snRNAs by cloning the U6 snRNA from USB1-null tissues and embryos.

Several references are not properly cited in the Discussion section. For example, on page 25, lines 473, 481, and 491, the relevant literature should be properly cited. Line 491 should cite the following paper on the recent complex structure of TUT1 and U6 snRNA:

Seisuke Yamashita, Kozo Tomita. Cryo-EM structure of human TUT1:U6 snRNA complex. *Nucleic Acids Research*, Volume 53, Issue 2, 27 January 2025, gkae1314, <https://doi.org/10.1093/nar/gkae1314>

Line 97: "day (E) 3.5" should be "day E3.5."

Line 462: "even although" should be "even though."

Line 778: "TUT1 catalyzes the post-transcriptional addition of an oligo(U) tail (depicted by 4 Us in red)" should be "TUT1 catalyzes the post-transcriptional addition of an oligo(U) tail (depicted by 2 Us in red)."

Referee #2:

In this manuscript, Fang et. al develop and characterize new mouse models to study biogenesis of the core spliceosome component U6 snRNA. They find that TUT1, but not Usb1, is required during embryogenesis. This result, recapitulated through several mouse models and stem cell models, challenges the prevailing idea that these two enzymes are equivalently important for U6 function, but does explain the human genetic data suggesting that loss of TUT1 results in stronger phenotypes. I cannot evaluate the mouse genetic work as this is outside my expertise. Overall, I found this to be an interesting manuscript that will be important to the field. However, I have the following key points, including an additional experiment and rewriting.

Major comments:

1. Transfected U6 snRNA with 4U's partially rescued the TUT1 knockout phenotype, while 6U's fully rescued TUT1 k.o. (Fig. 4F). The partial rescue by U6-4U could be because U6 snRNA is unstable in the absence of TUT1/3' end extension, and additional transfected U6 snRNA of the minimum appropriate length (4U, which is what is genomically encoded) can rescue the phenotype even though it's not the ideal length/ideally modified. The authors should test whether loss of TUT1 reduces U6 stability (expression level), ideally by probing for U6 on a northern blot. This experiment will also (likely) show that the U6 snRNA 3' tail is not extended in the absence of TUT1. This is an important aspect to their model and more important than showing a loss of interaction with the Lsm2-8 complex (Fig. 4G). In fact, it's a critical part of their model, as it's possible that TUT1 has a secondary effect that could lead to the drastic phenotype. The result of this experiment will influence their final model and claims (Fig. 7K). They claim that TUT1-extended RNA (and the 3'OH it leaves) is essential for splicing and viability, while a shortened cyclic phosphate tail installed by Usb1 is dispensable. The authors should note that Lsm2-8 can bind U tails with 3'OH, although the affinity is decreased. I think it's more likely that a 4U tail is too short for La/Lsm2-8 to bind and protect U6 from degradation, and in the absence of TUT1, you lose U6 expression. This experiment should be easily achievable, given that the authors probe for U6 in Fig. 6B.

2. I find the alternative splicing analysis (Fig. 4H-K and Fig. 5) a little surprising and superficial. First, very few splicing events are altered. Is this a technical challenge with read depth, or are there really few changes? Second, why would RNA splicing be enriched in the GO analysis? Do splicing components have more introns, or less consensus splice sites, etc.? It would be helpful to improve this analysis for both Figure 4 and 5.

3. Why was RNA-seq instead of long read sequencing performed for the Usb1 knockout? Given that few differentially expressed genes are detected in the TUT1 knockout (Fig. 5), this does not seem to be the appropriate assay to make the claim that Usb1 levels do not affect RNA splicing.

4. The Materials and Methods lack sufficient details in many sections, particularly the molecular biology. How was U6 snRNA in vitro transcribed? The citation (#33) refers to mRNA (capped and modified). How was the 3' end generated - runoff transcription from a template (which usually gives heterogenous but 3'OH products) or an HDV ribozyme (which does not generate a 3'OH)? How was the Lsm4/8 IP performed? How were cell lines grown and maintained? How was the "RNA blot" in Fig. 6B performed, and which primers were used, as there is no mention of this in the supplemental table?

5. Many figure legends (i.e., Fig. 4B) are insufficient. For Fig. 4B/E, how much RNA was loaded? What kind of gel? What size

are the products (a ladder would help)? This is a pattern throughout the paper.

Minor comments:

Line 38 typo - "Catalytical"

Line 293 and 295 - refers to Fig 4E to show efficiency of knockdown, but this is a gel of purified U6 snRNA.

Referee #3:

The TUT1 gene (now renamed as TENT1) participates in U6 snRNA processing and maintenance by adding uridines to the 3' end of U6 snRNA. The terminal uridine in U6 snRNA ends in a cyclic 2',3' phosphate, formed by the exonucleolytic activity of USB1, which helps stabilize U6 snRNA. Whether there are other substrates (other than U6 snRNA) for these enzymes, and what they are, is not clear. This paper shows that deletion of TUT1 in mice results in a different phenotype than deletion of USB1. However they do show that *Usb1* is an essential gene, since the *Usb1* KO mice die after E12.5, although it is not essential in all cells (since they knocked it out in embryonic stem cells, and these were not affected). As far as I can tell this lethality in the animal of a *Usb1* KO has never been reported before, and the authors should certainly consider this a main point of their paper. *Usb1* has been reported to also remove A-tails from some adenylated RNAs and that activity should be mentioned, since that activity might be critical for it being an essential gene (Huynh & Parker, 2023; Jeong et al., 2023; Mroczek & Dziembowski, 2013; Mroczek et al., 2012).

Specific comments:

1. The results of Fig.1-4 clearly that TUT1 is required for ES cell, early embryo development and development of the brain in mice. Somewhat surprisingly in *Camk2a-cre* conditional *Tut1* KO mice, the mice survive and develop normally. They do not indicate whether they tested whether the removal of TUT1 was successful in the cerebellum of the *fl/fl* mice. It seems like that control should have been done.
2. The loss of TUT1 in cultured neural stem cells in vitro can be rescued by injection of U6 snRNA in vitro if U6 snRNA with 6 uridines at the 3' end was introduced into cells. Somewhat surprisingly the levels of U6 snRNA bound to the LSm proteins was only reduced 50% compared to the unfloxed cells, which I would not have expected to affect splicing that much. Also in the last sentence on page 17 (l. 359). "These results demonstrate that TUT1 regulates mRNA splicing during embryogenesis". All this shows is that (as expected) *Tut1* is necessary for splicing during embryogenesis.
3. For the alternative splicing data, they need to indicate how many total mRNAs they sequenced and how many shows the splicing defects. Since they did long-read sequencing (presumably from the polyA tail; they don't mention that) they know how many mRNAs were sequenced from each sample and how many of those have splicing defects. Since the non-productive intron retention events should result in NMD and rapid loss of the transcripts it is likely that they are much more abundant than they observed and they should mention that possibility. It makes sense that inefficient splicing may result in more intron-retention events as they observed.
4. On page 20 (and particularly in the abstract, as well as throughout the manuscript), they should emphasize that *Usb1* is also an essential gene, with embryos dying after E10.5 and before E12.5 (Fig.7F). They should likely put this in the title of the paper also..
5. The data in Fig. 6J does not make sense. They show that in wild-type cells and in the heterozygotes (lane 1 and 2), U6 snRNA is shorter than in the *Usb1* KO cells. This suggests that the cyclin phosphate may affect the mobility of U6 snRNA and/or the loss of *Usb1* results in TUT1 being able to produce U6 snRNA with longer U tails (the length of the U tail will be determined by both the elongation of U6 snRNA by TUT1, and the removal of the U tail by exonucleases). The only way to accurately determine the length of the U6 tail is by sequencing the 3' end of the molecule directly.

A critical role for the phosphate at the end of U6 snRNA in splicing has been suggested by the work of Brow and coworkers (Montemayor et al., 2018)

References

- Huynh, T. N., & Parker, R. (2023). The PARN, TOE1, and USB1 RNA deadenylases and their roles in non-coding RNA regulation. *J Biol Chem*, 299(9), 105139. doi:10.1016/j.jbc.2023.105139
- Jeong, H. C., Shukla, S., Fok, W. C., Huynh, T. N., Batista, L. F. Z., & Parker, R. (2023). USB1 is a miRNA deadenylase that regulates hematopoietic development. *Science*, 379(6635), 901-907. doi:10.1126/science.abj8379
- Montemayor, E. J., Didychuk, A. L., Yake, A. D., Sidhu, G. K., Brow, D. A., & Butcher, S. E. (2018). Architecture of the U6 snRNP reveals specific recognition of 3'-end processed U6 snRNA. *Nat Commun*, 9(1), 1749. doi:10.1038/s41467-018-04145-4
- Mroczek, S., & Dziembowski, A. (2013). U6 RNA biogenesis and disease association. Wiley. *Interdiscip. Rev. RNA*. doi:10.1002/wrna.1181 [doi]
- Mroczek, S., Krwawicz, J., Kutner, J., Lazniewski, M., Kucinski, I., Ginalski, K., & Dziembowski, A. (2012). C16orf57, a gene mutated in poikiloderma with neutropenia, encodes a putative phosphodiesterase responsible for the U6 snRNA 3' end modification. *Genes Dev*, 26(17), 1911-1925. doi:gad.193169.112 [pii];10.1101/gad.193169.112 [doi]

Response to Reviewers:

We sincerely thank all three Reviewers for their effort and time invested in our manuscript. Our point-by-point responses to their extensive and constructive comments are appended below in blue.

Referee #1:

This study deals with the physiological roles of two enzymes, TUT1 and USB1, in the 3' end maturation of U6 snRNA using a mouse model. While conventional models suggested that both enzymes act sequentially in U6 snRNA maturation, this paper shows that TUT1 is essential for stem cell survival and RNA splicing, whereas USB1 is functionally dispensable in many embryonic cell types. TUT1 deficiency induces DNA damage and cell death, revealing its critical role in maintaining neural stem cells. These findings advance our understanding that the TUT1-catalyzed oligo(U) tail at the 3' end of U6 snRNA plays a key role in regulating RNA splicing, rather than USB1, which trims the 3'-oligo(U) for U6 snRNA stabilization. These findings suggest that TUT1 and USB1 may have distinct functions in the pathogenesis of human diseases associated with these enzymes.

Overall, the manuscript is well organized and clearly communicated. However, the conclusion that "We propose that TUT1, but not USB1, is a ubiquitous U6 snRNA 3'-end maturation or repair factor required for RNA splicing" is not sufficiently supported by the experimental results presented here. While this paper demonstrates that the phenotype in mouse development is weaker in USB1 KO than in TUT1 KO, the impact of USB1 on splicing has not been explicitly investigated.

Thank you for your constructive feedback, which has helped to improve the quality of the manuscript. We agree with the Reviewer that our data only show that "*Usb1* is functionally dispensable in many embryonic cell types", but do not speak to the role of *Usb1* in splicing. In the revised manuscript, we have changed the text to accurately reflect this point in multiple places.

Comments:

The oligonucleotide (U) tail catalyzed by TUT1 has been shown to promote the interaction between the essential splicing factor Lsm2-8 protein complex and U6

snRNA. In TUT1-deficient neural stem cells, the amount of U6 snRNA binding to Lsm4 or Lsm8 decreased (Figure 4G); however, it is necessary to confirm whether this is also the case in cells lacking USB1.

Following your suggestion, we carried out additional experiments to examine whether the interaction between U6 snRNA and Lsm proteins is compromised in *Usb1*-null neural stem cells. Unlike in *Tut1*-deficient cells, this interaction is not significantly altered in *Usb1*-null cells (new data, Appendix Figure S7 in the revised manuscript). The new data is reproduced here for your reference:

We added additional text in the revised manuscript as follows:

In Results, Line 420-424:

“We derived *Usb1*-null neural stem cells from *Usb1^{ff};Emx1-Cre* dorsal pallium (Appendix Fig. S7). These cells proliferated normally as their wild-type counterparts. The expression level of U6 snRNA was comparable in wild-type and mutant cells, but the length of U6 snRNA was extended as expected in *Usb1*-null cells. Nevertheless, the amount of U6 snRNA associated with Lsm4 or Lsm8 protein was unaltered in *Usb1*-null cells.”

In Appendix Figure S7:

“(A) Phase contrast images of cultured neural stem cells derived from E12.5 wild-type and *Usb1^{ff};Emx1-Cre* neocortices (n=3), respectively. Scale bars: 100 μ m.

(B) RNA blot analysis of U6 snRNA. Total RNA was extracted from wild-type (WT) and *Usb1* knockout (*Usb1^{ff};Emx1-Cre*) neural stem cells (n=3). U2 snRNA was used as a loading control. Of note, the length of U6 snRNA was extended in *Usb1^{ff};Emx1-Cre* cells.

(C) The interaction between U6 snRNA and Lsm proteins in *Usb1^{ff};Emx1-Cre* neural stem cells. Lsm4- or Lsm8-associated U6 snRNA was quantitated by qPCR. The data represents mean \pm SEM. A.U., arbitrary unit.”

Page 4, line 62 should cite the following paper:

Montemayor, E.J., Didychuk, A.L., Yake, A.D. et al. Architecture of the U6 snRNP reveals specific recognition of 3'-end processed U6 snRNA. Nat Commun 9, 1749 (2018). <https://doi.org/10.1038/s41467-018-04145-4>

This reference has been added in the revised manuscript (Line 70).

Related to comment 1), the total U6 snRNA (input) should be shown in Figure 4G. It might be possible that the expression level of U6 snRNA is changed by TUT1 KO.

Following your suggestion, we examined U6 snRNA level in neural stem cells by northern blot, after LNP/Cre-mRNA transfection for 2, 3 or 4 days (new data, Appendix Figure S4 in the revised manuscript). We found that U6 snRNA level was decreased at day 2 after transfection. In light of this new finding, we added additional text in the revised manuscript as follows:

In Results, Line 312-316:

“When *Tut1* was ablated in neural stem cells (2 days after LNP/Cre-mRNA transfection), the amount of U6 snRNA associated with Lsm4 or Lsm8 protein was decreased (Fig. 4G). Interestingly, the total U6 snRNA level was also decreased at this timepoint (Appendix Fig. S4), indicating that the 3'-end oligo(U) tail promotes the interaction between Lsm2-8 and U6 snRNA.”

The new data (Appendix Figure S4 in the revised manuscript) is reproduced here for your reference:

In Appendix Figure S4:

“Appendix Figure S4. Northern blot analysis of U6 snRNA expression in neural stem cells after LNP/Cre-mRNA transfection.

Time course analysis of U6 snRNA expression. Total RNA was derived from *Tut1^{ff}* neural stem cells with or without LNP/Cre-mRNA transfection for 2, 3 or 4 days (n=3).

U2 snRNA was used as a loading control.”

While it has been clearly demonstrated that TUT1 deficiency causes widespread RNA splicing defects (such as increased intron retention) affecting numerous genes, this manuscript does not show any direct data indicating that USB1 deficiency does not cause similar splicing defects.

Following your suggestion, we reanalyzed our RNA-seq datasets (Figure 6F and 6G), and used rMATS to deduce differentially regulated splicing events in wild-type and *Usb1* knockout cells (new data, Figure 6H-6L in the revised manuscript). We added additional text in the revised manuscript as follows:

In Results, Line 395-408:

“To deduce differentially regulated splicing events in wild-type and *Usb1* knockout cells, we used rMATS (Wang et al., 2024) to analyze RNA-seq datasets. In self-renewing embryonic stem cells (Fig. 6F), this analysis revealed that 1,058 splicing events in 949 genes were potentially altered in *Usb1*-null cells (Fig. 6H). Interestingly, only 5 out of these 949 genes are catalogued under “regulating pluripotency of stem cells” in KEGG pathway (mmu04550). This observation could explain why there is little functional impact on self-renewal in the absence of *Usb1*. In differentiating embryonic stem cells, 711 splicing events in 610 genes were potentially altered in *Usb1*-null cells at day 1 (Fig. 6I). Out of these 610 genes, 90 were overlapped with RA-responsive genes (RA treatment for 1 day, 2,105 genes) (Fig. 6J). At day 3 of RA treatment, 701 splicing events in 605 genes were potentially altered in *Usb1*-null cells (Fig. 6K). Out of these 605 genes, 171 were overlapped with RA-responsive genes (RA treatment for 3 days, 4,224 genes) (Fig. 6L). Given that RA-induced differentiation comprises multiple cell types, these observations indicate that these dysregulated splicing events may have some functional outcomes on specific cell types.”

The new data is reproduced here for your reference:

In Figure 6 legend, Line 1067-1086:

(H) Pie chart depicting the distribution of four alternative splicing types dysregulated in *Usb1*-null mESCs, compared to WT mESCs. In total, 1,058 events in 949 genes were detected by rMATS. Numbers in parentheses denote the dysregulated splicing events in *Usb1*-null mESCs. KEGG pathway “regulating pluripotency of stem cells” (mmu04550) consists of 140 genes. AE, alternative exon usage. IR, intron retention. A5SS, alternative 5' splice site. A3SS, alternative 3' splice site.

(I) Pie chart depicting the distribution of four alternative splicing types dysregulated in *Usb1*-null mESCs vs. WT mESCs upon retinoic acid (RA)-induced differentiation for 1 day. In total, 711 events in 610 genes were detected by rMATS. Numbers in parentheses denote the dysregulated splicing events in *Usb1*-null cells.

(J) Venn diagram depicting the overlap between differentially expressed genes (n=2,105) and alternatively spliced genes (n=610) in *Usb1*-null cells upon retinoic acid (RA)-induced differentiation for 1 day.

(K) Pie chart depicting the distribution of four alternative splicing types dysregulated in *Usb1*-null mESCs vs. WT mESCs upon retinoic acid (RA)-induced differentiation for 3 days. In total, 701 events in 605 genes were detected by rMATS. Numbers in parentheses denote the dysregulated splicing events in *Usb1*-null cells.

(L) Venn diagram depicting the overlap between differentially expressed genes (n=4,224) and alternatively spliced genes (n=605) in *Usb1*-null cells upon retinoic acid (RA)-induced differentiation for 3 days.”

Page 20, lines 420-429: The authors should confirm the length and composition of the 3'-tail of U6 snRNAs by cloning the U6 snRNA from *USB1*-null tissues and embryos.

Following your suggestion, we carried out 3' RACE experiment to determine the composition of the 3'-tail of U6 snRNA by cloning. The new data is presented in the revised manuscript (Appendix Figure S8), and reproduced here for your reference:

We added additional text in the revised manuscript as follows:

In Results, Line 424-426:

“Furthermore, we carried out 3' RACE experiment to analyze the composition of U6 snRNA 3'-end in *Usb1*-null cells. Consistent with RNA blot analysis (Fig. 6B, 7D, 7J and Appendix Fig. S7), we found that most U6 snRNA has 6 Us in *Usb1*-null cells (Appendix Fig. S8).”

In Appendix Figure S8:

“Appendix Figure S8. The length distribution of the 3'-end oligo(U) tail of U6 snRNA in *Usb1*-null neural stem cells.

3' RACE analysis of U6 oligo(U) tails in wild-type (*Usb1^{ff}*) and *Usb1* knockout (*Usb1^{ff};Emx1-Cre*) neural stem cells (n=3). The data represents mean ± SEM. P-

values, unpaired Student's t test."

Several references are not properly cited in the Discussion section. For example, on page 25, lines 473, 481, and 491, the relevant literature should be properly cited. Line 491 should cite the following paper on the recent complex structure of TUT1 and U6 snRNA:

Seisuke Yamashita, Kozo Tomita. Cryo-EM structure of human TUT1:U6 snRNA complex. *Nucleic Acids Research*, Volume 53, Issue 2, 27 January 2025, gkae1314, <https://doi.org/10.1093/nar/gkae1314>

Thank you for pointing this out. This new paper has been cited in the revised manuscript (Line 526).

Line 97: "day (E) 3.5" should be "day E3.5."

The text has been corrected in the revised manuscript (Line 103). Thank you.

Line 462: "even although" should be "even though."

The text has been corrected in the revised manuscript (Line 496). Thank you.

Line 778: "TUT1 catalyzes the post-transcriptional addition of an oligo(U) tail (depicted by 4 Us in red)" should be "TUT1 catalyzes the post-transcriptional addition of an oligo(U) tail (depicted by 2 Us in red)."

The text has been corrected in the revised manuscript (Line 859). Thank you.

Referee #2:

In this manuscript, Fang et. al develop and characterize new mouse models to study biogenesis of the core spliceosome component U6 snRNA. They find that TUT1, but not *Usb1*, is required during embryogenesis. This result, recapitulated through several mouse models and stem cell models, challenges the prevailing idea that these two enzymes are equivalently important for U6 function, but does explain the

human genetic data suggesting that loss of TUT1 results in stronger phenotypes. I cannot evaluate the mouse genetic work as this is outside my expertise. Overall, I found this to be an interesting manuscript that will be important to the field. However, I have the following key points, including an additional experiment and rewriting.

Thank you for your overall positive view of the manuscript. We are grateful for your suggestions, which we believe have improved the revised manuscript.

Major comments:

1. Transfected U6 snRNA with 4U's partially rescued the TUT1 knockout phenotype, while 6U's fully rescued TUT1 k.o. (Fig. 4F). The partial rescue by U6-4U could be because U6 snRNA is unstable in the absence of TUT1/3' end extension, and additional transfected U6 snRNA of the minimum appropriate length (4U, which is what is genomically encoded) can rescue the phenotype even though it's not the ideal length/ideally modified. The authors should test whether loss of TUT1 reduces U6 stability (expression level), ideally by probing for U6 on a northern blot. This experiment will also (likely) show that the U6 snRNA 3' tail is not extended in the absence of TUT1. This is an important aspect to their model and more important than showing a loss of interaction with the Lsm2-8 complex (Fig. 4G). In fact, it's a critical part of their model, as it's possible that TUT1 has a secondary effect that could lead to the drastic phenotype. The result of this experiment will influence their final model and claims (Fig. 7K). They claim that TUT1-extended RNA (and the 3'OH it leaves) is essential for splicing and viability, while a shortened cyclic phosphate tail installed by Usb1 is dispensable. The authors should note that Lsm2-8 can bind U tails with 3'OH, although the affinity is decreased. I think it's more likely that a 4U tail is too short for La/Lsm2-8 to bind and protect U6 from degradation, and in the absence of TUT1, you lose U6 expression. This experiment should be easily achievable, given that the authors probe for U6 in Fig. 6B.

Following your advice, we performed northern blot analysis of U6 snRNA to examine whether its expression level is altered in *Tut1*-null neural stem cells. Timepoint analyses demonstrated that U6 snRNA level start to decrease 2 days after *Tut1* ablation. This result indicates that *Tut1* may be required to maintain the steady level of U6 snRNA. The new data is presented in the revised manuscript (Appendix Figure S4), and reproduced here for your reference:

In light of this new finding, we added additional text in the revised manuscript as follows:

In Results, Line 312-316:

“When *Tut1* was ablated in neural stem cells (2 days after LNP/Cre-mRNA transfection), the amount of U6 snRNA associated with Lsm4 or Lsm8 protein was decreased (Fig. 4G). Interestingly, the total U6 snRNA level was also decreased at this timepoint (Appendix Fig. S4), indicating that the 3'-end oligo(U) tail promotes the interaction between Lsm2-8 and U6 snRNA.”

In Appendix Figure S4:

“Appendix Figure S4. Northern blot analysis of U6 snRNA expression in neural stem cells after LNP/Cre-mRNA transfection.

Time course analysis of U6 snRNA expression. Total RNA was derived from *Tut1^{fl/fl}* neural stem cells with or without LNP/Cre-mRNA transfection for 2, 3 or 4 days (n=3). U2 snRNA was used as a loading control.”

2. I find the alternative splicing analysis (Fig. 4H-K and Fig. 5) a little surprising and superficial. First, very few splicing events are altered. Is this a technical challenge with read depth, or are there really few changes? Second, why would RNA splicing be enriched in the GO analysis? Do splicing components have more introns, or less consensus splice sites, etc.? It would be helpful to improve this analysis for both Figure 4 and 5.

Thank you for pointing this out. We computed the sequencing statistics for Fig. 4H and Fig. 5 (new data, Appendix Figure S5 in the revised manuscript). These parameters are similar to those in published studies. For example, the percentage of full-length reads (% full length), calculated as the number of reads covering 80% of nucleotides for the transcript, was 19.6-34.3% for human blood samples in published

studies (Tang et al., 2020). In our study, this parameter was around 80% for cells and 55% for brain tissues. As pointed out by previous studies (Tang et al., 2020), Nanopore long-read sequencing usually detects much less altered events than short-read sequencing methodology. This is potentially because Nanopore long-read sequencing is not suitable to detect subtle splicing alterations, and/or because more stringent filtering criteria are applied in an effort to determine alternative splicing more accurately. In addition, our preliminary analyses indicate that the number of introns as well as the splicing sites of differentially spliced transcripts do not bear distinct features. Further investigation will be needed to address why splicing of these genes are affected.

The new data is presented in the revised manuscript (Appendix Figure S5), and reproduced here for your reference:

A

Sample	Data (Gb)	Total reads	% PASS	% full length	Median pass read length	Mean pass read length	No. of genes detected	% genes with AS
WT-1	12.022	26144584	33.24	88.11	623	900	13263	58.50%
WT-2	7.022	9851047	77.97	80.24	684	842	13549	56.40%
WT-3	7.343	11946731	74.21	82.99	621	737	13637	55.10%
Tut1^{fl/fl} ;Cre-1	7.365	10899850	76.67	81.26	661	803	13689	55.80%
Tut1^{fl/fl} ;Cre-2	10.536	16970857	72.30	77.43	632	762	13894	56.20%
Tut1^{fl/fl} ;Cre-3	6.736	8966639	80.15	79.49	700	871	13555	55.80%
p53^{fl/fl} ;Cre-1	8.24	12658680	76.61	81.63	651	772	13698	55.80%
p53^{fl/fl} ;Cre-1	8.401	11383350	79.96	82.17	691	855	13742	56.90%
p53^{fl/fl} ;Cre-1	7.597	10987475	79.62	84.05	666	801	13612	56.30%
p53^{fl/fl} ; Tut1^{fl/fl} ;Cre-1	7.393	9704333	79.20	79.15	707	893	13620	56.30%
p53^{fl/fl} ; Tut1^{fl/fl} ;Cre-1	6.934	9500305	80.38	80.95	686	841	13603	55.60%
p53^{fl/fl} ; Tut1^{fl/fl} ;Cre-1	6.311	9576746	76.89	82.41	648	778	13435	54.80%

B

Sample	Data (Gb)	Total reads	% PASS	% full length	Median pass read length	Mean pass read length	No. of genes detected	% genes with AS
Tut1^{fl/fl} -1	1.606	7436242	96.72	54.83	728	1068	12442	61.50%
Tut1^{fl/fl} -2	1.69	7886748	97.19	56.50	687	1011	12788	61.30%
Tut1^{fl/fl} ;Emx1-Cre-1	1.712	7903509	96.91	56.28	696	1039	13158	61.50%
Tut1^{fl/fl} ;Emx1-Cre-2	1.247	5850069	97.24	61.85	662	975	12522	60.30%

“Appendix Figure S5. Nanopore-sequencing statistics.

(A) Statistics for Fig. 4H.

(B) Statistics for Fig. 5.

The percentage of full-length (% full length) reads was calculated as the number of reads covering 80% of nucleotides for the transcript. The number of genes (No. of genes detected) was computed by counting the genes including all the isoforms. Genes with multiple isoforms identified were considered alternatively spliced (%

genes with AS).”

3. Why was RNA-seq instead of long read sequencing performed for the *Usb1* knockout? Given that few differentially expressed genes are detected in the TUT1 knockout (Fig. 5), this does not seem to be the appropriate assay to make the claim that *Usb1* levels do not affect RNA splicing.

Thank you for pointing this out. The purpose of RNA-seq experiment (Figure 6F and 6G) was to support that the overall cellular state is similar in wild-type and *Usb1*-null embryonic stem cells, indicated by the similarity of their transcriptomic profiles.

We agree with the Reviewer that lack of phenotype does not mean there is no defect in RNA splicing. To address this issue, we used rMATS to reanalyze the RNA-seq datasets to detect potential splicing defects in *Usb1*-null embryonic stem cells (new data, Figure 6H-6L in the revised manuscript). We added additional text in the revised manuscript as follows:

In Results, Line 395-408:

“To deduce differentially regulated splicing events in wild-type and *Usb1* knockout cells, we used rMATS (Wang et al., 2024) to analyze RNA-seq datasets. In self-renewing embryonic stem cells (Fig. 6F), this analysis revealed that 1,058 splicing events in 949 genes were potentially altered in *Usb1*-null cells (Fig. 6H). Interestingly, only 5 out of these 949 genes are catalogued under “regulating pluripotency of stem cells” in KEGG pathway (mmu04550). This observation could explain why there is little functional impact on self-renewal in the absence of *Usb1*. In differentiating embryonic stem cells, 711 splicing events in 610 genes were potentially altered in *Usb1*-null cells at day 1 (Fig. 6I). Out of these 610 genes, 90 were overlapped with RA-responsive genes (RA treatment for 1 day, 2,105 genes) (Fig. 6J). At day 3 of RA treatment, 701 splicing events in 605 genes were potentially altered in *Usb1*-null cells (Fig. 6K). Out of these 605 genes, 171 were overlapped with RA-responsive genes (RA treatment for 3 days, 4,224 genes) (Fig. 6L). Given that RA-induced differentiation comprises multiple cell types, these observations indicate that these dysregulated splicing events may have some functional outcomes on specific cell types.”

The new data is reproduced here for your reference:

In Figure 6 legend, Line 1067-1086:

(H) Pie chart depicting the distribution of four alternative splicing types dysregulated in *Usb1*-null mESCs, compared to WT mESCs. In total, 1,058 events in 949 genes were detected by rMATS. Numbers in parentheses denote the dysregulated splicing events in *Usb1*-null mESCs. KEGG pathway "regulating pluripotency of stem cells" (mmu04550) consists of 140 genes. AE, alternative exon usage. IR, intron retention. A5SS, alternative 5' splice site. A3SS, alternative 3' splice site.

(I) Pie chart depicting the distribution of four alternative splicing types dysregulated in *Usb1*-null mESCs vs. WT mESCs upon retinoic acid (RA)-induced differentiation for 1 day. In total, 711 events in 610 genes were detected by rMATS. Numbers in parentheses denote the dysregulated splicing events in *Usb1*-null cells.

(J) Venn diagram depicting the overlap between differentially expressed genes

(n=2,105) and alternatively spliced genes (n=610) in *Usb1*-null cells upon retinoic acid (RA)-induced differentiation for 1 day.

(K) Pie chart depicting the distribution of four alternative splicing types dysregulated in *Usb1*-null mESCs vs. WT mESCs upon retinoic acid (RA)-induced differentiation for 3 days. In total, 701 events in 605 genes were detected by rMATS. Numbers in parentheses denote the dysregulated splicing events in *Usb1*-null cells.

(L) Venn diagram depicting the overlap between differentially expressed genes (n=4,224) and alternatively spliced genes (n=605) in *Usb1*-null cells upon retinoic acid (RA)-induced differentiation for 3 days.”

4. The Materials and Methods lack sufficient details in many sections, particularly the molecular biology. How was U6 snRNA *in vitro* transcribed? The citation (#33) refers to mRNA (capped and modified). How was the 3' end generated - runoff transcription from a template (which usually gives heterogenous but 3'OH products) or an HDV ribozyme (which does not generate a 3'OH)? How was the Lsm4/8 IP performed? How were cell lines grown and maintained? How was the "RNA blot" in Fig. 6B performed, and which primers were used, as there is no mention of this in the supplemental table?

Additional details about the experimental conditions have been added to improve clarity in the revised manuscript as follows:

In Methods, Line 578-580:

“To generate recombinant U6 snRNA with 0, 2, 4 or 6 Us at the 3'-end, U6 snRNA cDNA with corresponding tails was cloned and used as the template for *in vitro* transcription by T7 polymerase.”

Line 600-611:

“Northern blot, 3' RACE analysis and RNA immunoprecipitation (RIP)

Northern blot analysis was carried out using Chemiluminescent Nucleic Acid Detection Module (Thermo Fisher), following the manufacturer's instructions. Briefly, 250 ng of total RNA were separated on 8% denaturing gel, and then transferred to a Nytran N⁺ membrane (GE). The membrane was crosslink by UV, blocked with NorthernMax™ (Invitrogen™), and then incubated with biotin labeled probes. 3' RACE experiments were performed as previously described (Shchepachev et al.,

2015), using total RNA derived from wild-type and *Usb1* knockout neural stem cells, respectively (n=3). Sixteen clones were sequenced for each genotype (96 clones in total). RIP procedure was carried out as previously described (Gagliardi and Matarazzo, 2016). Briefly, cell lysates were incubated with protein A/G beads and anti-Lsm4 or anti-Lsm8 antibody for 4 hours at 4°C, and the beads were washed four times. Associated RNA was extracted by TRIzol, reverse transcribed by M-MLV reverse transcriptase, and then amplified using U6 snRNA-specific primers.”

5. Many figure legends (i.e., Fig. 4B) are insufficient. For Fig. 4B/E, how much RNA was loaded? What kind of gel? What size are the products (a ladder would help)? This is a pattern throughout the paper.

We have added additional text in the revised manuscript as follows:

In Methods, Line 580-582:

“100 ng recombinant RNA was resolved by 8% denaturing gel, and stained by ethidium bromide to inspect the purity. The length and integrity of U6 snRNA and Cre-mRNA was routinely monitored by Agilent 2100 using RNA 6000 nano kit.”

Minor comments:

Line 38 typo - "Catalytical"

It has been changed to “catalytic” in the revised manuscript (Line 40). Thank you.

Line 293 and 295 - refers to Fig 4E to show efficiency of knockdown, but this is a gel of purified U6 snRNA.

It has been changed to “Fig. 4F” in the revised manuscript (Line 303 and Line 305). Thank you.

Referee #3:

The TUT1 gene (now renamed as TENT1) participates in U6 snRNA processing and maintenance by adding uridines to the 3' end of U6 snRNA. The terminal uridine in

U6 snRNA ends in a cyclic 2',3' phosphate, formed by the exonucleolytic activity of USB1, which helps stabilize U6 snRNA. Whether there are other substrates (other than U6ATAC snRNA) for these enzymes, and what they are, is not clear. This paper shows that deletion of TUT1 in mice results in a different phenotype than deletion of USB1. However they do show that *Usb1* is an essential gene, since the *Usb1* KO mice die after E12.5, although it is not essential in all cells (since they knocked it out in embryonic stem cells, and these were not affected). As far as I can tell this lethality in the animal of a *Usb1* KO has never been reported before, and the authors should certainly consider this a main point of their paper. *Usb1* has been reported to also remove A-tails from some adenylated RNAs and that activity should be mentioned, since that activity might be critical for it being an essential gene (Huynh & Parker, 2023; Jeong et al., 2023; Mroczek & Dziembowski, 2013; Mroczek et al., 2012).

References

- Huynh, T. N., & Parker, R. (2023). The PARN, TOE1, and USB1 RNA deadenylases and their roles in non-coding RNA regulation. *J Biol Chem*, 299(9), 105139. doi:10.1016/j.jbc.2023.105139
- Jeong, H. C., Shukla, S., Fok, W. C., Huynh, T. N., Batista, L. F. Z., & Parker, R. (2023). USB1 is a miRNA deadenylase that regulates hematopoietic development. *Science*, 379(6635), 901-907. doi:10.1126/science.abj8379
- Montemayor, E. J., Didychuk, A. L., Yake, A. D., Sidhu, G. K., Brow, D. A., & Butcher, S. E. (2018). Architecture of the U6 snRNP reveals specific recognition of 3'-end processed U6 snRNA. *Nat Commun*, 9(1), 1749. doi:10.1038/s41467-018-04145-4
- Mroczek, S., & Dziembowski, A. (2013). U6 RNA biogenesis and disease association. *Wiley. Interdiscip. Rev. RNA*. doi:10.1002/wrna.1181 [doi]
- Mroczek, S., Krwawicz, J., Kutner, J., Lazniewski, M., Kucinski, I., Ginalski, K., & Dziembowski, A. (2012). C16orf57, a gene mutated in poikiloderma with neutropenia, encodes a putative phosphodiesterase responsible for the U6 snRNA 3' end modification. *Genes Dev*, 26(17), 1911-1925. doi:gad.193169.112 [pii];10.1101/gad.193169.112 [doi]

We are grateful for your constructive feedback, which has helped to improve the quality of the manuscript. In the previous manuscript, we have cited these references. We also discussed the ability of USB1 to remove A-tails in the previous manuscript (Line: 494-502), and in the revised manuscript (Line: 534-542).

Specific comments:

1. The results of Fig.1-4 clearly that TUT1 is required for ES cell, early embryo development and development of the brain in mice. Somewhat surprisingly in Camk2a-cre conditional Tut1 KO mice, the mice survive and develop normally. They do not indicate whether they tested whether the removal of TUT1 was successful in the cerebellum of the fl/fl mice It seems like that control should have been done.

Thank you for pointing this out. In the revised manuscript, we used qPCR (as in Figure 4C) to determine the knockout efficiency in *Tut1^{fl/fl};Camk2a-Cre* mice (new data, Appendix Figure S2 in the revised manuscript). We have added additional text in the revised manuscript as follows:

In Results, Line 201:

“The efficiency of *Tut1* deletion was around 90% as determined by qPCR (Appendix Fig. S2).”

The new data is reproduced here for your reference:

2. The loss of TUT 1 in cultured neural stem cells in vitro can be rescued by injection of U6 snRNA in vitro if U6 snRNA with 6 uridines at the 3' end was introduced into cells. Somewhat surprisingly the levels of U6 snRNA bound to the LSm proteins was only reduced 50% compared to the unfloxed cells, which I would not have expected to affect splicing that much. Also in the last sentence on page 17 (l. 359). "These results demonstrate that TUT1 regulates mRNA splicing during embryogenesis". All this shows is that (as expected) Tut1 is necessary for splicing during embryogenesis.

Thank you for pointing this out. We examined U6 snRNA level in neural stem cells by northern blot, after LNP/Cre-mRNA transfection for 2, 3 or 4 days (new data, Appendix Figure S4 in the revised manuscript). We found that U6 snRNA level was decreased at day 2 after transfection. In light of this new finding, we added additional text in the revised manuscript as follows:

In Results, Line 312-316:

“When *Tut1* was ablated in neural stem cells (2 days after LNP/Cre-mRNA transfection), the amount of U6 snRNA associated with Lsm4 or Lsm8 protein was decreased (Fig. 4G). Interestingly, the total U6 snRNA level was also decreased at this timepoint (Appendix Fig. S4), indicating that the 3'-end oligo(U) tail promotes the interaction between Lsm2-8 and U6 snRNA.”

The new data is reproduced here for your reference:

In Appendix Figure S4:

“Appendix Figure S4. Northern blot analysis of U6 snRNA expression in neural stem cells after LNP/Cre-mRNA transfection.

Time course analysis of U6 snRNA expression. Total RNA was derived from *Tut1^{ff}* neural stem cells with or without LNP/Cre-mRNA transfection for 2, 3 or 4 days (n=3). U2 snRNA was used as a loading control.”

Following your advice, the text has been changed to “These results demonstrate that *TUT1* is required for mRNA splicing during embryogenesis” in the revised manuscript (Line 372-373).

3. For the alternative splicing data, they need to indicate how many total mRNAs they sequenced and how many shows the splicing defects. Since they did long-read sequencing (presumably from the polyA tail; they don't mention that) they know how many mRNAs were sequenced from each sample and how many of those have splicing defects. Since the non-productive intron retention events should result in NMD and rapid loss of the transcripts it is likely that they are much more abundant than they observed and they should mention that possibility. It makes sense that inefficient splicing may result in more intron-retention events as they observed.

Thank you for pointing this out. Poly(A)-enriched transcripts were sequenced. We computed the sequencing statistics for Figure 4H and Figure 5 (new data,

Appendix Figure S5 in the revised manuscript). We added additional text in the revised manuscript as follows:

In Results, Line 325-326:

“More than 13,000 genes were detected for each group (Appendix Fig. S5A).”

Line 344-345:

“More than 12,000 genes were detected for each group (Appendix Fig. S5B).”

Line 366-370:

“This observation suggests an underestimation of intron retention events, because unproductive splicing is expected to trigger rapid degradation of transcripts via quality control mechanisms such as nonsense-mediated mRNA decay (Lykke-Andersen and Jensen, 2015).”

The new data is presented in the revised manuscript (Appendix Figure S5), and reproduced here for your reference:

A

Sample	Data (Gb)	Total reads	% PASS	% full length	Median pass read length	Mean pass read length	No. of genes detected	% genes with AS
WT-1	12.022	26144584	33.24	88.11	623	900	13263	58.50%
WT-2	7.022	9851047	77.97	80.24	684	842	13549	56.40%
WT-3	7.343	11946731	74.21	82.99	621	737	13637	55.10%
Tut1^{fl/fl} ;Cre-1	7.365	10899850	76.67	81.26	661	803	13689	55.80%
Tut1^{fl/fl} ;Cre-2	10.536	16970857	72.30	77.43	632	762	13894	56.20%
Tut1^{fl/fl} ;Cre-3	6.736	8966639	80.15	79.49	700	871	13555	55.80%
p53^{fl/fl} ;Cre-1	8.24	12658680	76.61	81.63	651	772	13698	55.80%
p53^{fl/fl} ;Cre-1	8.401	11383350	79.96	82.17	691	855	13742	56.90%
p53^{fl/fl} ;Cre-1	7.597	10987475	79.62	84.05	666	801	13612	56.30%
p53^{fl/fl} ; Tut1^{fl/fl} ;Cre-1	7.393	9704333	79.20	79.15	707	893	13620	56.30%
p53^{fl/fl} ; Tut1^{fl/fl} ;Cre-1	6.934	9500305	80.38	80.95	686	841	13603	55.60%
p53^{fl/fl} ; Tut1^{fl/fl} ;Cre-1	6.311	9576746	76.89	82.41	648	778	13435	54.80%

B

Sample	Data (Gb)	Total reads	% PASS	% full length	Median pass read length	Mean pass read length	No. of genes detected	% genes with AS
Tut1^{fl/fl} -1	1.606	7436242	96.72	54.83	728	1068	12442	61.50%
Tut1^{fl/fl} -2	1.69	7886748	97.19	56.50	687	1011	12788	61.30%
Tut1^{fl/fl} ;Emx1-Cre-1	1.712	7903509	96.91	56.28	696	1039	13158	61.50%
Tut1^{fl/fl} ;Emx1-Cre-2	1.247	5850069	97.24	61.85	662	975	12522	60.30%

In Appendix Figure S5:

“Appendix Figure S5. Nanopore-sequencing statistics.

(A) Statistics for Fig. 4H.

(B) Statistics for Fig. 5.

The percentage of full-length (% full length) reads was calculated as the number of reads covering 80% of nucleotides for the transcript. The number of genes (No. of genes detected) was computed by counting the genes including all the isoforms. Genes with multiple isoforms identified were considered alternatively spliced (% genes with AS).”

4. On page 20 (and particularly in the abstract, as well as throughout the manuscript), they should emphasize that *Usb1* is also an essential gene, with embryos dying after E10.5 and before E12.5 (Fig.7F). They should likely put this in the title of the paper also.

Thank you for pointing this out. We agree with the Reviewer that *Usb1* is also an essential gene at midgestation, although the underlying cause of lethality is unclear. Because this is the first *in vivo* study of *Tut1* and *Usb1*, we intended to convey the idea that *Tut1* and *Usb1* have distinct physiological functions. This notion is supported by the fact that mice with *Usb1* ablation by *Emx1-Cre* do not exhibit any overt phenotypes (Figure 7A-7D and Appendix Figure S6), in sharp contrast to *Tut1* ablation (Figure 2 and Figure 3). In addition, *Tut1*-null embryos fail to develop at the earliest stage. In contrast, *Usb1*-null embryos are morphologically similar to wild-type embryos around E8.5 (Figure 7G), and contain most intra-embryonic cell types (Figure 7H and 7I).

In the revised manuscript, we made several changes to improve clarity as follows:

In Abstract, Line: 26-29

“We propose that TUT1-catalyzed oligo(U) tail is essential for splicing and cell proliferation. Further modification of this oligo(U) tail by USB1 is ubiquitous, but only functionally required in specific cell types.”

In Discussion, Line: 499-503

“Of note, *Usb1*-null embryos can develop till mid-embryogenesis, but then degenerate. Our preliminary analyses indicate that extra-embryonic defects may underlie this lethality. Nevertheless, we cannot rule out the possibility that *Usb1* may be required in certain intra-embryonic lineages. Further studies will be needed to

address these issues.”

5. The data in Fig. 6J does not make sense. They show that in wild-type cells and in the heterozygotes (lane 1 and 2), U6 snRNA is shorter than in the *Usb1* KO cells. This suggests that the cyclin phosphate may affect the mobility of U6 snRNA and/or the loss of *Usb1* results in TUT1 being able to produce U6 snRNA with longer U tails (the length of the U tail will be determined by both the elongation of U6 snRNA by TUT1, and the removal of the U tail by exonucleases). The only way to accurately determine the length of the U6 tail is by sequencing the 3' end of the molecule directly. A critical role for the phosphate at the end of U6 snRNA in splicing has been suggested by the work of Brow and coworkers (Montemayor et al., 2018)

We suspect that the Reviewer meant Figure 7J. We would like to point out that previous biochemical, structural and cell-based studies have established that *Usb1* functions as an exonuclease to shorten the 3'-tail of U6 snRNA. One of these studies (Mroczek et al., 2012) was also mentioned by the Reviewer. In these published studies (Hilcenko et al., 2013; Mroczek et al., 2012; Shchepachev et al., 2012; Shchepachev et al., 2015), U6 snRNA was cloned and sequenced (as the Reviewer suggested) to demonstrate that the 3'-tail of most U6 snRNA contains extra 2 Us when *Usb1* is absent. To confirm, we carried out 3' RACE experiment to determine the composition of the 3'-tail of U6 snRNA by cloning. The new data is presented in the revised manuscript (Appendix Figure S8), and reproduced here for your reference:

We added additional text in the revised manuscript as follows:

In Results, Line 424-426:

“Furthermore, we carried out 3' RACE experiment to analyze the composition of U6 snRNA 3'-end in *Usb1*-null cells. Consistent with RNA blot analysis (Fig. 6B, 7D, 7J and Appendix Fig. S7), we found that most U6 snRNA has 6 Us in *Usb1*-null cells (Appendix Fig. S8).”

In Appendix Figure S8:

“Appendix Figure S8. The length distribution of the 3'-end oligo(U) tail of U6 snRNA in *Usb1*-null neural stem cells.

3' RACE analysis of U6 oligo(U) tails in wild-type (*Usb1^{ff}*) and *Usb1* knockout (*Usb1^{ff}*;Emx1-Cre) neural stem cells (n=3). The data represents mean ± SEM. P-values, unpaired Student's t test.”

In the previous manuscript, we omitted the fact that we used recombinant U6 snRNA with 0, 2, 4 and 6 Us as size markers (e.g., Figure 7J). Upon reflection, this might have caused some confusion. We agree with the Reviewer that 2',3'-cyclic phosphate modification may also affect the mobility of U6 snRNA on the gel. However, in the condition we used, this effect is minimal. We used *in vitro*-transcribed recombinant U6 snRNA containing either 0, 2, 4 or 6 Us at the 3'-end as the size marker on the gel (Figures 6B, 7D, and 7J). These recombinant U6 snRNAs contains a 3'-OH but not 2',3'-cyclic phosphate modification. Recombinant U6 snRNA containing either 0, 2, 4 or 6 Us can be well separated on the gel (Figure 4E). U6 snRNA derived from *Usb1*-null cells (Figure 6B) and tissues (Figure 7D and 7J) has the same mobility as the recombinant U6 snRNA with 2 Us on the gel. These results are also consistent with 3' RACE experiment that most U6 snRNA has 6 Us in *Usb1*-null cells (new data, Appendix S8 in the revised manuscript).

The Reviewer also commented on the study by Montemayor et al., 2018. Upon reflection, we agree with the Reviewer that our data only show that *Usb1* is functionally dispensable in many embryonic cell types, but do not speak to the role of *Usb1* in splicing. In the revised manuscript, we have clarified that lack of phenotype in *Usb1*-null cells/tissues does not mean there is no defect in RNA splicing. We reanalyzed our RNA-seq datasets (Figure 6F and 6G), and used rMATS to deduce differentially regulated splicing events in wild-type and *Usb1* knockout cells (new data, Figure 6H-6L in the revised manuscript). We added additional text in the revised manuscript as follows:

In Results, Line 395-408:

“To deduce differentially regulated splicing events in wild-type and *Usb1* knockout cells, we used rMATS (Wang et al., 2024) to analyze RNA-seq datasets. In self-

renewing embryonic stem cells (Fig. 6F), this analysis revealed that 1,058 splicing events in 949 genes were potentially altered in *Usb1*-null cells (Fig. 6H). Interestingly, only 5 out of these 949 genes are catalogued under “regulating pluripotency of stem cells” in KEGG pathway (mmu04550). This observation could explain why there is little functional impact on self-renewal in the absence of *Usb1*. In differentiating embryonic stem cells, 711 splicing events in 610 genes were potentially altered in *Usb1*-null cells at day 1 (Fig. 6I). Out of these 610 genes, 90 were overlapped with RA-responsive genes (RA treatment for 1 day, 2,105 genes) (Fig. 6J). At day 3 of RA treatment, 701 splicing events in 605 genes were potentially altered in *Usb1*-null cells (Fig. 6K). Out of these 605 genes, 171 were overlapped with RA-responsive genes (RA treatment for 3 days, 4,224 genes) (Fig. 6L). Given that RA-induced differentiation comprises multiple cell types, these observations indicate that these dysregulated splicing events may have some functional outcomes on specific cell types.”

The new data is reproduced here for your reference:

H

I

J

K

L

In Figure 6 legend, Line 1067-1086:

(H) Pie chart depicting the distribution of four alternative splicing types dysregulated in *Usb1*-null mESCs, compared to WT mESCs. In total, 1,058 events in 949 genes were detected by rMATS. Numbers in parentheses denote the dysregulated splicing events in *Usb1*-null mESCs. KEGG pathway “regulating pluripotency of stem cells” (mmu04550) consists of 140 genes. AE, alternative exon usage. IR, intron retention. A5SS, alternative 5' splice site. A3SS, alternative 3' splice site.

(I) Pie chart depicting the distribution of four alternative splicing types dysregulated in *Usb1*-null mESCs vs. WT mESCs upon retinoic acid (RA)-induced differentiation for 1 day. In total, 711 events in 610 genes were detected by rMATS. Numbers in parentheses denote the dysregulated splicing events in *Usb1*-null cells.

(J) Venn diagram depicting the overlap between differentially expressed genes (n=2,105) and alternatively spliced genes (n=610) in *Usb1*-null cells upon retinoic acid (RA)-induced differentiation for 1 day.

(K) Pie chart depicting the distribution of four alternative splicing types dysregulated in *Usb1*-null mESCs vs. WT mESCs upon retinoic acid (RA)-induced differentiation for 3 days. In total, 701 events in 605 genes were detected by rMATS. Numbers in parentheses denote the dysregulated splicing events in *Usb1*-null cells.

(L) Venn diagram depicting the overlap between differentially expressed genes (n=4,224) and alternatively spliced genes (n=605) in *Usb1*-null cells upon retinoic acid (RA)-induced differentiation for 3 days.”

References:

Gagliardi, M., and Matarazzo, M. R. (2016). RIP: RNA Immunoprecipitation. *Methods in molecular biology* 1480, 73-86.

Hilcenko, C., Simpson, P. J., Finch, A. J., Bowler, F. R., Churcher, M. J., Jin, L., Packman, L. C., Shlien, A., Campbell, P., Kirwan, M., *et al.* (2013). Aberrant 3' oligoadenylation of spliceosomal U6 small nuclear RNA in poikiloderma with neutropenia. *Blood* 121, 1028-1038.

Lykke-Andersen, S., and Jensen, T. H. (2015). Nonsense-mediated mRNA decay: an intricate machinery that shapes transcriptomes. *Nature reviews Molecular cell biology* 16, 665-677.

Mroczek, S., Krwawicz, J., Kutner, J., Lazniewski, M., Kucinski, I., Ginalski, K., and Dziembowski, A. (2012). C16orf57, a gene mutated in poikiloderma with neutropenia, encodes a putative phosphodiesterase responsible for the U6 snRNA 3' end

modification. *Genes & development* 26, 1911-1925.

Shchepachev, V., Wischnewski, H., Missiaglia, E., Sonesson, C., and Azzalin, C. M. (2012). Mpn1, mutated in poikiloderma with neutropenia protein 1, is a conserved 3'-to-5' RNA exonuclease processing U6 small nuclear RNA. *Cell reports* 2, 855-865.

Shchepachev, V., Wischnewski, H., Sonesson, C., Arnold, A. W., and Azzalin, C. M. (2015). Human Mpn1 promotes post-transcriptional processing and stability of U6atac. *FEBS Lett* 589, 2417-2423.

Tang, A. D., Soulette, C. M., van Baren, M. J., Hart, K., Hrabeta-Robinson, E., Wu, C. J., and Brooks, A. N. (2020). Full-length transcript characterization of SF3B1 mutation in chronic lymphocytic leukemia reveals downregulation of retained introns. *Nature communications* 11, 1438.

Dear Prof. Li,

Thank you for the submission of your revised manuscript to our editorial offices. I have now received the reports from the three referees that I asked to re-evaluate the study, you will find below.

As you will see, the referees now support publication of your study in EMBO reports. However, Referees #2 and #3 have final comments and suggestions to improve the manuscript, mainly requesting text changes, I ask you to address in a final revised manuscript. Please also include a more detailed methods part, as indicated by referee #2. Please also provide a detailed final p-b-p-response regarding the remaining referee points and the editorial requests below.

Editorial requests:

- The Expanded View format, which will be displayed in the main HTML of the paper in a collapsible format, has replaced the Supplementary information. Thus, please combine the figures presently shown in the Appendix to have not more than 5 EV figures and upload these as separate EV figures. Please follow the nomenclature Figure EV1, Figure EV2 etc. and put the figure legend for these in the main manuscript document file in a section called Expanded View Figure Legends after the main Figure Legends section (see below). Please update all callouts and finally remove the Appendix file.

- Please add up to five keywords to the manuscript and order the manuscript sections like this, using only these names: Title page - Abstract - Keywords - Introduction - Results - Discussion - Methods - Data availability section - Acknowledgements - Disclosure and Competing Interests Statement - References - Figure legends - Expanded View Figure legends

- We now use CRediT to specify the contributions of each author in the journal submission system. CRediT replaces the author contribution section. Please use the free text box to provide more detailed descriptions and do NOT provide your final manuscript text file with an author contributions section. See also our guide to authors (section 'Author contributions'): <https://link.springer.com/journal/44319/submission-guidelines#cms-Revised-submissions>

- Please check again that the number "n" for how many independent experiments were performed, their nature (biological versus technical replicates), the bars and error bars (e.g. SEM, SD) and the test used to calculate p-values is indicated in the respective figure legends (main and EV figures). Please also check that all the p-values are explained in the legend, and that these fit to those shown in the figure. Please provide statistical testing where applicable. Please avoid the phrase 'independent experiment' but clearly state if these were biological or technical replicates. Please also indicate (e.g. with n.s.) if testing was performed, but the differences are not significant. In case n=2, please show the data as separate datapoints without error bars and statistics. See also:

<https://link.springer.com/journal/44319/submission-guidelines#cms-Figure-and-data-presentation>

If $n < 5$, please show single datapoints for diagrams. Moreover:

- Please note that the exact p values are not provided in the legends of figures 1D, 4C, 5G
- Please indicate the statistical test used for data analysis in the legend of figure 4I
- Please note that information related to n is missing in the legends of figures 1D, 4C, G, K, 5G
- Please note that the dashed lines are not defined in the legend of figure 2M-O. This needs to be rectified.

- Please make sure that all microscopic images have scale bars of similar style and thickness, using clearly visible black or white bars (depending on the background). Please place these in the lower right corner of the images themselves. Please do not write on or near the bars in the image but define the size in the respective figure legend.

- The correct nomenclature for movie file is "Movie EV1". Please change this in all places. Please also provide a legend for the movie. This needs to be provided as separate text file and then uploaded, ZIPped together with the movie.

- The data availability section (DAS) is restricted to information about large datasets generated in the study that have been deposited externally. Please provide here only access information for the sequencing data and direct links. Please remove now all referee tokens and make sure that the datasets are public latest upon online publication of the manuscript.

- There are presently two files uploaded as reagents & tools table. Please provide one final file using the .doc template (see below) including all primer and sequence information. Do not provide the reagents and tools table as excel file. More information on how to adhere to this format as well as downloadable templates (.doc) for the Reagents and Tools Table can be found in our author guidelines (section 'Structured Methods'):

<https://link.springer.com/journal/44319/submission-guidelines#cms-Manuscript-organisation-and-formatting>

Please also remove the instructions text from the final reagents and tools table.

- Thanks for providing part of the source data. It seems, though, that for many panels appropriate source data was not provided. Please go again through the source data checklist, update the source data checklist accordingly, and provide source data for each panel (except those that show data from the deposited large datasets). Please provide the numerical source data as excel files. Please also provide microscopic source data. Moreover, please upload the source data as one folder per main figure, grouping together the separate files for each panel and ZIPed together, and one folder with all the source data for the EV figures, grouping together all files for each EV figure and ZIPed together.

In addition, I would need from you uploaded separately:

I look forward to seeing the further revised version of your manuscript when it is ready. Please let me know if you have questions regarding the revision.

Best,

Referee #1:

As far as this reviewer is concerned, the authors have adequately revised the manuscript with additional experiments and data analysis. This revision is satisfactory for publication in EMBO Reports.

Referee #2:

This remains an important paper that will be of great interest to the field. I appreciate the effort that the authors put in to address the reviewer's comments. I have minor remaining concerns and do not think additional experiments should be conducted prior to publication.

I am glad that the authors looked at U6 levels upon loss of Tut1, as I think this explains the phenotypes well (extension by Tut1 is required for U6 stability, but Usb1 is not essential because Lsm2-8 can still bind Tut1-extended U6). However, I would suggest some rephrasing, as the conclusion that "Tut1-catalyzed oligo(U) tail at the 3' end of U6 snRNA is required for mRNA splicing to maintain neural stem cells" (lines 337-338) is likely an indirect effect. The authors' new data (Appendix Fig. S4) shows that U6 RNA levels are decreased in the absence of Tut1 modification, likely because of loss of protection from interaction with Lsm2-8. Thus, the Tut1 splicing phenotype likely arises from low U6 levels. The major conclusion should be that Tut1 modification is important for U6 snRNA stability, and that defects in splicing arising from low U6 levels drives neural stem cell death during embryogenesis (or something to that effect). Similarly, concluding that Tut1 "regulates mRNA splicing" ignores the effect of U6 levels.

Line 91 - "functionally dispensable in most cell types" should clarify that USB1 is functionally dispensable until mid-embryogenesis. As written, it implies that USB1 is not an essential gene.

My primary remaining concern is the lack of sufficient detail for the methods section, particularly for the molecular biology. For instance, the technical details for in vitro transcription matter (Lines 574-576: What was the cDNA cloned into? How was the template generated? Buffer conditions?) as they affect the likelihood of nontemplated addition by T7 polymerase. How was the RNA purified prior to transfection?

Referee #3:

The authors have carried out additional experiments to address previous comments of the reviewers. They clearly show that TUT1 is essential for survival in all cell types they tested, and that the major defect is in RNA splicing, as expected from

reduction of functional U6 snRNP. In contrast Usb1 (which catalyzes the formation of the 2'3' cyclic phosphate on the 3' end of the oligo U-tail) is not essential. Moreover all the U6 snRNA (Fig. 7J) present in the Usb1 KO cells have extended U-tails (about 6 nts). These tails will bind Lsm2-8 and hence promote splicing. Note that in wild-type cells U6 snRNA (which has a 5 nt U tail with a 2'3 phosphate), has an identical mobility with the 4U in vitro transcribed RNA, and no data is shown for the position of U6 with no U tails.

Why is TUT1 lethal and Usb1 has minor phenotypes? To me it seems likely that in the Usb1 mutant cells, there is removal of the U-tail put on by TUT1 by a 3' to 5' exonuclease (since the cyclic phosphate will block degradation, and continuous regeneration of the U-tail by Tut1, to maintain sufficient levels of active U6 snRNA bound to Lsm2-8. The role of Usb1 is to stabilize the U-tail (blocking the exonuclease) and promoting the binding of Lsm 2-8 to the U-tail, which presumably would also slow exonuclease activity. In the absence of Usb1, the tails are longer, but they are almost certainly being continuously removed and then extended again. Likely this results in a lower concentration of functional U6 snRNP, and the milder phenotypes observed.

Note that when one is studying rare human inherited "diseases", it is very common that mutations in proteins that are constitutively expressed result in neurological defects, since the brain and nervous system are the most complex system and govern the function of all other tissues. There is no reason to emphasize the effects are on stem cells. They show the effects are on all cultured cells. As one example, the same is true for mutations in many of the subunits of the integrator complex, involved in RNA pol II pause/release in every cell and thousands of genes. Mutations in humans that have been reported all affect the brain.

SPECIFIC COMMENTS:

1. In the abstract they should indicate that the oligo U-tail on U6 is bound to Lsm 2-8, which is also an important contributor to the function of U6 snRNP.: The 3' tail actually consists of the oligo(U) tail, the 2'3' phosphate and the Lsm2-8 complex which together provide the functions of the tail.
2. Similarly at the bottom of page 4, they should make it clear that the modified 3' end is involved directly in U6 snRNA binding to Lsm 2-8; Note that it is not essential for binding, but enhances the binding affinity by a factor of 5-10, and this reference should be cited:Montemayor, E. J. et al. Molecular basis for the distinct cellular functions of the Lsm1-7 and Lsm2-8 complexes. RNA 26, 1400-1413 (2020).
3. pg. 6, l. 105. The viability of the 3.5 day embryo could be due to remaining maternal supply of TUT1 protein or U6 snRNP. Indeed their results show that KO of TUT1 is lethal in cultured ES cells as well. The shRNA experiments support this possibility, since in these cells a great reduction of TUT1 is lethal, although there is clearly a small amount of protein present by their Western blot.
3. The p53 results show that the cell damage is detected by p53 which then induces apoptosis. I presume that in the absence of p53 the cells stop growing and ultimately die.
4. In line 351 they say "Thus, these splicing events are directly regulated by Tut1 (Fig. 4H)". This statement and any other statements using the word "regulate" as one of TUT1's function are incorrect. TUT1 is required for formation of active U6 snRNP. If there are low levels of U6 snRNP, then many splicing events, particularly those that may be less efficient will not occur. This is not "regulation."
5. They end the results section with the sentence "These results challenge the current model of post-transcriptional maturation of U6 snRNA, and position Tut1, but not Usb1, as an essential regulator of RNA splicing during mammalian embryonic development"
The sentence should be deleted. Their results don't challenge the current model. TUT1 is not an essential regulator of splicing. It is essential for the extension of U6 snRNA to allow Lsm2-8 binding and mature U6 snRNP, which is part of the current model, but it certainly doesn't regulate anything.
6. In the discussion l.467-480, they provide strong evidence that TUT1 is essential in stem cells for splicing, but it also seems to be essential for all cultured cells. They cannot conclude there is anything special about TUT1 in stem cells. The proper conclusion is lines 480-483 "TUT1, but not USB1, is a ubiquitous, essential U6 snRNA 3'-end maturation or repair factor required for cell proliferation". Their study is the first to do this, and agrees with the DepMap public data.
7. The authors are correct in that "the Mellman et al., paper is incorrect", and all other papers about STAR-PAP, all of which came from one group, are incorrect (l. 519).

We are truly grateful to the anonymous Reviewers for their insightful suggestions and constructive feedback, which have helped improve the quality of this manuscript.

Referee #1:

As far as this reviewer is concerned, the authors have adequately revised the manuscript with additional experiments and data analysis. This revision is satisfactory for publication in EMBO Reports.

Thank you for your effort and time invested in our manuscript.

Referee #2:

This remains an important paper that will be of great interest to the field. I appreciate the effort that the authors put in to address the reviewer's comments. I have minor remaining concerns and do not think additional experiments should be conducted prior to publication.

Thank you for your effort and time invested in our manuscript.

I am glad that the authors looked at U6 levels upon loss of Tut1, as I think this explains the phenotypes well (extension by Tut1 is required for U6 stability, but Ubs1 is not essential because Lsm2-8 can still bind Tut1-extended U6). However, I would suggest some rephrasing, as the conclusion that "Tut1-catalyzed oligo(U) tail at the 3' end of U6 snRNA is required for mRNA splicing to maintain neural stem cells" (lines 337-338) is likely an indirect effect. The authors' new data (Appendix Fig. S4) shows that U6 RNA levels are decreased in the absence of Tut1 modification, likely because of loss of protection from interaction with Lsm2-8. Thus, the Tut1 splicing phenotype likely arises from low U6 levels. The major conclusion should be that Tut1 modification is important for U6 snRNA stability, and that defects in splicing arising from low U6 levels drives neural stem cell death during embryogenesis (or something to that effect). Similarly, concluding that Tut1 "regulates mRNA splicing" ignores the effect of U6 levels.

Following your suggestion, the text has been modified in the revised manuscript as follows:

In Results, Line 343-345:

“Taken together, we conclude that *Tut1*-catalyzed oligo(U) tail at the 3'-end of U6 snRNA is important for U6 snRNA stability. Loss of *Tut1* lowers U6 snRNA level, leading to defects in splicing and subsequent neural stem cell death during embryogenesis.”

Line 91 - "functionally dispensable in most cell types" should clarify that USB1 is functionally dispensable until mid-embryogenesis. As written, it implies that USB1 is not an essential gene.

The text has been modified in the revised manuscript as follows:

In Introduction, Line 93-94:

“functionally dispensable in most embryonic cell types until mid-embryogenesis”.

My primary remaining concern is the lack of sufficient detail for the methods section, particularly for the molecular biology. For instance, the technical details for *in vitro* transcription matter (Lines 574-576: What was the cDNA cloned into? How was the template generated? Buffer conditions?) as they affect the likelihood of nontemplated addition by T7 polymerase. How was the RNA purified prior to transfection?

Following your suggestion, experimental details have been added in the revised manuscript as follows:

In Methods, Line 582-590:

“U6 snRNA cDNA was cloned using total RNA derived from mouse primary neural stem cells, and its sequence is identical to NR_003027.2 (without the terminal 5 Ts). To generate recombinant U6 snRNA with 0, 2, 4 or 6 Us at the 3'-end, U6 snRNA cDNA with corresponding tails was sub-cloned into the pUC57 vector using primers listed in Table EV6 (forward primer: U6-F; reverse primers: U6-0U-R, U6-2U-R, U6-4U-R and U6-6U-R), linearized by Esp3I (Thermo Fisher), and then used as the template for *in vitro* transcription by T7 polymerase (Novoprotein, China) following the manufacturer's instruction. Of note, Esp3I is a Type IIS enzyme generating a precise 3'-end without any vector-derived nucleotides. The recombinant U6 snRNA was purified by RNeasy Mini kit (QIAGEN).”

Referee #3:

The authors have carried out additional experiments to address previous comments of the reviewers. They clearly show that TUT1 is essential for survival in all cell types they tested, and that the major defect is in RNA splicing, as expected from reduction of functional U6 snRNP. In contrast *Usb1* (which catalyzes the formation of the 2'3' cyclic phosphate on the 3' end of the oligo U-tail) is not essential. Moreover all the U6 snRNA (Fig. 7J) present in the *Usb1* KO cells have extended U-tails (about 6 nts). These tails will bind Lsm2-8 and hence promote splicing. Note that in wild-type cells U6 snRNA (which has a 5 nt U tail with a 2'3' phosphate), has an identical mobility with the 4U in vitro transcribed RNA, and no data is shown for the position of U6 with no U tails.

Why is TUT1 lethal and *Usb1* has minor phenotypes? To me it seems likely that in the *Usb1* mutant cells, there is removal of the U-tail put on by TUT1 by a 3' to 5' exonuclease (since the cyclic phosphate will block degradation, and continuous regeneration of the U-tail by Tut1, to maintain sufficient levels of active U6 snRNA bound to Lsm2-8. The role of *Usb1* is to stabilize the U-tail (blocking the exonuclease) and promoting the binding of Lsm 2-8 to the U-tail, which presumably would also slow exonuclease activity. In the absence of *Usb1*, the tails are longer, but they are almost certainly being continuously removed and then extended again. Likely this results in a lower concentration of functional U6 snRNP, and the milder phenotypes observed.

Note that when one is studying rare human inherited "diseases", it is very common that mutations in proteins that are constitutively expressed result in neurological defects, since the brain and nervous system are the most complex system and govern the function of all other tissues. There is no reason to emphasize the effects are on stem cells. They show the effects are on all cultured cells. As one example, the same is true for mutations in many of the subunits of the integrator complex, involved in RNA pol II pause/release in every cell and thousands of genes. Mutations in humans that have been reported all affect the brain.

Thank you for your insightful comments. This manuscript is the first to address the physiological functions of *Tut1* and *Usb1* in mouse models. We are working on generating *Tut1* and *Usb1* conditional knockout models in other tissues. Hopefully, we will be able to address some of your comments in subsequent studies.

SPECIFIC COMMENTS:

1. In the abstract they should indicate that the oligo U-tail on U6 is bound to Lsm 2-8, which is also an important contributor to the function of U6 snRNP.: The 3' tail actually consists of the oligo(U) tail, the 2'3' phosphate and the Lsm2-8 complex which together provide the functions of the tail.

“Lsm2-8 protein complex” has been added in Abstract in the revised manuscript.

2. Similarly at the bottom of page 4, they should make it clear that the modified 3' end is involved directly in U6 snRNA binding to Lsm 2-8; Note that it is not essential for binding, but enhances the binding affinity by a factor of 5-10, and this reference should be cited:Montemayor, E. J. et al. Molecular basis for the distinct cellular functions of the Lsm1-7 and Lsm2-8 complexes. RNA 26, 1400-1413 (2020).

Following your suggestion, the text has been modified in the revised manuscript as follows:

In Introduction, Line 71-73:

“U6 snRNA can bind to Lsm2-8 protein complex without a 2',3'-cyclic phosphate group *in vitro*. However, this modification greatly enhances the affinity of Lsm2-8 to U6 snRNA, enabling further assembly of functional spliceosome for RNA splicing.”

In Introduction, Line 74:

The study by Montemayor et al. has been cited in the revised manuscript.

3. pg. 6, l. 105. The viability of the 3.5 day embryo could be due to remaining maternal supply of TUT1 protein or U6 snRNP. Indeed their results show that KO of TUT1 is lethal in cultured ES cells as well. The shRNA experiments support this possibility, since in these cells a great reduction of TUT1 is lethal, although there is clearly a small amount of protein present by their Western blot.

3. The p53 results show that the cell damage is detected by p53 which then induces apoptosis. I presume that in the absence of p53 the cells stop growing and ultimately die.

Thank you for raising this interesting point. At this stage, we do not know

whether potential maternal effect could account for the viability of E3.5 *Tut1* knockout embryos. Further investigation will be needed to address this possibility.

As for p53, our results demonstrate that in the absence of p53, *Tut1* knockout neural stem cells continue to grow *in vitro* and *in vivo* (Figure 3J, 3K and 3P). We assume that you meant “in the presence of”, and if so, we agree with you that the activation of p53 induces cell cycle arrest and cell death.

4. In line 351 they say "Thus, these splicing events are directly regulated by Tut1 (Fig. 4H)".

This statement and any other statements using the word "regulate" as one of TUT1's function are incorrect. TUT1 is required for formation of active U6 snRNP. If there are low levels of U6 snRNP, then many splicing events, particularly those that may be less efficient will not occur. This is not "regulation."

In the revised manuscript (Line 335), the text has been changed to “Thus, these defective splicing events are caused by *Tut1* loss (Fig. 4H).” In addition, in several places in the revised manuscript, “regulate” has been replaced by “require” or “*Tut1*-dependent”.

5. They end the results section with the sentence "These results challenge the current model of post-transcriptional maturation of U6 snRNA, and position Tut1, but not Usb1, as an essential regulator of RNA splicing during mammalian embryonic development"

The sentence should be deleted. Their results don't challenge the current model. TUT1 is not an essential regulator of splicing. It is essential for the extension of U6 snRNA to allow Lsm2-8 binding and mature U6 snRNP, which is part of the current model, but it certainly doesn't regulate anything.

In the revised manuscript, this sentence has been deleted.

6. In the discussion 1.467-480, they provide strong evidence that TUT1 is essential in stem cells for splicing, but it also seems to be essential for all cultured cells. They cannot conclude there is anything special about TUT1 in stem cells. The proper conclusion is lines 480-483 "TUT1, but not USB1, is a ubiquitous, essential U6 snRNA 3'-end maturation or repair factor required for cell proliferation". Their study is the first to do this, and agrees with the DepMap public data.

In the revised manuscript, to be more precise, the text has been changed to “to maintain embryonic and neural stem cells” (Line 482-483), considering that we have only investigated these two cell types.

7. The authors are correct in that "the Mellman et al., paper is incorrect", and all other papers about STAR-PAP, all of which came from one group, are incorrect (l. 519).

In the revised manuscript, the text “from one laboratory” has been deleted.

Prof. Qintong Li
Sichuan University
West China Second University Hospital
No.20, Section 3, Renmin South Road
Chengdu, Sichuan 610041
China

Dear Prof. Li,

Thank you for the submission of your final revised manuscript to EMBO reports. I now went through it and your final p-b-p-response and consider the remaining referee concerns and the editorial requests as adequately addressed.

I am thus very pleased to accept your manuscript for publication in the next available issue of EMBO reports. Thank you for your contribution to our journal.

You may qualify for financial assistance for your publication charges - either via a Springer Nature fully open access agreement or an EMBO initiative. Check your eligibility: <https://link.springer.com/journal/44319/how-to-publish-with-us>

Yours sincerely,

>>> Please note that it is EMBO Reports policy for the transcript of the editorial process (containing referee reports and your response letter) to be published as an online supplement to each paper. If you do NOT want this, you will need to inform the Editorial Office via email immediately. More information is available here: <https://link.springer.com/partners/embo-press/editorial-policies#Peer%20review>